# CORRELATIONAL LAGRANGIAN SCHRÖDINGER BRIDGE: LEARNING DYNAMICS WITH POPULATION-LEVEL REGULARIZATION

## ABSTRACT

Modeling population dynamics is a fundamental problem with broad scientific applications. Motivated by real-world applications including biosystems with diverse populations, we consider a class of population dynamics modeling with two technical challenges: (i) dynamics to learn for individual particles are *heterogeneous* and (ii) available data to learn from are not *time-series* (i.e, each individual's state trajectory over time) but *cross-sectional* (i.e, the whole population's aggregated states without individuals matched over time). To address the challenges, we introduce a novel computational framework dubbed **correlational Lagrangian Schrödinger bridge** (**CLSB**) that builds on optimal transport to "bridge" cross-sectional data distributions. In contrast to prior methods regularizing all individuals' transport "costs" and then applying them to the population *homogeneously*, CLSB directly regularizes *population* cost allowing for population *heterogeneity* and potentially improving model generalizability. Specifically our contributions include **(1)** a novel population perspective of the transport cost and a new class of population regularizers capturing the temporal variations in multivariate relations, with the tractable formulation derived, **(2)** three domain-informed instantiations of population regularizers on covariance, and **(3)** integration of population regularizers into data-driven generative models as constrained optimization and an approximate numerical solution, with further extension to conditional generative models. Empirically, we demonstrate the superiority of CLSB in single-cell sequencing data analyses (including cell differentiation and drug-conditioned cell responses) and opinion depolarization. Codes will be released upon acceptance.

## 1 INTRODUCTION

Population dynamics sheds insight on the temporal evolution of systems, such as cytodynamics (La Manno et al., 2018), fluid mechanics (Kundu et al., 2015) and single-cell omics (Macosko et al., 2015), yet their direct observation is often restricted. Motivated by such real-world systems, this paper targets generative population-dynamics models for heterogeneous populations whose states are not available to track individual trajectories (time-series data) but only observed at the population level at times (cross-sectional data as referred to in (Tong et al., 2020; Koshizuka & Sato, 2022)). In the cross-sectional setting, states of a population are measured at each timestamp without individual match or even population match across timestamps. In other words, the cross-sectional data are sampled independently at various timestamps rather than jointly across timestamps. One such example is single-cell omics that study cell populations behaviours with unprecedented data (Gaston & Spicer, 2013; Purvis & Hector, 2000): As each measurement at any timestamp is made with cells fixed and stained or chemically destroyed, measurements across time or condition can only be observed from different samples of the cell population but not individual trajectories of the same set of cells (e.g., developmental/immun omics (Keller, 2005; Schluter et al., 2020)).

Lacking individual trajectory data for direct supervision, current machine learning methods attempt to "bridge" among cross-sectional distributions under certain principles, such as optimal transport (Villani et al., 2009; Santambrogio, 2015). To characterize the evolutionary nature of the system, besides matching the cross-sectional distributions, these methods also regularize certain transport costs, which are typically determined by the domain knowledge of the system. These costs are asso-

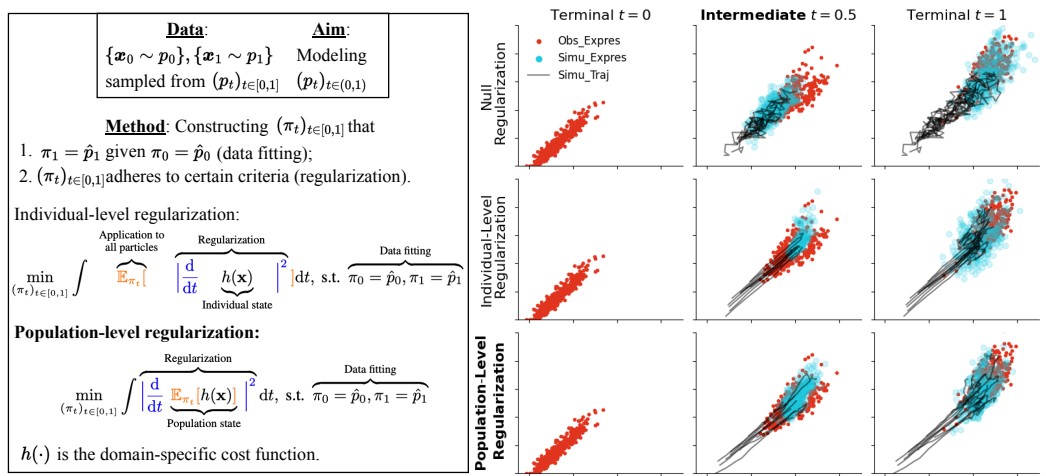

**Figure 1:** (Left) Overview of the proposed approach and (right) computational simulation of the expressions of gene ABCA3 (x-axis) and A1BG (y-axis) during the embryonic stem cell development under different regularizations. Notably, our proposed population-level regularization facilitates a more accurate modeling of distributions, with quantitative evidence detailed in Sec. 4 and more visualization in Appdx. I.

ciated with certain physical quantities on individual particle's states, such as the restraint on particle motions (Schiebinger et al., 2019; Yang & Uhler, 2018), or the alignment to empirical densities or velocities (Tong et al., 2020; Koshizuka & Sato, 2022). However, some physical quantities are only defined in the population level. For example of gene co-expressions: Gene expression covariance is only among a population of cells, while each individual cell has different/heterogeneous gene expressions. Uniformly restraining the states of individuals is thus oblivious to such knowledge.

To fill such a gap, we hypothesize that principled regularizers, if directly and appropriately formulated for the states of the population (as opposed to the states of individuals), can lead to more accurate modeling of dynamics for heterogeneous systems. The rationale of the hypothesis is directly related to the needs: As the ensemble statistics of individual states, population states (i) respect the diversity (heterogeneity) of individual states, and importantly, (ii) can accommodate domain priors previously not utilized at the population level, , e.g., the co-expression relations among genes of cellular systems, derived from bulk sequencing techniques (Stuart et al., 2003; Horvath & Dong, 2008). A nutshell overview of individual v.s. population restraint can be found in Fig. 1 (left).

**Contributions.** We propose a novel learning framework dubbed **correlational Lagrangian Schrödinger bridge** (**CLSB**) to model the dynamics of heterogeneous systems from *cross-sectional data using principled regularization at the population level* (see Fig. 1 for an overview). To the best of our knowledge, CLSB is the *first* framework to incorporate population-level domain prior into diffusion Schrödinger bridge for generative dynamics, with *substantial* benefits demonstrated for heterogeneous systems including biosystems. Specifically we make the following contributions.

**(1)** A novel perspective of the principled regularizer for transport cost with tractable formulations. *How to formulate the principled regularizer at the population level?* Motivated by the principle of least action (Schiebinger et al., 2019; Yang & Uhler, 2018), we propose to *conserve certain population states* when bridging across cross-sectional data, where the extent of conservation is measured by the temporal variations in certain statistical characteristics. Accordingly, we introduce a new category of population regularizers termed *correlational Lagrangian*, which is designed to capture the extent of temporal changes in multivariate relations expressed as moments (Parzen, 1999; Kumar & Varaiya, 2015) (Sec. 3.1). As the novel population regularizer poses a challenge of intractability that it cannot be computed numerically, we derive a computationally tractable formulation by applying the Fokker-Planck equation (Risken, 1996) with mild assumptions (Sec. 3.2).

**(2)** Effective domain-informed instantiations of population regularization. *How to instantiate population regularizers with domain-informed priors?* The generic formulation of the correlational Lagrangian is highly versatile, that is able to characterize arbitrary multivariate relations in arbitrary orders. Inspired by the concept of *co-expression stability* in genetics that the co-expression rela-

tions among genes should be robust to environments (Patil et al., 2011; Srihari & Leong, 2012), we propose to enforce temporal conservation on the states of covariance, focusing on the 1st- and 2nd-order variations of bivariate statistics, termed *covariance kinetics* (Sec. 3.3.1). We also leverage the existing evidence of co-expression by constructing *covariance potential*, which enforces alignment between the modeled covariance and the observed interactions from literature (Sec. 3.3.2).

(**3**) Practical numerical solution to model training. *How to integrate population regularizers with data-driven generative modeling of dynamics?* We formulate a *constrained optimization* problem referred to as CLSB, which is designed to minimize correlational Lagrangian subject to the constraints imposed by cross-sectional observations, optimizing on the parametrized dynamics using neural stochastic differential equations (SDEs) (Li et al., 2020; Tzen & Raginsky, 2019). To solve CLSB, we propose a *numerical approximation* via unconstrained optimization (Sec. 3.4).

Furthermore, we extend the CLSB framework into conditional dynamics generation, by re-engineering neural SDEs for taking the additional conditions as inputs. Empirically, we validate that CLSB outperforms state-of-the-art competitors in the experiments of (unconditional) developmental simulation and (conditional) drug-response prediction in cellular systems (Sec. 4). Population regularizers also showed benefits in opinion depolarization (Appdx. I).

## 2 PRELIMINARIES

**Data generation from dynamics.** The main notation used in the paper is described in Tab. 1. Let's assume that data are generated from a stochastic process $(\mathbf{x}_t)_{t\in[0,1]}$ following the distribution $(p_t)_{t\in[0,1]}$ and obeying the dynamics below:

$$\mathrm{d}\mathbf{x}_t = \boldsymbol{f}_t(\mathbf{x}_t)\mathrm{d}t + \boldsymbol{G}_t(\mathbf{x}_t)\mathrm{d}\mathbf{w}_t, \quad (1)$$

where $\mathbf{x}_t \in \mathbb{R}^d$, $\boldsymbol{f}_t : \mathbb{R}^d \to \mathbb{R}^d$ is the drift function, $(\mathbf{w}_t)_{t\in[0,1]}$ is a Wiener process in $\mathbb{R}^{d_{\mathrm{wie}}}$, and $\boldsymbol{G}_t : \mathbb{R}^d \to \mathbb{R}^{d\times d_{\mathrm{wie}}}$ is the diffusion function. Consequently, the evolution of marginal distribution $p_t$ satisfies the Fokker–Planck equation (Risken, 1996) as:

**Table 1:** Notation settings.

| Notations | Descriptions |
|---|---|
| Upright letters ($\mathbf{x}$) | Random variables |
| Italicized letters ($\boldsymbol{x}$) | Their realizations |
| Lowercase boldfaced ($\boldsymbol{x}$) | Vectors |
| Uppercase boldfaced ($\boldsymbol{X}$) | Matrices |
| Lowercase non-boldfaced ($x$) | Scalars |
| Superscripts with brackets ($\boldsymbol{x}^{(i)}$) | For multiple realizations |
| Subscripts with square brackets ($\boldsymbol{x}_{[i]}$) | For indexed elements |
| $\nabla$ | Divergence operator |
| $\cdot$ | Inner product |
| $s$ | Number of Time Stamps |
| $d$ | Variable Dimensionality |
| $m, k$ | Order of Prior (in Derivative) |
| $\tilde{\mathcal{M}}$ | Multiset of Varible Indices |
| $\mathcal{M}$ | Set Collection of Multisets $\tilde{\mathcal{M}}$ |

$$\frac{\partial}{\partial t}p_t(\boldsymbol{x}) = -\nabla \cdot (p_t(\boldsymbol{x})\boldsymbol{f}_t(\boldsymbol{x})) + \frac{1}{2}(\nabla\nabla^\top) \cdot \Big(p_t(\boldsymbol{x})\boldsymbol{G}_t(\boldsymbol{x})\boldsymbol{G}_t^\top(\boldsymbol{x})\Big). \quad (2)$$

**Generative modeling via Schrödinger bridge.** As stated in Eq. (2), the distribution $(p_t)_{t\in[0,1]}$ is characterized by the drift and diffusion terms in Eq. (1). Thus, by parametrizing $\boldsymbol{f}_t(\cdot)$ and $\boldsymbol{G}_t(\cdot)$ with neural networks $\boldsymbol{v}_t(\cdot;\theta)$ and $\boldsymbol{\Sigma}_t(\cdot;\theta)$, respectively, and with the observations from the finite-dimensional distribution as $\mathcal{D}_{\mathrm{fdim}} = \{\boldsymbol{x}_t^{(i)} : t \in \{t_1,...,t_s\}, i \in \{1,...,n\}, (\boldsymbol{x}_{t_1}^{(i)},...,\boldsymbol{x}_{t_s}^{(i)}) \sim p_{t_1,...,t_s}\}$ where $t_1 = 0, t_s = 1$, a line of prior works attempt to construct the generative model $(\pi_t)_{t\in[0,1]}$ (parametrized by $\boldsymbol{v}_t(\cdot;\theta), \boldsymbol{\Sigma}_t(\cdot;\theta)$) via solving the collective form of the (static) Schrödinger bridge problem as (De Bortoli et al., 2021; Liu et al., 2022):

$$\min_{\theta} \quad \frac{1}{s-1}\sum_{i=1}^{s-1}\mathrm{KL}(\pi_{t_i,t_{i+1}}||\hat{p}_{t_i,t_{i+1}}), \qquad\qquad \text{(Trajectory Fitting)} \quad (3.1)$$

$$\text{s.t.} \quad \pi_{t_i} = \hat{p}_{t_i}, i \in \{1,...,s\}, \qquad\qquad\qquad \text{(Marginal Fitting)} \quad (3.2)$$

$$(\pi_t)_{t\in[0,1]} \text{ is induced from } \boldsymbol{v}_t(\cdot;\theta), \boldsymbol{\Sigma}_t(\cdot;\theta), t \in [0,1] \text{ via Eq. (1),} \quad \text{(Parametrization)} \quad (3.3)$$

where $\hat{p}_{t_1,...,t_s}$ is the empirical distribution of $\mathcal{D}_{\mathrm{fdim}}$. Conceptually, Optimization in (3) requires the parametrized $(\pi_t)_{t\in[0,1]}$ to align with the reference joint distribution $\hat{p}_{t_i,t_{i+1}}$ as well as marginal $\hat{p}_{t_i}$ of the data.

**Lagrangian Schrödinger bridge for cross-sectional data.** Trajectory observations $\mathcal{D}_{\mathrm{fdim}}$ from the finite-dimensional distribution are not always available. In practice, data might be only observed

from the marginal distributions $\mathcal{D}_{\mathrm{marg}} = \{\boldsymbol{x}_t^{(i)} : t \in \{t_1, ..., t_s\}, i \in \{1, ..., n\}, \boldsymbol{x}_t^{(i)} \sim p_t\}$ where $s$ is the number of time stamps which can be observed. For such cross-sectional observations, Opt. (3) is not applicable since the reference distributions $\hat{p}_{t_i, t_{i+1}}, i \in \{1, ..., s-1\}$ in the objective (3.1) are not available. Accordingly, existing solutions propose to solve an alternative optimization problem called Lagrangian Schrödinger bridge (LSB), which adopts the *principled regularizer of least action* instead to guide the evolution of dynamics as (Koshizuka & Sato, 2022; Neklyudov et al., 2023):

$$\min_{\theta} \quad \frac{1}{(s-1)d} \sum_{i=1}^{s-1} \sum_{j=1}^{d} \int_{t_i}^{t_{i+1}} L_{\mathrm{ind}}(\pi_t, j, m)\mathrm{d}t, \qquad \text{s.t.} \quad \text{Constraints (3.2) \& (3.3),} \qquad (4)$$

where $\quad L_{\mathrm{ind}}(\pi_t, j, m) = \overbrace{\mathbb{E}_{\pi_t}}^{\text{Summarization}} [\ \overbrace{\left| \frac{\mathrm{d}}{\mathrm{d}t}\big((\mathrm{x}_{t,[j]})^m\big) \right|^2}^{\substack{\text{Lagrangian to measure} \\ \text{individual action}}}\ ],$ 

(Principled Regularizer for Individuals)

where $\hat{p}_{t_1,...,t_s}$ is overwrote re-wrote as the empirical distribution of $\mathcal{D}_{\mathrm{marg}}$. It is typical to set $m = 1$ and approximate the Lagrangian with expectation as $|\frac{\mathrm{d}}{\mathrm{d}t}\mathrm{x}_{t,[j]}|^2 \approx |v_{t,[j]}(\mathbf{x}_t; \theta)|^2 + \mathbf{\Sigma}_{t,[j,:]}^{\top}(\mathbf{x}_t; \theta)\mathbf{\Sigma}_{t,[j,:]}(\mathbf{x}_t; \theta)$ to restrain individual motions (Mikami, 2008; Tong et al., 2020). More related works are detailed in Appdx. C.

## 3 METHODS

We first introduce a novel regularizer from a fresh perspective of population state conservation (Sec. 3.1) and then address the intractability issue of regularizers resulting from the implicit distribution parametrization in diffusion models, by using the Fokker-Planck equation (Sec. 3.2). For practical implementation, we provide three biology-inspired instantiations of covariance regularizers (Sec. 3.3.1) and an approximate numerical solution to unconstrained optimization of the framework, while the exact solution of constrained, non-convex optimization is daunting (Sec. 3.4).

### 3.1 THE PRINCIPLE OF LEAST POPULATION ACTION

The Lagrangian Schrödinger bridge (LSB) problem (4) enforces the least actions for individual particles during the evolution, i.e., the conservation of individual states. Two assumptions could be violated in real-world applications such as single-cell omics. First, time-series data for individuals may not be available, e.g., measuring trajectories of individual cells is still technically challenging. Second, particles could be heterogeneous in nature (Gaston & Spicer, 2013; Purvis & Hector, 2000), while the simple regularizer $L_{\mathrm{ind}}(\cdot)$ is formulated for individual states (via action measurement $|\frac{\mathrm{d}}{\mathrm{d}t}(\cdot)|^2$) and then applied homogeneously to all particles (via summarization $\mathbb{E}_{\pi_t}[\cdot]$), which violates the nature of population heterogeneity.

To address the two challenges above, we propose to shift the focus of regularizers to population-level and reorient the emphasis of conservation strategies to population states. Specifically, we formulate an optimization problem with population regularizer $L_{\mathrm{pop}}(\cdot)$, by interchanging the order of action measurement $|\frac{\mathrm{d}}{\mathrm{d}t}(\cdot)|^2$ and summarization $\mathbb{E}_{\pi_t}[\cdot]$ in $L_{\mathrm{ind}}(\cdot)$ as follows:

$$\min_{\theta} \quad \frac{1}{(s-1)d} \sum_{i=1}^{s-1} \sum_{j=1}^{d} \int_{t_i}^{t_{i+1}} L_{\mathrm{pop}}(\pi_t, j, m)\mathrm{d}t, \qquad \text{s.t.} \quad \text{Constraints (3.2) \& (3.3),} \qquad (5)$$

where $\quad L_{\mathrm{pop}}(\pi_t, j, m) = \underbrace{\left| \frac{\mathrm{d}}{\mathrm{d}t} \overbrace{\mathbb{E}_{\pi_t}\big[(\mathrm{x}_{t,[j]})^m\big]}^{\text{Population state}} \right|^2}_{\text{Action measurement}},$ 

(Principled Regularizer **for Population**)

where it is assumed $\int |x_{[j]}|^m \pi_t(\boldsymbol{x})d\boldsymbol{x} < \infty, j \in \{1, ..., d\}$ (Spanos, 2019) for the interchangeability. The proposed population-level regularizer $L_{\mathrm{pop}}(\cdot)$ essentially captures the temporal variations in certain population characteristics, contrasting with the focus on individual-state dynamics in $L_{\mathrm{ind}}(\cdot)$. In this context, the population state is quantified by the $m$th-order moment of each variable $j$, the determinacy of which is studied in the Hamburger moment problem (Shohat & Tamarkin, 1950; Akhiezer, 2020). Thus, Opt. (5) aims to find the evolution $(\pi_t)_{t \in [t_i, t_{i+1}]}$ between terminal distributions $\pi_{t_i} = \hat{p}_{t_i}$ and $\pi_{t_{i+1}} = \hat{p}_{t_{i+1}}$ such that the characteristics of distributions evolve smoothly.

Moving beyond Opt. (5) we will present a conceptually more complete and computationally tractable formulation for population regularizers in the next subsections.

## 3.2 Correlational Lagrangian Schrödinger Bridge

**Conservation of correlation during evolution.** The initial extension from individual to population regularization in Opt. (5) falls short in capturing the relations among data variables, which accounts for a rich family of observed behaviors especially in biological systems (Patil et al., 2011; Srihari & Leong, 2012). Thus, we propose to extend the objective formulation in Opt. (5) by involving multivariate relations, resulting in the optimization problem referred as correlational Lagrangian Schrödinger bridge (CLSB) as:

$$\min_{\theta} \quad \frac{1}{(s-1)|\mathcal{M}|} \sum_{i=1}^{s-1} \sum_{\widetilde{\mathcal{M}} \in \mathcal{M}} \int_{t_i}^{t_{i+1}} L_{\text{corr}}(\pi_t, \widetilde{\mathcal{M}}, k)\mathrm{d}t, \qquad \text{s.t.} \quad \text{Constraints (3.2) \& (3.3)} \qquad (6)$$

where $\quad L_{\text{corr}}(\pi_t, \widetilde{\mathcal{M}}, k) = \Big| \underbrace{\frac{\mathrm{d}^k}{\mathrm{d}t^k} \overbrace{\mathbb{E}_{\pi_t}[ \prod_{(j,m)\in\widetilde{\mathcal{M}}} (\mathrm{x}_{t,[j]})^m]}^{\text{Correlational characteristic}}}_{\text{Correlational Lagrangian}} \Big|^2,$   (Population Regularizer **on Multivariate Correlation**)

where $k \in \mathbb{Z}_>$, $\mathcal{M} = \{..., \widetilde{\mathcal{M}}_i, ...\}$ that each $\widetilde{\mathcal{M}}_i = \{(j, \bar{m}_i(j)) : j \in \bar{\mathcal{M}}_i \subset \{1, ..., d\}, \bar{m}_i : \{1, ..., d\} \to \mathbb{Z}_>\}$ is a multiset consisting of variable indices and their corresponding occurrences, identifying the targeted multivariate relation which is quantified by the mixed moment (Parzen, 1999; Kumar & Varaiya, 2015). We refer to $L_{\text{corr}}(\cdot)$ as *correlational Lagrangian* capturing temporal variations in multivariate correlations.

The CLSB formulation (6) is the more general framework, capable of imposing domain priors for arbitrary multivariate relations (specified by $\widetilde{\mathcal{M}}$ in $L_{\text{corr}}(\cdot)$) in arbitrary order (specified by $k$). For instance, by setting $k = 1$ and $\mathcal{M} = \{\widetilde{\mathcal{M}}_j : \widetilde{\mathcal{M}}_j = \{(j, m)\}, j = 1, ..., d\}$, CLSB degenerates to Opt. (5) that involves null multivariate correlations.

**Analytical expression of correlational Lagrangian.** The current formulation of correlational Lagrangian $L_{\text{corr}}(\cdot)$ in Opt. (6) is not yet in a tractable form for practical implementation due to the existence of the $k$-order time derivative. Our following proposition provides the derivation of tractable analytical expressions under mild assumptions.

**Proposition 1.** For $k = 1$, correlational Lagrangian in Opt. (6) admits the analytical expression as:

$$L_{\text{corr}}(\pi_t, \widetilde{\mathcal{M}}, 1) = \Big| \mathbb{E}_{\pi_t}[\nabla \Big( \prod_{(j,m)\in\widetilde{\mathcal{M}}} (\mathrm{x}_{t,[j]})^m \Big) \overbrace{\cdot \boldsymbol{v}_t(\mathbf{x}_t; \theta)}^{\substack{\text{Variation resulting} \\ \text{from drift } \boldsymbol{v}_t(\cdot;\theta)}}]$$
$$+ \frac{1}{2}\mathbb{E}_{\pi_t}[(\nabla\nabla^\top \Big( \prod_{(j,m)\in\widetilde{\mathcal{M}}} (\mathrm{x}_{t,[j]})^m \Big)) \overbrace{\cdot(\boldsymbol{\Sigma}_t(\mathbf{x}_t; \theta)\boldsymbol{\Sigma}_t^\top(\mathbf{x}_t; \theta))]}^{\substack{\text{Variation resulting} \\ \text{from diffusion } \boldsymbol{\Sigma}_t(\cdot;\theta)}} \Big|^2, \qquad (7)$$

if for the set of functions $\mathcal{H} = \{h(\boldsymbol{x})\pi_t(\boldsymbol{x})\boldsymbol{v}_t(\boldsymbol{x}), \pi_t(\boldsymbol{x})\boldsymbol{D}(\boldsymbol{x})\nabla h(\boldsymbol{x}), \pi_t(\boldsymbol{x})\nabla^\top \boldsymbol{D}_t(\boldsymbol{x})h(\boldsymbol{x}), h(\boldsymbol{x})\boldsymbol{D}_t(\boldsymbol{x})\nabla \pi_t(\boldsymbol{x})\}$ ($\theta$ is omitted for simplicity) that $h(\boldsymbol{x}) = \prod_{(j,m)\in\widetilde{\mathcal{M}}} (\mathrm{x}_{t,[j]})^m$, $\boldsymbol{D}_t(\boldsymbol{x}) = \boldsymbol{\Sigma}_t(\boldsymbol{x})\boldsymbol{\Sigma}_t^\top(\boldsymbol{x})$, it satisfies: (i) Continuity: $h' \in \mathcal{H}$ is continuously differentiable w.r.t. $\boldsymbol{x}$; (ii) Light tail: The probability density function $\pi_t(\boldsymbol{x})$ is characterized by tails that are sufficiently light, such that $\oint_{S_\infty} h'(\boldsymbol{x}) \cdot \mathrm{d}\boldsymbol{a} = 0$ for $h' \in \mathcal{H}$, where $\boldsymbol{a}$ is the outward pointing unit normal on the $S_\infty$ boundary.

For $k \geq 2$, correlational Lagrangian $L_{\text{corr}}(\pi_t, \widetilde{\mathcal{M}}, k)$ in Opt. (6) admits a more complex analytical expression, which can be derived iteratively in a similar way if certain conditions of continuity and light tails are met. The detailed formulation is postponed to Appdx. A to avoid distraction.

*Proof.* See Appdx. A.

The key step in the derivation involves applying the Fokker–Planck equation (2) to establish a connection between the temporal variation $\frac{\mathrm{d}}{\mathrm{d}t}\mathbb{E}_{\pi_t}[\cdot]$ and the (parametrized) force field $\boldsymbol{v}_t(\cdot;\theta), \boldsymbol{\Sigma}_t(\cdot;\theta)$.

Both conditions are moderate and can be easily ensured by appropriately constructing the architectures of the drift and diffusion functions (Schulz et al., 2018; Song et al., 2020). Thereby, Propos. 1 enables the tractable computation of correlational Lagrangian for practical implementation.

### 3.3 Domain-Informed Instantiations of Correlational Lagrangian in Biological Systems

#### 3.3.1 Covariance Kinetics

**Conserving bivariate relations for co-expression stability.** Existing literature in genetics indicates the phenomenon of co-expression stability, i.e., the co-expression among genes could be robust to environments (Patil et al., 2011; Srihari & Leong, 2012). We numerically validate such phenomena in our dataset (see Appdx. D for details). We are therefore inspired to incorporate such prior into the population regularizer, by focusing correlational Lagrangian specifically on bivariate relations, thereby restraining temporal variations of the states of covariance. We term it as *covariance kinetics* to demonstrate the idea of restricting the "motion" of the population (Frost & Pearson, 1961; Lifschitz & Pitajewski, 1983), with two specific instantiations as follows.

**Instantiation 1: Restraining the "velocity" of covariance.** The first instantiation to enforce models simulate stably co-expressed genes in cells, is through restraining the 1st-order moment of the covariance, such that it (representing co-expression relations) changes slowly during temporal evolution. In formulation, denoting $\mathcal{M}_{\text{cov}} = \left\{ \{(i,1),(j,1)\} : i \in \{1,...,d\}, j \in \{1,...,d\} \right\}$ for all the pairs among $d$ variables, the objective in Opt. (6) is analytically instantiated as:

$$\sum_{\widetilde{\mathcal{M}} \in \mathcal{M}_{\text{cov}}} L_{\text{corr}}(\pi_t, \widetilde{\mathcal{M}}, 1) = \left\| \frac{\mathrm{d}}{\mathrm{d}t} \mathbb{E}_{\pi_t}[\mathbf{x}_t \mathbf{x}_t^\top] \right\|_{\mathsf{F}}^2$$

$$= \left\| \mathbb{E}_{\pi_t}[\mathbf{x}_t \boldsymbol{v}_t(\mathbf{x}_t)^\top + \boldsymbol{v}_t(\mathbf{x}_t)\mathbf{x}_t^\top + \frac{1}{2}\boldsymbol{\Sigma}_t(\mathbf{x}_t)\boldsymbol{\Sigma}_t^\top(\mathbf{x}_t)] \right\|_{\mathsf{F}}^2. \quad (8)$$

**Instantiation 2: Restraining the "acceleration" of covariance.** The second instantiation is more relaxed, allowing greater temporal variation in co-expression, which however, should not be "irregular". We achieve this by restraining the second-order moment of the covariance, ensuring that it evolves "regularly" during dynamics, with the objective formulated as:

$$\sum_{\widetilde{\mathcal{M}} \in \mathcal{M}_{\text{cov}}} L_{\text{corr}}(\pi_t, \widetilde{\mathcal{M}}, 2) = \left\| \frac{\mathrm{d}^2}{\mathrm{d}t^2} \mathbb{E}_{\pi_t}[\mathbf{x}_t \mathbf{x}_t^\top] \right\|_{\mathsf{F}}^2$$

$$= \left\| \mathbb{E}_{\pi_t}\left[ \mathbf{x}_t(\frac{\mathrm{d}}{\mathrm{d}t}\boldsymbol{v}_t(\mathbf{x}_t))^\top + (\frac{\mathrm{d}}{\mathrm{d}t}\boldsymbol{v}_t(\mathbf{x}_t))\mathbf{x}_t^\top + \frac{1}{2}\frac{\mathrm{d}}{\mathrm{d}t}(\boldsymbol{\Sigma}_t(\mathbf{x}_t)\boldsymbol{\Sigma}_t^\top(\mathbf{x}_t)) \right]\right.$$

$$+ \mathbb{E}_{\pi_t}\left[ \mathbf{x}_t(\nabla\boldsymbol{v}_t(\mathbf{x}_t)\boldsymbol{v}_t(\mathbf{x}_t))^\top + (\nabla\boldsymbol{v}_t(\mathbf{x}_t)\boldsymbol{v}_t(\mathbf{x}_t))\mathbf{x}_t^\top + 2\boldsymbol{v}_t(\mathbf{x}_t)\boldsymbol{v}_t(\mathbf{x}_t)^\top + \frac{1}{2}\nabla(\boldsymbol{\Sigma}_t(\mathbf{x}_t)\boldsymbol{\Sigma}_t^\top(\mathbf{x}_t))_{\underline{i_1}i_2i_3}\boldsymbol{v}_t^{i_3}(\mathbf{x}_t) \right]$$

$$+ \mathbb{E}_{\pi_t}\left[ \nabla\boldsymbol{v}_t(\mathbf{x}_t)\boldsymbol{\Sigma}_t(\mathbf{x}_t)\boldsymbol{\Sigma}_t^\top(\mathbf{x}_t) + \boldsymbol{\Sigma}_t(\mathbf{x}_t)\boldsymbol{\Sigma}_t^\top(\mathbf{x}_t)\nabla^\top\boldsymbol{v}_t(\mathbf{x}_t) + \frac{1}{2}\mathbf{x}_t(\nabla\nabla^\top(\boldsymbol{v}_t(\mathbf{x}_t))_{\underline{i_1}i_2i_3}(\boldsymbol{\Sigma}_t(\mathbf{x}_t)\boldsymbol{\Sigma}_t^\top(\mathbf{x}_t))^{\underline{i_2i_3}})^\top \right.$$

$$\left. + \frac{1}{2}(\nabla\nabla^\top(\boldsymbol{v}_t(\mathbf{x}_t))_{\underline{i_1}i_2i_3}(\boldsymbol{\Sigma}_t(\mathbf{x}_t)\boldsymbol{\Sigma}_t^\top(\mathbf{x}_t))^{\underline{i_2i_3}})\mathbf{x}_t^\top + \frac{1}{4}\nabla\nabla^\top(\boldsymbol{\Sigma}_t(\mathbf{x}_t)\boldsymbol{\Sigma}_t^\top(\mathbf{x}_t))_{\underline{i_1}i_2i_3i_4}(\boldsymbol{\Sigma}_t(\mathbf{x}_t)\boldsymbol{\Sigma}_t^\top(\mathbf{x}_t))^{\underline{i_3i_4}}\right]\right\|_{\mathsf{F}}^2, \quad (9)$$

where we adopt the Einstein notation $\boldsymbol{C} = \boldsymbol{A}_{ij}\boldsymbol{B}^{\underline{jk}}$ (Barr, 1991) for tensor operations that $\boldsymbol{C}_{[i,k]} = \sum_j \boldsymbol{A}_{[i,j]}\boldsymbol{B}_{[j,k]}$. The matrix-form derivations of instantiations (8) & (9) are based on Propos. 1, and we provide a more detailed explanation of the derivation of complicated Eq. (9) in Appdx. B.

**Standardized covariance kinetics within a projected space.** The original covariance may be sensitive to dataset-dependent parameters, such as batch effects in sequencing techniques (Zhang et al., 2019; Luo et al., 2010), which can limit its generalizability. We have further re-wrote the regularization (8) & (9) to account for the standardized covariance, which is more robust and better encompasses co-expression priors across different datasets. The re-written objective for the velocity term (8) is formulated as:

$$\sum_{\widetilde{\mathcal{M}} \in \mathcal{M}_{\text{cov}}} L_{\text{corr-std}}(\pi_t, \widetilde{\mathcal{M}}, 1) = \left\| \frac{\mathrm{d}}{\mathrm{d}t}\left( \mathbb{E}_{\pi_t}[\mathbf{x}_t \mathbf{x}_t^\top] - \mathbb{E}_{\pi_t}[\mathbf{x}_t]\mathbb{E}_{\pi_t}^\top[\mathbf{x}_t] \right) \right\|_{\mathsf{F}}^2$$

$$= \left\| \mathbb{E}_{\pi_t}[\mathbf{x}_t \boldsymbol{v}_t(\mathbf{x}_t)^\top + \boldsymbol{v}_t(\mathbf{x}_t)\mathbf{x}_t^\top + \frac{1}{2}\boldsymbol{\Sigma}_t(\mathbf{x}_t)\boldsymbol{\Sigma}_t^\top(\mathbf{x}_t)] - \mathbb{E}_{\pi_t}[\mathbf{x}_t]\mathbb{E}_{\pi_t}^\top[\boldsymbol{v}_t(\mathbf{x}_t)] - \mathbb{E}_{\pi_t}[\boldsymbol{v}_t(\mathbf{x}_t)]\mathbb{E}_{\pi_t}^\top[\mathbf{x}_t] \right\|_{\mathsf{F}}^2. \tag{10}$$

Similarly for the acceleration term (9), the standardized formulation is detailed in Appdx. E.

Furthermore, the regularizer constructed with domain-specific priors operates in a space that may not correspond to that of the data (where the diffusion generative model is built in). For instance, high-dimensional single-cell sequencing data often undergoes principal component analysis (Wold et al., 1987) prior to further processing. In this scenario, when gene expressions are linearly mapped from the principal components as $\boldsymbol{x}_{\mathrm{gene}} = \boldsymbol{W}\boldsymbol{x} + \boldsymbol{b}$, the projected (and standardized) $k$-th order covariance kinetics are then represented in matrix form as:

$$\sum_{\widetilde{\mathcal{M}} \in \mathcal{M}_{\mathrm{cov}}} L_{\mathrm{corr\text{-}std\text{-}linproj}}(\pi_t, \widetilde{\mathcal{M}}, k) = \left\| \boldsymbol{W} \frac{\mathrm{d}^k}{\mathrm{d}t^k} \Big( \mathbb{E}_{\pi_t}[\mathbf{x}_t \mathbf{x}_t^\top] - \mathbb{E}_{\pi_t}[\mathbf{x}_t]\mathbb{E}_{\pi_t}^\top[\mathbf{x}_t] \Big) \boldsymbol{W}^\top \right\|_{\mathsf{F}}^2. \tag{11}$$

The derivation of the more complicated, non-linear projected space is detailed in Appdx. G.

### 3.3.2 COVARIANCE POTENTIAL

**Instantiation 3: Aligning covariance with observed bivariate interactions ("prior position").** There exists abundant observed evidence of co-expression relations among genes, sourced from numerous experiments which represent these interactions in a statistical context (Mering et al., 2003; Oughtred et al., 2019). We hope to leverage such prior knowledge in the generative modeling of cells. Specifically, denoting the observed co-expression as $\boldsymbol{Y} \in [0,1]^{d \times d}$ where $\boldsymbol{Y}_{[i,j]}$ is the confidence score of genes $i$ and $j$ being co-expressing, we construct the principled regularizer termed *covariance potential*, which borrows the idea of enforcing alignment with the "correct position" of the population states as:

$$\sum_{\widetilde{\mathcal{M}} \in \mathcal{M}_{\mathrm{cov}}} L_{\mathrm{corr}}(\pi_t, \widetilde{\mathcal{M}}, 0) = U\Big( \mathbb{E}_{\pi_t}[\mathbf{x}_t \mathbf{x}_t^\top], \boldsymbol{Y} \Big), \tag{12}$$

where $U(\cdot)$ is the designated potential function detailed in Appdx. F, and the notation $L_{\mathrm{corr}}(\cdot)$ is reused here, as it was previously undefined for $k = 0$.

### 3.4 NUMERICAL SOLUTIONS TO CLSB

**Approximation via unconstrained optimization.** The exact solution to CLSB (6) remains challenging despite that we provide a tractable objective in Sec. 3.3, due to its non-convex objective and constraints w.r.t. network parameters. Thus, we propose a practical, approximate solution via grappling with an unconstrained optimization problem as:

$$\min_{\theta} \quad \frac{1}{(s-1)} \sum_{i=1}^{s-1} \Big( L_{\mathrm{dist}}(\pi_{t_{i+1}}, \hat{p}_{t_i+1}) + \alpha_{\mathrm{ind}} \frac{1}{d} \sum_{j=1}^{d} \int_{t_i}^{t_{i+1}} L_{\mathrm{ind}}(\pi_t, j, 1)\mathrm{d}t$$

$$+ \sum_{k=0}^{2} \alpha_{\mathrm{corr},k} \frac{1}{|\mathcal{M}_{\mathrm{cov}}|} \sum_{\widetilde{\mathcal{M}} \in \mathcal{M}_{\mathrm{cov}}} \int_{t_i}^{t_{i+1}} L_{\mathrm{corr}}(\pi_t, \widetilde{\mathcal{M}}, k)\mathrm{d}t \Big), \tag{13}$$

where $L_{\mathrm{dist}}(\cdot)$ is the distribution discrepancy measure, and $\alpha_{\mathrm{ind}}, \alpha_{\mathrm{corr},0}, \alpha_{\mathrm{corr},1}, \alpha_{\mathrm{corr},2}$ are the weights for different regularization objectives, which are treated as hyperparameters with tuning details described in Appdx. I. Here we also adopt the individual regularizer $L_{\mathrm{ind}}(\cdot)$ and adjust its weight $\alpha_{\mathrm{ind}}$ for a more general framework encompassing Opt. (4) & (6), which is solved via gradient descent. The parametrization of neural SDEs ($\boldsymbol{v}_t(\cdot; \theta)$ and $\boldsymbol{\Sigma}_t(\cdot; \theta)$) is described in Appdx. H.

**Extension to conditional generative modeling.** We further extend CLSB into the conditional generation scenario, where we are tasked to model $(p_t(\cdot|\mathbf{c}))_{t \in [0,1]}$. The application encompasses modeling cellular systems in response to perturbations $\mathbf{c}$ such as drug treatments or genetic mutations (Srivatsan et al., 2020; Dong et al., 2023). Such extension can be achieved by re-engineering the neural SDEs $\boldsymbol{v}_t(\cdot; \theta), \boldsymbol{\Sigma}_t(\cdot; \theta)$ to input additional featurized conditions, which is re-written as $\boldsymbol{v}_t(\cdot, \boldsymbol{c}; \theta), \boldsymbol{\Sigma}_t(\cdot, \boldsymbol{c}; \theta)$, without altering the rest of the framework. We detail the neural network parametrization in Appdx. H.

## 4 EXPERIMENTS

We evaluate the proposed CLSB (13) in two real-world applications of modeling cellular systems in the unconditional (Sec. 4.1) and conditional generation scenarios (Sec. 4.2).

### 4.1 UNCONDITIONAL GENERATION: DEVELOPMENTAL MODELING OF EMBRYONIC STEM CELLS

**Data.** Deciphering the developmental behavior of cells is the quintessential goal in the field of stem cell research (Alison et al., 2002; Zakrzewski et al., 2019). The experiment is conducted on scRNA-seq data of embryonic stem cells (Moon et al., 2019), which is collected during the developmental stages over a period of 27 days, split into five phases: $t_0$ (days 0-3), $t_1$ (days 6-9), $t_2$ (days 12-15), $t_3$ (days 18-21), and $t_4$ (days 24-27). Following the setting in (Tong et al., 2020; Koshizuka & Sato, 2022), gene expressions are (linearly) projected into a lower-dimensional space through principal component analysis (PCA) (Wold et al., 1987) prior to conducting the experiments, which also can be re-projected to the original space for evaluation. We also conduct experiments on an additional cell-differentiation dataset (Weinreb et al., 2020) in Appdx. I to demonstrate the effectiveness of our method.

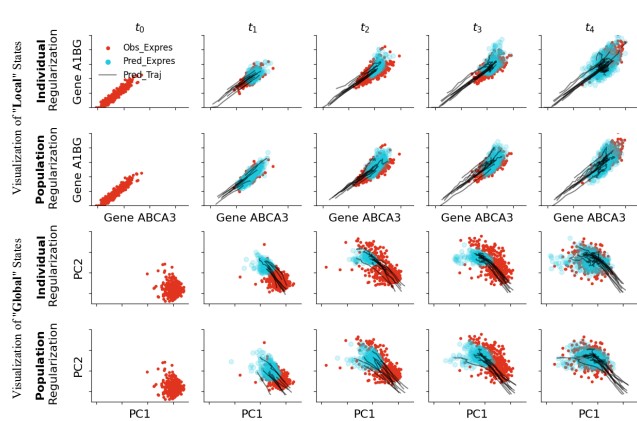

**Figure 2:** Visualization of the simulated gene expressions and trajectories with different methods. The trajectories are plotted for the gene pairs with the highest correlation (ABCA3 and A1BG), and along the first two PCs.

**Evaluation and compared methods.** To evaluate the (biological) validity of the proposed population regularizers, we conduct model training using data from the terminal stages $(t_0, t_4)$ without access to the intermediate $(t_1, t_2, t_3)$, which are held out for evaluation. Following the setting in (Tong et al., 2020; Koshizuka & Sato, 2022), the model is evaluated in the scenarios where the trajectories $\mathbf{x}_{t_i}$ are generated based on $\hat{p}_{t_0}$ (referred as "all-step" prediction) or based on $\hat{p}_{t_{i-1}}$ for $\pi_{t_i}$ ("one-step"), and the performance is quantified for the intermediate stages, based on the discrepancy in the Wasserstein distance (Villani et al., 2009; Santambrogio, 2015) between the predicted and the ground truth principal components, using the GeomLoss library (Feydy et al., 2019).

The compared baselines include random expressions sampled from a non-informative uniform distribution and simple population average across time stamps, ODE-based approaches OT-Flow (Onken et al., 2021) and TrajectoryNet (Tong et al., 2020), and SDE-based approaches DMSB (Chen et al., 2023), NeuralSDE (Li et al., 2020; Tzen & Raginsky, 2019) and NLSB (Koshizuka & Sato, 2022). The proposed CLSB falls under the category of SDE-based approaches. We adopt neural SDEs for parametrizing dynamics, with the regularization weights tuned via grid search.

**Results (i). Population regularization leads to more accurate modeling of cell developmental dynamics.** The results of developmental modeling of embryonic stem cells are shown in Tab. 2. Compared with the competitors, CLSB with population regularizers alone and without individual regularizers ($\alpha_{\text{ind}} = 0$) attains the lowest average rank, predicting developmental gene expressions closest in Wasserstein distance to the ground truth. The improvement is particularly evident in the most challenging stage of $t_2$, which is far from both end points observed at $t_0$ and $t_4$. This demonstrates the effectiveness of the proposed population regularizers in the heterogeneous systems of cell clusters. We also observe that in comparison to ODE-based methods, SDE-based ones perform better, echoing the inherently stochastic and diffusive nature of cell expression priors (Koshizuka & Sato, 2022). The predicted gene-expression trajectories are visualized and compared in Fig. 2 for two genes with the highest correlation (top two rows) and for all genes along the first two principal

**Table 2:** Evaluation in the unconditional generation scenario of modeling embryonic stem-cell development. Reported are Wasserstein distances, where lower values are preferable, with means ± standard deviations across experiments. The best and the second-best performances in each case and across cases ('A.R.' stands for average ranking) are highlighted in **red** and salmon, respectively. Methods are evaluated in the scenarios of all-step prediction on $\pi_{t_i|t_0}$ where $\pi_{t_0} = \hat{p}_{t_0}$, and one-step prediction on $\pi_{t_i|t_{i-1}}$ where $\pi_{t_{i-1}} = \hat{p}_{t_{i-1}}$, $i \in \{1, 2, 3\}$.

| Methods | All-Step Prediction | | | One-Step Prediction | | | A.R. |
|---|---|---|---|---|---|---|---|
| | $t_1$ | $t_2$ (Most Challenging) | $t_3$ | $t_1$ | $t_2$ | $t_3$ | |
| Random | 1.873±0.014 | 2.082±0.011 | 1.867±0.011 | 1.870±0.013 | 2.084±0.010 | 1.868±0.012 | 10.0 |
| SimpleAvg | 1.670±0.019 | 1.801±0.014 | 1.749±0.016 | 1.872±0.014 | 2.085±0.011 | 1.868±0.012 | 9.3 |
| OT-Flow | 1.921 | 2.421 | 1.542 | 1.921 | 1.151 | 1.438 | 9.0 |
| OT-Flow+OT | 1.726 | 2.154 | 1.397 | 1.726 | 1.186 | 1.240 | 7.6 |
| TrajectoryNet | 1.774 | 1.888 | 1.076 | 1.774 | 1.178 | 1.315 | 6.8 |
| TrajectoryNet+OT | 1.134 | 1.336 | 1.008 | 1.134 | 1.151 | 1.132 | 3.6 |
| DMSB | 1.593 | 2.591 | 2.058 | – | – | – | 10.3 |
| NeuralSDE | 1.507±0.014 | 1.743±0.031 | 1.586±0.038 | 1.504±0.013 | 1.384±0.016 | 0.962±0.014 | 6.1 |
| NLSB(E) | 1.128±0.007 | 1.432±0.022 | 1.132±0.034 | 1.130±0.007 | 1.099±0.010 | 0.839±0.012 | 2.6 |
| NLSB(E+D+V) | 1.499±0.005 | 1.945±0.006 | 1.619±0.016 | 1.498±0.005 | 1.418±0.009 | 0.966±0.016 | 6.8 |
| CLSB($\alpha_{ind} > 0$) | 1.099±0.019 | 1.419±0.028 | 1.132±0.038 | 1.098±0.018 | 1.117±0.009 | 0.826±0.010 | 2.5 |
| CLSB($\alpha_{ind} = 0$) | 1.074±0.009 | 1.244±0.016 | 1.255±0.022 | 1.095±0.009 | 1.106±0.014 | 0.842±0.012 | 2.1 |

components (bottom two rows), which also qualitatively attests to the effectiveness of population regularizers. Visualizations for more gene pairs and more principal components are provided in Appdx. I. Comparison with more SOTAs (Tong et al., 2023a;b) is also shown in Tab. 12, and experiments on dataset CITE-Seq (Kim et al., 2020) at larger scale are shown in Tab. 13.

Beyond cellular systems, we also conduct more experiments on a well-controlled synthetic dataset with results shown in Tab. 15, and of other applications of opinion depolarization (Liu et al., 2022) to validate our method with results shown in Tab. 14.

**(ii) Different population regularization strategies serve varied functions.** We further carry out ablation studies to examine the contributions of the three population regularizers to overall performance, as detailed in Appdx. I. Interestingly, we find that they serve different functions. Specifically, restraining the "acceleration" of covariance ($k = 2$, instantiation 2 (9)) provides more benefit in the early stage of development (i.e. $t_1$), and restraining the "velocity" of covariance ($k = 1$, instantiation 1 (8)) does so in the later stages (i.e. $t_2, t_3$). This observation could indicate that in nature, co-expression relations among genes undergo larger magnitude variations in early development stages, and tend to stabilize as development progresses. The benefit of aligning with known gene-gene interactions ($k = 0$, instantiation 3 (12)) is present across all stages, albeit modestly.

## 4.2 CONDITIONAL GENERATION: DOSE-DEPENDENT CELLULAR RESPONSE PREDICTION TO PERTURBATIONS

**Data.** Examining cellular responses to chemical perturbations is one of the fundamental tasks in the drug discovery process (Dong et al., 2023; Bunne et al., 2023). The experiment utilizes the sci-Plex

**Table 3:** Evaluation in the conditional generation scenario of dose-dependent cellular response prediction to chemical perturbations. Numbers indicate the mean and median Wasserstein distances on all drug conditions, and the best and the second-best performances in each case and across cases ('A.R.' stands for average ranking) are highlighted in **red** and salmon, respectively.

| Methods | $t_1$ | | $t_2$ (Most Challenging) | | $t_3$ | | A.R. |
|---|---|---|---|---|---|---|---|
| | Mean | Median | Mean | Median | Mean | Median | |
| Random | 5.236±3.349 | 4.895±4.080 | 5.215±3.416 | 5.037±4.311 | 5.247±3.346 | 5.011±4.108 | 8.0 |
| NeuralSDE(RandInit) | 2.300±1.204 | 2.235±1.212 | 2.314±1.224 | 2.285±1.332 | 2.317±1.208 | 2.265±1.259 | 7.0 |
| VAE | 1.387±0.926 | 1.144±0.676 | 1.029±0.524 | 0.935±0.453 | 0.855±0.290 | 0.804±0.294 | 4.6 |
| NeuralODE | 0.914±0.272 | 0.831±0.206 | 1.064±0.413 | 0.985±0.414 | 1.004±0.296 | 0.937±0.286 | 5.3 |
| NeuralSDE | 0.905±0.416 | 0.829±0.425 | 1.053±0.547 | 0.962±0.532 | 1.032±0.409 | 0.943±0.351 | 5.0 |
| NLSB(E) | 0.503±0.106 | 0.418±0.054 | 0.574±0.115 | 0.496±0.063 | 0.667±0.159 | 0.555±0.058 | 2.8 |
| CLSB($\alpha_{ind} > 0$) | 0.516±0.163 | 0.401±0.054 | 0.571±0.189 | 0.452±0.062 | 0.631±0.235 | 0.471±0.072 | 2.1 |
| CLSB($\alpha_{ind} = 0$) | 0.476±0.109 | 0.393±0.052 | 0.531±0.121 | 0.449±0.063 | 0.564±0.122 | 0.455±0.056 | 1.0 |

data for three cancer cell lines under different drug treatments (Srivatsan et al., 2020), where data are collected for treatment doses of 10 nM, 100 nM, 1 $\mu$M, and 10 $\mu$M. In this context, we consider the drug dose as pseudo-time (denoted as $t_1, t_2, t_3, t_4$, respectively; whereas zero-dose control is denoted as $t_0$). Gene expression dynamics is conditioned on the embedding of graph-structured drug data (see more details of datasets in Appdx. H).

**Evaluation and compared methods.** We train our model using samples from the terminal stages $(t_0, t_4)$, reserving samples from the intermediate stages $(t_1, t_2, t_3)$ for evaluation. During inference, expressions are generated based on the state at $t_0$. Performance is then assessed on the Wasserstein distance on PCs across all drug conditions, which is compared on the mean and median values. Evaluation on the original gene expressions is also provided in Appdx. I. The compared baselines include the random expressions, VAE (Kingma & Welling, 2013), NeuralODE (Onken et al., 2021), NeuralSDE (Li et al., 2020; Tzen & Raginsky, 2019), and NLSB (Koshizuka & Sato, 2022).

**Results (iv). Application of population regularization leads to more accurate prediction of perturbation effects.** The results of dose-dependent cellular response prediction to chemical perturbations are shown in Tab. 3. Compared with the competitors, CLSB with population regularization alone ($\alpha_{\text{ind}} = 0$) attains the lowest average rank, which indicates it replicates treated gene expressions in better alignment with the ground truth, and the benefit of population regularization is presented in all the three stages. This coincides the effectiveness of the proposed population regularization. We also observe the similar phenomenon that SDE-based approaches outperform ODE-based approaches, and the classical VAE. Lastly, we split the data based on drug perturbations and showed our model's superior predictions for new drugs in Tab. 5.

## 5 CONCLUSIONS

In this paper, we introduce a novel framework termed Correlational Lagrangian Schrödinger Bridge (CLSB), effectively addressing the challenges posed by restricted cross-sectional samples and the heterogeneous nature of individual particles. By shifting the focus of regularization from individual-level to population, CLSB acknowledges and leverages the inherent heterogeneity in systems to improve model generalizability. In developing CLSB, we address the technical challenges including (1) a new class of population regularizers capturing with the tractable formulation, (2) domain-informed instantiations, and (3) the integration of into data-driven generative models. Numerically, we validate the superiority of CLSB in modeling cellular systems.

Admittedly, there are remaining gaps that need to be filled in the future. These include, but are not limited to, the reliance on the domain-informed priors of CLSB instantiations (Sec. 3.3) and the approximability of the numerical solution (Sec. 3.4). In broader impacts, the proposed approach could be used to help develop new treatments, such as for cancer cells.

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

APPENDIX

## A  PROOF FOR PROPOSITION 1

**Proposition 1.** For revisit, correlational Lagrangian is defined as:

$$L_{\text{corr}}(\pi_t, \widetilde{\mathcal{M}}, k) = \left| \frac{\mathrm{d}^k}{\mathrm{d}t^k} \mathbb{E}_{\pi_t}[ \prod_{(j,m)\in\widetilde{\mathcal{M}}} (\mathrm{x}_{t,[j]})^m ] \right|^2.$$

For $k=1$, correlational Lagrangian in Opt. (6) admits the analytical expression as:

$$L_{\text{corr}}(\pi_t, \widetilde{\mathcal{M}}, 1) = \left| \mathbb{E}_{\pi_t}[\nabla\Big( \prod_{(j,m)\in\widetilde{\mathcal{M}}} (\mathrm{x}_{t,[j]})^m \Big) \cdot \boldsymbol{v}_t(\mathbf{x}_t; \theta)] \right.$$
$$\left. + \frac{1}{2}\mathbb{E}_{\pi_t}[(\nabla\nabla^\top\Big( \prod_{(j,m)\in\widetilde{\mathcal{M}}} (\mathrm{x}_{t,[j]})^m \Big)) \cdot (\boldsymbol{\Sigma}_t(\mathbf{x}_t; \theta)\boldsymbol{\Sigma}_t^\top(\mathbf{x}_t; \theta))] \right|^2,$$

if for the set of functions $\mathcal{H} = \{h(\boldsymbol{x})\pi_t(\boldsymbol{x})\boldsymbol{v}_t(\boldsymbol{x}),\ \pi_t(\boldsymbol{x})\boldsymbol{D}(\boldsymbol{x})\nabla h(\boldsymbol{x}),\ \pi_t(\boldsymbol{x})\nabla^\top\boldsymbol{D}_t(\boldsymbol{x})h(\boldsymbol{x}),$ $h(\boldsymbol{x})\boldsymbol{D}_t(\boldsymbol{x})\nabla\pi_t(\boldsymbol{x})\}$ ($\theta$ is omitted for simplicity) that $h(\boldsymbol{x}) = \prod_{(j,m)\in\widetilde{\mathcal{M}}}(\mathrm{x}_{t,[j]})^m,\ \boldsymbol{D}_t(\boldsymbol{x}) = \boldsymbol{\Sigma}_t(\boldsymbol{x})\boldsymbol{\Sigma}_t^\top(\boldsymbol{x})$, it satisfies: (i) Continuity: $h' \in \mathcal{H}$ is continuously differentiable w.r.t. $\boldsymbol{x}$; (ii) Light tail: The probability density function $\pi_t(\boldsymbol{x})$ is characterized by tails that are sufficiently light, such that $\oint_{S_\infty} h'(\boldsymbol{x}) \cdot \mathrm{d}\boldsymbol{a} = 0$ for $h' \in \mathcal{H}$, where $\boldsymbol{a}$ is the outward pointing unit normal on the $S_\infty$ boundary.

**Table 4:** Notation settings for $k \geq 2$.

| Notations | Descriptions |
|---|---|
| $\mathcal{S}_{<k>}$ | Set of Operators Defining $L_{\text{corr}}(\pi_t, \widetilde{\mathcal{M}}, k)$ |
| $\widetilde{\mathcal{S}} \in \mathcal{S}_{<k>}$ | Sequence of Operators of the form $\widetilde{\mathcal{S}}' \cup \{\Upsilon'_{<i',j'>}\}$ in $\mathcal{S}_{<k>}$ |
| $\widetilde{\mathcal{S}}'$ | Sequence of Operators up to the Penultimate One in $\widetilde{\mathcal{S}}$ |
| $\Upsilon'_{<i',j'>}$ | The Last Operator in $\widetilde{\mathcal{S}}$ |

We here use another notation to re-express $L_{\text{corr}}(\pi_t, \widetilde{\mathcal{M}}, 1)$, in order to facilitate the following iterative derivation from $k-1$ to $k$ for $L_{\text{corr}}(\pi_t, \widetilde{\mathcal{M}}, k)$. Specifically, we re-express $L_{\text{corr}}(\pi_t, \widetilde{\mathcal{M}}, 1)$ in the form of:

$$L_{\text{corr}}(\pi_t, \widetilde{\mathcal{M}}, 1) = \left| \sum_{\widetilde{\mathcal{S}}\in\left\{ \{\nabla_{<0,0>}\}, \{(\nabla\nabla^\top)_{<0,0>}\} \right\}} \Gamma(\widetilde{\mathcal{S}}) \right|^2$$

$$= \left| \Gamma(\{\nabla_{<0,0>}\}) + \Gamma(\{(\nabla\nabla^\top)_{<0,0>}\}) \right|^2$$

such that we show that for all $L_{\text{corr}}(\pi_t, \widetilde{\mathcal{M}}, k)$ it can be expressed in the form of $|\sum_{\widetilde{\mathcal{S}}} \Gamma(\widetilde{\mathcal{S}})|^2$. By denoting $\mathcal{S}_{<1>} = \{\{\nabla_{<0,0>}\}, \{(\nabla\nabla^\top)_{<0,0>}\}\}$ the collection of these $\widetilde{\mathcal{S}}$ for $k=1$, we have that for $k \geq 2$, it admits the analytical expression in an iterative manner as:

$$L_{\text{corr}}(\pi_t, \widetilde{\mathcal{M}}, k) = \left| \sum_{\substack{\widetilde{\mathcal{S}}\in\mathcal{S}_{<k-1>} \text{ that} \\ \widetilde{\mathcal{S}}=\widetilde{\mathcal{S}}'\cup\{\Upsilon'_{<i',j'>}\}}} \Gamma(\widetilde{\mathcal{S}}' \cup \Upsilon'_{<i'+1,j'>}) + \Gamma(\widetilde{\mathcal{S}}' \cup \Upsilon'_{<i',j'+1>}) \right.$$

$$\left. + \Gamma(\widetilde{\mathcal{S}} \cup \nabla_{<0,0>}) + \Gamma(\widetilde{\mathcal{S}} \cup (\nabla\nabla^\top)_{<0,0>}) \right|^2, \quad (14)$$

where we denote the ordered sequence of operators $\widetilde{\mathcal{S}} = \{..., \Upsilon_{<i,j>}, ...\}$ that $\Upsilon \in \{\nabla, \nabla\nabla^\top\}, i \in \mathbb{Z}_>, j \in \mathbb{Z}_>$ such that:

$$\Upsilon_{<i,j>}(\boldsymbol{x}) = \frac{\mathrm{d}^i}{\mathrm{d}t^i}\Upsilon( \prod_{(k,m)\in\widetilde{\mathcal{M}}} (x_{[k]})^m) \cdot \frac{\mathrm{d}^j}{\mathrm{d}t^j}\gamma(\boldsymbol{x}), \qquad \gamma(\boldsymbol{x}) = \left\{ \begin{matrix} \boldsymbol{v}_t(\boldsymbol{x}), \text{ if } \Upsilon=\nabla \\ \boldsymbol{\Sigma}_t(\boldsymbol{x})\boldsymbol{\Sigma}_t^\top(\boldsymbol{x}), \text{ else if } \Upsilon=\nabla\nabla^\top \end{matrix} \right.,$$

and denote the function $\Gamma(\cdot)$ operating on the ordered sequence $\widetilde{\mathcal{S}}$ and $\circ$ as function composition such that:

$$\Gamma(\widetilde{\mathcal{S}}) = c_{\widetilde{\mathcal{S}}}\mathbb{E}_{\pi_t}[\circ_{\Upsilon_{<i,j>}\in\widetilde{\mathcal{S}}}\Upsilon_{<i,j>}(\mathbf{x}_t)], \qquad c_{\widetilde{\mathcal{S}}} = 2^{-|\{\Upsilon_{<i,j>}:\Upsilon_{<i,j>}\in\widetilde{\mathcal{S}},\Upsilon=\nabla\nabla^{\top}\}|},$$

and further denote $\mathcal{S}_{<k>}$ as the set of $\widetilde{\mathcal{S}}$ used to compute $L_{\text{corr}}(\pi_t, \widetilde{\mathcal{M}}, k)$, e.g., $\mathcal{S}_{<1>} = \{\{\nabla_{<0,0>}\}, \{(\nabla\nabla^{\top})_{<0,0>}\}\}$. With the given notations, it is noticed that $L_{\text{corr}}(\pi_t, \widetilde{\mathcal{M}}, 1)$ can be rewritten in the form of Eq. (14) as $L_{\text{corr}}(\pi_t, \widetilde{\mathcal{M}}, 1) = |\Gamma(\{\nabla_{<0,0>}\}) + \Gamma(\{(\nabla\nabla^{\top})_{<0,0>}\})|^2$.

The conditions for if for the equality in Eq. (14) are that, for the set of functions $\mathcal{H} = \{h(\boldsymbol{x})\pi_t(\boldsymbol{x})\boldsymbol{v}_t(\boldsymbol{x}),\ \pi_t(\boldsymbol{x})\boldsymbol{D}(\boldsymbol{x})\nabla h(\boldsymbol{x}),\ \pi_t(\boldsymbol{x})\nabla^{\top}\boldsymbol{D}_t(\boldsymbol{x})h(\boldsymbol{x}),\ h(\boldsymbol{x})\boldsymbol{D}_t(\boldsymbol{x})\nabla\pi_t(\boldsymbol{x})\}$ that $h(\boldsymbol{x}) = \circ_{\Upsilon_{<i,j>}\in\widetilde{\mathcal{S}}}\Upsilon_{<i,j>}(\boldsymbol{x}), \forall\widetilde{\mathcal{S}} \in \mathcal{S}_{<k-1>}$, it satisfies: (i) Continuity. $h' \in \mathcal{H}$ is continuously differentiable w.r.t. $\boldsymbol{x}$; (ii) Light tail. The probability density function $\pi_t(\boldsymbol{x})$ is characterized by tails that are sufficiently light, such that $\oint_{S_\infty} h'(\boldsymbol{x})\cdot\mathrm{d}\boldsymbol{a} = 0$ for $h' \in \mathcal{H}$.

*Proof.* **Expression for $k = 1$.** For simplicity, we omit $\theta$ in notations that $\boldsymbol{v}_t(\boldsymbol{x}; \theta), \boldsymbol{\Sigma}_t(\boldsymbol{x}; \theta)$ are referred as $\boldsymbol{v}_t(\boldsymbol{x}), \boldsymbol{\Sigma}_t(\boldsymbol{x})$, respectively. Denoting $h : \mathbb{R}^d \to \mathbb{R}$ as the mapping satisfying the continuity and light tail conditions, for $k = 1$, we have:

$$\frac{\mathrm{d}}{\mathrm{d}t}\mathbb{E}_{\pi_t}[h(\mathbf{x}_t)]$$

$$\stackrel{(a)}{=} \frac{\mathrm{d}}{\mathrm{d}t}\int h(\boldsymbol{x})\pi_t(\boldsymbol{x})\mathrm{d}\boldsymbol{x} = \int h(\boldsymbol{x})(\frac{\mathrm{d}}{\mathrm{d}t}\pi_t(\boldsymbol{x}))\mathrm{d}\boldsymbol{x}$$

$$\stackrel{(b)}{=} \int h(\boldsymbol{x})\Big(-\nabla\cdot(\pi_t(\boldsymbol{x})\boldsymbol{v}_t(\boldsymbol{x})) + \frac{1}{2}\nabla\cdot\nabla\cdot(\pi_t(\boldsymbol{x})\boldsymbol{\Sigma}_t(\boldsymbol{x})\boldsymbol{\Sigma}_t^{\top}(\boldsymbol{x}))\Big)\mathrm{d}\boldsymbol{x}$$

$$= \int\Big(-h(\boldsymbol{x})\Big)\Big(\nabla\cdot(\pi_t(\boldsymbol{x})\boldsymbol{v}_t(\boldsymbol{x}))\Big)\mathrm{d}\boldsymbol{x} + \int\Big(\frac{1}{2}h(\boldsymbol{x})\Big)\Big(\nabla\cdot\nabla\cdot(\pi_t(\boldsymbol{x})\boldsymbol{\Sigma}_t(\boldsymbol{x})\boldsymbol{\Sigma}_t^{\top}(\boldsymbol{x}))\Big)\mathrm{d}\boldsymbol{x}$$

$$\stackrel{(c)}{=} \overbrace{\int\nabla\Big(h(\boldsymbol{x})\Big)\cdot\Big(\pi_t(\boldsymbol{x})\boldsymbol{v}_t(\boldsymbol{x})\Big)\mathrm{d}\boldsymbol{x}}^{\text{Part (i)}} + \overbrace{\int-\nabla\cdot\Big(h(\boldsymbol{x})\pi_t(\boldsymbol{x})\boldsymbol{v}_t(\boldsymbol{x})\Big)\mathrm{d}\boldsymbol{x}}^{\text{Part (ii)}}$$

$$+ \overbrace{\int-\nabla\Big(\frac{1}{2}h(\boldsymbol{x})\Big)\cdot\Big(\nabla\cdot(\pi_t(\boldsymbol{x})\boldsymbol{\Sigma}_t(\boldsymbol{x})\boldsymbol{\Sigma}_t^{\top}(\boldsymbol{x}))\Big)\mathrm{d}\boldsymbol{x}}^{\text{Part (iii)}}$$

$$+ \overbrace{\int\nabla\cdot\Big(\Big(\frac{1}{2}h(\boldsymbol{x})\Big)\Big(\nabla\cdot(\pi_t(\boldsymbol{x})\boldsymbol{\Sigma}_t(\boldsymbol{x})\boldsymbol{\Sigma}_t^{\top}(\boldsymbol{x}))\Big)\Big)\mathrm{d}\boldsymbol{x}}^{\text{Part (iv)}},$$

where (a) is attained through the standard definition of integration, (b) results from the application of the Fokker-Planck equation (2) to substitute the time derivative term with the divergence term, and (c) is achieved by applying integration by parts on the right-hand side (RHS) of Eq. (b). Next, we solve the four parts on RHS of Eq. (c). For part (i), we have:

$$\int\nabla\Big(h(\boldsymbol{x})\Big)\cdot\Big(\pi_t(\boldsymbol{x})\boldsymbol{v}_t(\boldsymbol{x})\Big)\mathrm{d}\boldsymbol{x}$$

$$= \int\Big(\nabla h(\boldsymbol{x})\cdot\boldsymbol{v}_t(\boldsymbol{x})\Big)\pi_t(\boldsymbol{x})\mathrm{d}\boldsymbol{x}$$

$$= \mathbb{E}_{\pi_t}[\nabla h(\boldsymbol{x})\cdot\boldsymbol{v}_t(\boldsymbol{x})].$$

For part (ii), we have:

$$\int-\nabla\cdot\Big(h(\boldsymbol{x})\pi_t(\boldsymbol{x})\boldsymbol{v}_t(\boldsymbol{x})\Big)\mathrm{d}\boldsymbol{x}$$

$$\stackrel{(a)}{=} -\oint_{S_\infty}\Big(h(\boldsymbol{x})\pi_t(\boldsymbol{x})\boldsymbol{v}_t(\boldsymbol{x})\Big)\cdot\mathrm{d}\boldsymbol{a}$$

$$\stackrel{(b)}{=} 0,$$

where (a) is accomplished through the application of Gauss's theorem (Temple, 1936), that $\boldsymbol{a}$ is the outward pointing unit normal at each point on the boundary at infinity $S_\infty$, under the satisfaction of the continuity condition, and (b) is achieved by considering the light tail condition.

For part (iii), we have:

$$\int \nabla\Big(h(\boldsymbol{x})\Big)\cdot\Big(\nabla\cdot(\pi_t(\boldsymbol{x})\boldsymbol{\Sigma}_t(\boldsymbol{x})\boldsymbol{\Sigma}_t^\top(\boldsymbol{x}))\Big)\mathrm{d}\boldsymbol{x}$$

$$\stackrel{(a)}{=} \int \nabla\cdot\Big(\pi_t(\boldsymbol{x})\boldsymbol{\Sigma}_t(\boldsymbol{x})\boldsymbol{\Sigma}_t^\top(\boldsymbol{x})\nabla h(\boldsymbol{x})\Big)\mathrm{d}\boldsymbol{x} - \int \Big(\nabla\nabla^\top h(\boldsymbol{x})\Big)\cdot\Big(\pi_t(\boldsymbol{x})\boldsymbol{\Sigma}_t(\boldsymbol{x})\boldsymbol{\Sigma}_t^\top(\boldsymbol{x})\Big)\mathrm{d}\boldsymbol{x}$$

$$\stackrel{(b)}{=} \oint_{S_\infty} \Big(\pi_t(\boldsymbol{x})\boldsymbol{\Sigma}_t(\boldsymbol{x})\boldsymbol{\Sigma}_t^\top(\boldsymbol{x})\nabla h(\boldsymbol{x})\Big)\cdot\mathrm{d}\boldsymbol{a} - \mathbb{E}_{\pi_t}\big[\big(\nabla\nabla^\top h(\boldsymbol{x})\big)\cdot\big(\boldsymbol{\Sigma}_t(\boldsymbol{x})\boldsymbol{\Sigma}_t^\top(\boldsymbol{x})\big)\big]$$

$$\stackrel{(c)}{=} -\mathbb{E}_{\pi_t}\big[\big(\nabla\nabla^\top h(\boldsymbol{x})\big)\cdot\big(\boldsymbol{\Sigma}_t(\boldsymbol{x})\boldsymbol{\Sigma}_t^\top(\boldsymbol{x})\big)\big],$$

where (a) is achieved by applying integration by parts, (b) is accomplished through the application of Gauss's theorem under the satisfaction of the continuity condition, and (c) is achieved by considering the light tail condition.

For part (iv), we have:

$$\int \nabla\cdot\Big(\big(h(\boldsymbol{x})\big)\big(\nabla\cdot(\pi_t(\boldsymbol{x})\boldsymbol{\Sigma}_t(\boldsymbol{x})\boldsymbol{\Sigma}_t^\top(\boldsymbol{x}))\big)\Big)\mathrm{d}\boldsymbol{x}$$

$$\stackrel{(a)}{=} \int \nabla\cdot\Big(\big(h(\boldsymbol{x})\big)\big(\nabla\cdot(\boldsymbol{\Sigma}_t(\boldsymbol{x})\boldsymbol{\Sigma}_t^\top(\boldsymbol{x}))\big)\pi_t(\boldsymbol{x})\Big)\mathrm{d}\boldsymbol{x} + \int \nabla\cdot\Big(\big(h(\boldsymbol{x})\big)\big(\nabla\pi_t(\boldsymbol{x})\cdot(\boldsymbol{\Sigma}_t(\boldsymbol{x})\boldsymbol{\Sigma}_t^\top(\boldsymbol{x}))\big)\Big)\mathrm{d}\boldsymbol{x}$$

$$\stackrel{(b)}{=} \oint_{S_\infty} \Big(\big(h(\boldsymbol{x})\big)\big(\nabla\cdot(\boldsymbol{\Sigma}_t(\boldsymbol{x})\boldsymbol{\Sigma}_t^\top(\boldsymbol{x}))\big)\pi_t(\boldsymbol{x})\Big)\cdot\mathrm{d}\boldsymbol{a} + \oint_{S_\infty} \Big(\big(h(\boldsymbol{x})\big)\big(\nabla\pi_t(\boldsymbol{x})\cdot(\boldsymbol{\Sigma}_t(\boldsymbol{x})\boldsymbol{\Sigma}_t^\top(\boldsymbol{x}))\big)\Big)\cdot\mathrm{d}\boldsymbol{a}$$

$$\stackrel{(c)}{=} 0,$$

where (a) is achieved by applying the product rule, (b) is accomplished through the application of Gauss's theorem under the satisfaction of the continuity condition, and (c) is achieved by considering the light tail condition.

By combining them and setting $h(\boldsymbol{x}) = \prod_{(j,m)\in\widetilde{\mathcal{M}}}(\mathrm{x}_{[j]})^m$, we eventually have:

$$L_{\mathrm{corr}}(\pi_t, \widetilde{\mathcal{M}}, 1) = \Big|\mathbb{E}_{\pi_t}[\nabla\Big(\prod_{(j,m)\in\widetilde{\mathcal{M}}}(\mathrm{x}_{t,[j]})^m\Big)\cdot\boldsymbol{v}_t(\mathbf{x}_t)]$$

$$+ \frac{1}{2}\mathbb{E}_{\pi_t}[(\nabla\nabla^\top\Big(\prod_{(j,m)\in\widetilde{\mathcal{M}}}(\mathrm{x}_{t,[j]})^m\Big))\cdot(\boldsymbol{\Sigma}_t(\mathbf{x}_t)\boldsymbol{\Sigma}_t^\top(\mathbf{x}_t))]\Big|^2.$$

**Expression for $k \geq 2$.** We present a more general form of correlational Lagrangian, by denoting the ordered sequence of operators $\widetilde{\mathcal{S}} = \{..., \Upsilon_{<i,j>}, ...\}$ that $\Upsilon \in \{\nabla, \nabla\nabla^\top\}, i \in \mathbb{Z}_>, j \in \mathbb{Z}_>$ such that

$$\Upsilon_{<i,j>}(\boldsymbol{x}) = \frac{\mathrm{d}^i}{\mathrm{d}t^i}\Upsilon\Big(\prod_{(k,m)\in\widetilde{\mathcal{M}}}(x_{[k]})^m\Big)\cdot\frac{\mathrm{d}^j}{\mathrm{d}t^j}\gamma(\boldsymbol{x}), \qquad \gamma(\boldsymbol{x}) = \begin{cases} \boldsymbol{v}_t(\boldsymbol{x}), \text{ if } \Upsilon=\nabla \\ \boldsymbol{\Sigma}_t(\boldsymbol{x})\boldsymbol{\Sigma}_t^\top(\boldsymbol{x}), \text{ else if } \Upsilon=\nabla\nabla^\top \end{cases},$$

and denote the function $\Gamma(\cdot)$ operating on the ordered sequence $\widetilde{\mathcal{S}}$ such that

$$\Gamma(\widetilde{\mathcal{S}}) = c_{\widetilde{\mathcal{S}}}\mathbb{E}_{\pi_t}[\circ_{\Upsilon_{<i,j>}\in\widetilde{\mathcal{S}}}\Upsilon_{<i,j>}(\mathbf{x}_t)], \qquad c_{\widetilde{\mathcal{S}}} = 2^{-|\{\Upsilon_{<i,j>}:\Upsilon_{<i,j>}\in\widetilde{\mathcal{S}},\Upsilon=\nabla\nabla^\top\}|},$$

and then we can rewrite correlational Lagrangian for the $k = 1$ case:

$$L_{\mathrm{corr}}(\pi_t, \widetilde{\mathcal{M}}, 1) = \Big|\Gamma(\{\nabla_{<0,0>}\}) + \Gamma(\{(\nabla\nabla^\top)_{<0,0>}\})\Big|^2.$$

We further denote $\mathcal{S}_{<k>}$ as the set of $\widetilde{\mathcal{S}}$ used to compute $L_{\mathrm{corr}}(\pi_t, \widetilde{\mathcal{M}}, k)$, e.g., $\mathcal{S}_{<1>} = \{\{\nabla_{<0,0>}\}, \{(\nabla\nabla^\top)_{<0,0>}\}\}$ according to the above formulation. Thus, for $k \geq 2$, we have:

$$L_{\mathrm{corr}}(\pi_t, \widetilde{\mathcal{M}}, k) = \Big|\sum_{\widetilde{\mathcal{S}}\in\mathcal{S}_{<k-1>}}\frac{\mathrm{d}}{\mathrm{d}t}\Gamma(\widetilde{\mathcal{S}})\Big|^2.$$

To calculate this general formulation, denoting $\widetilde{\mathcal{S}} = \widetilde{\mathcal{S}}' \cup \{\Upsilon'_{<i',j'>}\}$, we utilize the following equation:

$$\frac{\mathrm{d}}{\mathrm{d}t}\Gamma(\widetilde{\mathcal{S}}) = \frac{\mathrm{d}}{\mathrm{d}t}\Gamma(\widetilde{\mathcal{S}}' \cup \{\Upsilon'_{<i',j'>}\})$$

$$\overset{(a)}{=} c_{\widetilde{\mathcal{S}}}\frac{\mathrm{d}}{\mathrm{d}t}\mathbb{E}_{\pi_t}[\circ_{\Upsilon_{<i,j>}\in\widetilde{\mathcal{S}}}\Upsilon_{<i,j>}(\mathbf{x}_t)]$$

$$\overset{(b)}{=} c_{\widetilde{\mathcal{S}}}\frac{\mathrm{d}}{\mathrm{d}t}\int \frac{\mathrm{d}^{i'}}{\mathrm{d}t^{i'}}\Upsilon'\Big(\circ_{\Upsilon_{<i,j>}\in\widetilde{\mathcal{S}}'}\Upsilon_{<i,j>}(\boldsymbol{x})\Big)\cdot\Big(\frac{\mathrm{d}^{j'}}{\mathrm{d}t^{j'}}\gamma'(\boldsymbol{x})\Big)\pi_t(\boldsymbol{x})\mathrm{d}\boldsymbol{x}$$

$$\overset{(c)}{=} c_{\widetilde{\mathcal{S}}}\int \frac{\mathrm{d}^{i'+1}}{\mathrm{d}t^{i'+1}}\Upsilon'\Big(\circ_{\Upsilon_{<i,j>}\in\widetilde{\mathcal{S}}'}\Upsilon_{<i,j>}(\boldsymbol{x})\Big)\cdot\Big(\frac{\mathrm{d}^{j'}}{\mathrm{d}t^{j'}}\gamma'(\boldsymbol{x})\Big)\pi_t(\boldsymbol{x})\mathrm{d}\boldsymbol{x}$$

$$+ c_{\widetilde{\mathcal{S}}}\int \frac{\mathrm{d}^{i'}}{\mathrm{d}t^{i'}}\Upsilon'\Big(\circ_{\Upsilon_{<i,j>}\in\widetilde{\mathcal{S}}'}\Upsilon_{<i,j>}(\boldsymbol{x})\Big)\cdot\Big(\frac{\mathrm{d}^{j'+1}}{\mathrm{d}t^{j'+1}}\gamma'(\boldsymbol{x})\Big)\pi_t(\boldsymbol{x})\mathrm{d}\boldsymbol{x}$$

$$+ c_{\widetilde{\mathcal{S}}}\int \frac{\mathrm{d}^{i'}}{\mathrm{d}t^{i'}}\Upsilon'\Big(\circ_{\Upsilon_{<i,j>}\in\widetilde{\mathcal{S}}'}\Upsilon_{<i,j>}(\boldsymbol{x})\Big)\cdot\Big(\frac{\mathrm{d}^{j'}}{\mathrm{d}t^{j'}}\gamma'(\boldsymbol{x})\Big)\Big(\frac{\mathrm{d}}{\mathrm{d}t}\pi_t(\boldsymbol{x})\Big)\mathrm{d}\boldsymbol{x}$$

$$\overset{(d)}{=} \Gamma(\widetilde{\mathcal{S}}' \cup \{\Upsilon'_{<i'+1,j'>}\}) + \Gamma(\widetilde{\mathcal{S}}' \cup \{\Upsilon'_{<i',j'+1>}\}) + c_{\widetilde{\mathcal{S}}}\int \Big(\circ_{\Upsilon_{<i,j>}\in\widetilde{\mathcal{S}}}\Upsilon_{<i,j>}(\boldsymbol{x})\Big)\Big(\frac{\mathrm{d}}{\mathrm{d}t}\pi_t(\boldsymbol{x})\Big)\mathrm{d}\boldsymbol{x},$$

where (a, b) is established through definitions, (c) is realized by applying the product rule, and (d) is also derived from standard definitions. Denoting $h(\boldsymbol{x}) = \circ_{\Upsilon_{<i,j>}\in\widetilde{\mathcal{S}}}\Upsilon_{<i,j>}(\boldsymbol{x})$, we have:

$$c_{\widetilde{\mathcal{S}}}\int h(\boldsymbol{x})(\frac{\mathrm{d}}{\mathrm{d}t}\pi_t(\boldsymbol{x}))\mathrm{d}\boldsymbol{x}$$

$$\overset{(a)}{=} c_{\widetilde{\mathcal{S}}}\mathbb{E}_{\pi_t}[\nabla h(\boldsymbol{x})\cdot\boldsymbol{v}_t(\mathbf{x}_t)] + \frac{c_{\widetilde{\mathcal{S}}}}{2}\mathbb{E}_{\pi_t}[(\nabla\nabla^\top(h(\boldsymbol{x})))\cdot(\boldsymbol{\Sigma}_t(\mathbf{x}_t)\boldsymbol{\Sigma}_t^\top(\mathbf{x}_t))]$$

$$\overset{(b)}{=} \Gamma(\widetilde{\mathcal{S}} \cup \{\nabla_{<0,0>}\}) + \Gamma(\widetilde{\mathcal{S}} \cup \{(\nabla\nabla^\top)_{<0,0>}\}),$$

where (a) follows the same derivation of correlational Lagrangian for $k = 1$, under the satisfaction of the continuity condition and light tail conditions, and (b) is established through definitions.

By combining them, we eventually have:

$$L_{\mathrm{corr}}(\pi_t, \widetilde{\mathcal{M}}, k) = \Big|\sum_{\substack{\widetilde{\mathcal{S}}\in\mathcal{S}_{<k-1>},\\ \widetilde{\mathcal{S}}=\widetilde{\mathcal{S}}'\cup\{\Upsilon'_{<i',j'>}\}}} \Gamma(\widetilde{\mathcal{S}}' \cup \Upsilon'_{<i'+1,j'>}) + \Gamma(\widetilde{\mathcal{S}}' \cup \Upsilon'_{<i',j'+1>})$$

$$+ \Gamma(\widetilde{\mathcal{S}} \cup \nabla_{<0,0>}) + \Gamma(\widetilde{\mathcal{S}} \cup (\nabla\nabla^\top)_{<0,0>})\Big|^2.$$

# B DERIVATION OF COVARIANCE ACCELERATION

The derivation of covariance acceleration (Eq. (9)) is carried out by applying Propos. 1 (Eq. (14)) as follows:

$$\sum_{\widetilde{\mathcal{M}}\in\mathcal{M}_{\mathrm{cov}}} L_{\mathrm{corr}}(\pi_t, \widetilde{\mathcal{M}}, 2) = \Big\|\frac{\mathrm{d}^2}{\mathrm{d}t^2}\mathbb{E}_{\pi_t}[\mathbf{x}_t\mathbf{x}_t^\top]\Big\|_{\mathsf{F}}^2$$

$$= \Big\|\overbrace{\mathbb{E}_{\pi_t}\Big[\mathbf{x}_t(\frac{\mathrm{d}}{\mathrm{d}t}\boldsymbol{v}_t(\mathbf{x}_t))^\top + (\frac{\mathrm{d}}{\mathrm{d}t}\boldsymbol{v}_t(\mathbf{x}_t))\mathbf{x}_t^\top + \frac{1}{2}\frac{\mathrm{d}}{\mathrm{d}t}(\boldsymbol{\Sigma}_t(\mathbf{x}_t)\boldsymbol{\Sigma}_t^\top(\mathbf{x}_t))\Big]}^{\text{Matrix collection of }\Gamma(\widetilde{\mathcal{S}}'\cup\Upsilon'_{<i',j'+1>})\text{ terms in Eq. (14)}} + \overbrace{\mathbf{0}}^{\Gamma(\widetilde{\mathcal{S}}'\cup\Upsilon'_{<i'+1,j'>})}$$

$$+ \overbrace{\mathbb{E}_{\pi_t}\Big[\mathbf{x}_t(\nabla\boldsymbol{v}_t(\mathbf{x}_t)\boldsymbol{v}_t(\mathbf{x}_t))^\top + (\nabla\boldsymbol{v}_t(\mathbf{x}_t)\boldsymbol{v}_t(\mathbf{x}_t))\mathbf{x}_t^\top}^{\Gamma(\widetilde{\mathcal{S}}\cup\nabla_{<0,0>})}$$

$$+ 2\boldsymbol{v}_t(\mathbf{x}_t)\boldsymbol{v}_t(\mathbf{x}_t)^\top + \frac{1}{2}\nabla(\boldsymbol{\Sigma}_t(\mathbf{x}_t)\boldsymbol{\Sigma}_t^\top(\mathbf{x}_t))_{\underline{i_1i_2i_3}}\boldsymbol{v}_t^{i_3}(\mathbf{x}_t)\Big]$$

$$\begin{aligned}
&\overbrace{+\ \mathbb{E}_{\pi_t}\Big[\nabla\boldsymbol{v}_t(\mathbf{x}_t)\boldsymbol{\Sigma}_t(\mathbf{x}_t)\boldsymbol{\Sigma}_t^\top(\mathbf{x}_t)+\boldsymbol{\Sigma}_t(\mathbf{x}_t)\boldsymbol{\Sigma}_t^\top(\mathbf{x}_t)\nabla^\top\boldsymbol{v}_t(\mathbf{x}_t)}^{\Gamma(\widetilde{\mathcal{S}}\cup(\nabla\nabla^\top)_{<0,0>})} \\
&+\frac{1}{2}\mathbf{x}_t(\nabla\nabla^\top(\boldsymbol{v}_t(\mathbf{x}_t))_{\underline{i_1i_2i_3}}(\boldsymbol{\Sigma}_t(\mathbf{x}_t)\boldsymbol{\Sigma}_t^\top(\mathbf{x}_t))^{\underline{i_2i_3}})^\top+\frac{1}{2}(\nabla\nabla^\top(\boldsymbol{v}_t(\mathbf{x}_t))_{\underline{i_1i_2i_3}}(\boldsymbol{\Sigma}_t(\mathbf{x}_t)\boldsymbol{\Sigma}_t^\top(\mathbf{x}_t))^{\underline{i_2i_3}})\mathbf{x}_t^\top \\
&+\frac{1}{4}\nabla\nabla^\top(\boldsymbol{\Sigma}_t(\mathbf{x}_t)\boldsymbol{\Sigma}_t^\top(\mathbf{x}_t))_{\underline{i_1i_2i_3i_4}}(\boldsymbol{\Sigma}_t(\mathbf{x}_t)\boldsymbol{\Sigma}_t^\top(\mathbf{x}_t))^{\underline{i_3i_4}}\Big]\Big\|_{\mathsf{F}}^2.
\end{aligned}$$

## C   MORE RELATED WORKS

**Modeling population dynamics with machine learning.** A significant body of research has been dedicated to modeling population dynamics using data-driven approaches. This includes the development of continuous normalizing flows (Chen et al., 2018; Mathieu & Nickel, 2020), which model the dynamics through ordinary differential equations (ODEs). Furthermore, an advancement of neural ODEs, namely neural SDEs, has been introduced to capture both drift and diffusion processes using neural networks (Li et al., 2020; Tzen & Raginsky, 2019). In scenarios where ground truth trajectories are inaccessible, regularization strategies for flows have been developed. These strategies emphasize enforcing constraints on the motion of individual trajectories. Examples include the regularization of kinetic energy and its Jacobian (Tong et al., 2020; Finlay et al., 2020), as well as the inclusion of dual terms derived from the Hamilton–Jacobi–Bellman equation (Koshizuka & Sato, 2022; Onken et al., 2021), aiming to guide the model towards realistic dynamic behaviors.

In a very general sense, these methods are categorized under the optimal transport framework, characterized by varying choices of cost objectives (Somnath et al., 2023; De Bortoli et al., 2021; Bunne et al., 2023; 2022; Schiebinger et al., 2019; Neklyudov et al., 2023; Liu et al., 2022; Pariset et al., 2023; Tamir et al., 2023; You et al., 2023). It is crucial within this framework to thoughtfully construct cost functions, as they impose various priors on the dynamics data. This often leads to the imposition of homogeneous priors across all individual particles, affecting both learning accuracy and efficiency. In contrast, our work aims to model heterogeneous particle behaviors, as observed in various real-world population dynamics. For example, cell-to-cell variations in gene expression are inherent to biological systems, with changes in such variations linked to disease phenotypes and aging. Consequently, our approach enhances accuracy by employing appropriate and justifiable population-level priors to learn the dynamics of heterogeneous particles.

**Developmental modeling of embryonic stem cells.** The modeling of embryonic stem cell development represents a cutting-edge intersection of developmental biology, computational science, and systems biology (Alison et al., 2002; Zakrzewski et al., 2019; Weinreb et al., 2020). This field aims to unravel the complex processes governing the differentiation and proliferation of embryonic stem cells into the diverse cell types that form an organism. Given the foundational role of these processes in understanding both normal development and various diseases, developmental modeling of embryonic stem cells has garnered significant interest. At its core, developmental modeling seeks to simulate and predict the dynamic behavior of stem cells as they progress through various stages of development. This involves mapping the intricate pathways that lead to cell fate decisions, a challenge that requires sophisticated computational models and deep biological insights.

**Dose-dependent cellular response prediction to chemical perturbations.** The prediction of dose-dependent cellular responses to chemical perturbations is pivotal in pharmacology, toxicology, and systems biology (Dong et al., 2023; Bunne et al., 2023; Roohani et al., 2022; Lotfollahi et al., 2023). It aims to understand how cells react to varying concentrations of chemical compounds, which is crucial for drug development, safety assessment, and personalized medicine. This field combines quantitative biology, computational modeling, and high-throughput experimental techniques to map out the intricate cellular mechanisms activated or inhibited by drugs and other chemical agents at different doses. At the heart of dose-dependent cellular response prediction is the need to accurately model the complex, nonlinear interactions between chemical perturbations and cellular pathways. This involves determining the specific dose at which a chemical agent begins to have a biological effect (the threshold), the range over which the response changes (the dynamic range), and the dose causing maximal response (the ceiling).

**Connection with Probability Flow Ordinary Differential Equation.** Our model is able to integrate the Probability Flow Ordinary Differential Equation (Song et al., 2020) to accelarate

the sampling in scenarios where the score function can be expressed. For our parametrized SDE $dx_t = v_t(x_t)dt + \Sigma_t(x_t)d\omega_t$, the corresponding probability flow ODE sharing the same marginal probability densities is formulated as $dx_t = v_t(x_t) - \frac{1}{2}\nabla \cdot \left[\Sigma_t(x_t)\Sigma_t(x_t)^\top\right] - \frac{1}{2}\Sigma_t(x_t)\Sigma_t(x_t)^\top\nabla_x \log p_t(x_t)dt$ which requires the expression of $\nabla_x \log p_t(x_t)$ (the score function). Since $\nabla_x \log p_t(x_t)$ is in general not directly derivable, (Song et al., 2020) constructs the (known) artificial dynamics between data and white noise in a certain way such that $\nabla_x \log p_t(x_t)$ can be approximated with a neural-network parametrized score model.

Thus, in scenarios where the score function can be explicitly expressed, we are able to construct PF-ODE. An example is described as follows: $p_0$ is the mixture of Gaussian that $p_0(x) = \sum_{i=1}^n w_i \mathcal{N}(x; \mu_{0,i}, \sigma_{0,i}^2)$; The SDE is linear such that $v_t(x_t) = ax_t + b$, $\Sigma_t(x_t) = c$; Denoting $\mu_{t,i} = \exp(at)\mu_{t,i} + \frac{b}{a}(\exp(at) - 1)$, $\sigma_{t,i}^2 = \sigma_{0,i}^2 \exp(2at) + \frac{c^2}{2a}(\exp(2at) - 1)$; The score function can then be expressed as $\nabla_x p_t(x_t) = \frac{\sum_{i=1}^n w_i \frac{-(x_t - \mu_{t,i})}{\sigma_{t,i}^2}\mathcal{N}(x_t; \mu_{t,i}, \sigma_{t,i}^2)}{\sum_{i=1}^n w_i \mathcal{N}(x_t; \mu_{t,i}, \sigma_{t,i}^2)}$. In the scenarios where the PF-ODE is constructed, we can speed up the sampling process via: By denoting $h_t(x_t) = v_t(x_t) - \frac{1}{2}\nabla\left[\Sigma_t(x_t)\Sigma_t(x_t)^\top\right] - \frac{1}{2}\Sigma_t(x_t)\Sigma_t(x_t)^\top\nabla_x \log p_t(x_t)$, we can then compute the exact log-likelihood via $\log p_t(x_t) = \log p_0(x_0) + \int_0^t \nabla \cdot h_s(x_s)ds$.

**Connection with Flow Matching.** Our model is potentially capable of integrating the flow matching objective (FM) (Lipman et al., 2022), since FM is an orthogonal objective to our proposed regularization, focusing on capturing the mismatch between generated and observed data. More specifically, FM is an alternative to the (Wasserstein) data matching loss in our framework formulated in Opt. (13). The integration can be conducted by further adding the FM loss into our optimization objective. The advantages of the FM loss are well-known: it is simple, effective in capturing distribution mismatches, and stable during training (Lipman et al., 2022). Therefore, integrating it could lead to better estimation of the terminal distribution, and faster, more stable convergence when training the diffusion model which is left to the future works.

## D  STABILITY OF GENETIC CO-EXPRESSION RELATIONS

**Stability of genetic co-expression relations.** The majority of co-expression relationships among gene pairs remain relatively stable over time, as evidenced by the first column of Fig. 3. Population regularization effectively preserves this stability, a feature that is often lost with individual-level regularization by comparing between the second and third columns of Fig. 3.

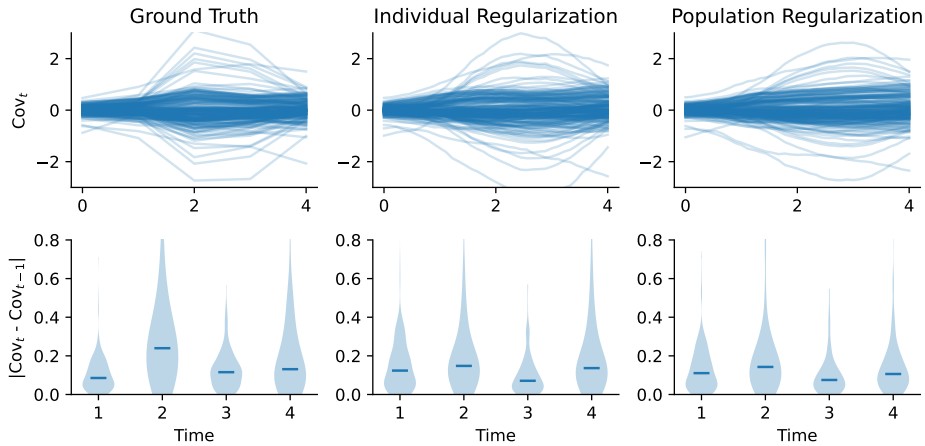

**Figure 3:** Visualization of temporal variations of the covariance of embryonic stem cell expression. The first row of figures presents direct plots of the covariance at time $t$, while the second row displays violin plots illustrating the differences between time $t$ and $t-1$.

## E  FORMULATION TO RESTRAIN "ACCELERATION" OF STANDARDIZED COVARIANCE

The formulation to restrain the "acceleration" of standardized covariance is expressed as:

$$\sum_{\widetilde{\mathcal{M}}\in\mathcal{M}_{\mathrm{cov}}} L_{\mathrm{std\text{-}corr}}(\pi_t,\widetilde{\mathcal{M}},2) = \left\|\frac{\mathrm{d}^2}{\mathrm{d}t^2}\left(\mathbb{E}_{\pi_t}[\mathbf{x}_t\mathbf{x}_t^\top] - \mathbb{E}_{\pi_t}[\mathbf{x}_t]\mathbb{E}_{\pi_t}^\top[\mathbf{x}_t]\right)\right\|_{\mathsf{F}}^2$$

$$\overbrace{= \left\|\mathbb{E}_{\pi_t}\left[\mathbf{x}_t(\frac{\mathrm{d}}{\mathrm{d}t}\boldsymbol{v}_t(\mathbf{x}_t))^\top + (\frac{\mathrm{d}}{\mathrm{d}t}\boldsymbol{v}_t(\mathbf{x}_t))\mathbf{x}_t^\top + \frac{1}{2}\frac{\mathrm{d}}{\mathrm{d}t}(\boldsymbol{\Sigma}_t(\mathbf{x}_t)\boldsymbol{\Sigma}_t^\top(\mathbf{x}_t))\right]}^{\text{Matrix collection of } \Gamma(\widetilde{\mathcal{S}}'\cup\Upsilon'_{<i',j'+1>}) \text{ terms in Eq. (14)}}$$

$$- \mathbb{E}_{\pi_t}[\mathbf{x}_t]\mathbb{E}_{\pi_t}^\top[\frac{\mathrm{d}}{\mathrm{d}t}\boldsymbol{v}_t(\mathbf{x}_t)] - \mathbb{E}_{\pi_t}[\frac{\mathrm{d}}{\mathrm{d}t}\boldsymbol{v}_t(\mathbf{x}_t)]\mathbb{E}_{\pi_t}^\top[\mathbf{x}_t] \qquad + \quad \overbrace{\mathbf{0}}^{\Gamma(\widetilde{\mathcal{S}}'\cup\Upsilon'_{<i'+1,j'>})}$$

$$+ \mathbb{E}_{\pi_t}\overbrace{\left[\mathbf{x}_t(\nabla(\boldsymbol{v}_t(\mathbf{x}_t))\boldsymbol{v}_t(\mathbf{x}_t))^\top + (\nabla(\boldsymbol{v}_t(\mathbf{x}_t))\boldsymbol{v}_t(\mathbf{x}_t))\mathbf{x}_t^\top + 2\boldsymbol{v}_t(\mathbf{x}_t)\boldsymbol{v}_t(\mathbf{x}_t)^\top\right.}^{\Gamma(\widetilde{\mathcal{S}}\cup\nabla_{<0,0>})}$$

$$+ \left.\frac{1}{2}\nabla(\boldsymbol{\Sigma}_t(\mathbf{x}_t)\boldsymbol{\Sigma}_t^\top(\mathbf{x}_t))_{i_1i_2i_3}\boldsymbol{v}_t^{i_3}(\mathbf{x}_t)\right]$$

$$- \mathbb{E}_{\pi_t}[\mathbf{x}_t]\mathbb{E}_{\pi_t}^\top[\nabla(\boldsymbol{v}_t(\mathbf{x}_t))\boldsymbol{v}_t(\mathbf{x}_t)] - \mathbb{E}_{\pi_t}[\nabla(\boldsymbol{v}_t(\mathbf{x}_t))\boldsymbol{v}_t(\mathbf{x}_t)]\mathbb{E}_{\pi_t}^\top[\mathbf{x}_t] - 2\mathbb{E}_{\pi_t}[\boldsymbol{v}_t(\mathbf{x}_t)]\mathbb{E}_{\pi_t}^\top[\boldsymbol{v}_t(\mathbf{x}_t)]$$

$$+ \mathbb{E}_{\pi_t}\overbrace{\left[\nabla(\boldsymbol{v}_t(\mathbf{x}_t))\boldsymbol{\Sigma}_t(\mathbf{x}_t)\boldsymbol{\Sigma}_t^\top(\mathbf{x}_t) + \boldsymbol{\Sigma}_t(\mathbf{x}_t)\boldsymbol{\Sigma}_t^\top(\mathbf{x}_t)\nabla^\top(\boldsymbol{v}_t(\mathbf{x}_t))\right.}^{\Gamma(\widetilde{\mathcal{S}}\cup\nabla\nabla^\top_{<0,0>})}$$

$$+ \frac{1}{4}\nabla\nabla^\top(\boldsymbol{\Sigma}_t(\mathbf{x}_t)\boldsymbol{\Sigma}_t^\top(\mathbf{x}_t))_{i_1i_2i_3i_4}(\boldsymbol{\Sigma}_t(\mathbf{x}_t)\boldsymbol{\Sigma}_t^\top(\mathbf{x}_t))^{i_3i_4}$$

$$+ \left.\frac{1}{2}\mathbf{x}_t(\nabla\nabla^\top(\boldsymbol{v}_t(\mathbf{x}_t))_{\underline{i_1i_2i_3}}(\boldsymbol{\Sigma}_t(\mathbf{x}_t)\boldsymbol{\Sigma}_t^\top(\mathbf{x}_t))^{\underline{i_2i_3}})^\top + \frac{1}{2}(\nabla\nabla^\top(\boldsymbol{v}_t(\mathbf{x}_t))_{\underline{i_1i_2i_3}}(\boldsymbol{\Sigma}_t(\mathbf{x}_t)\boldsymbol{\Sigma}_t^\top(\mathbf{x}_t))^{\underline{i_2i_3}})\mathbf{x}_t^\top\right]$$

$$- \frac{1}{2}\mathbb{E}_{\pi_t}[\mathbf{x}_t]\mathbb{E}_{\pi_t}^\top[\nabla\nabla^\top(\boldsymbol{v}_t(\mathbf{x}_t))_{\underline{i_1i_2i_3}}(\boldsymbol{\Sigma}_t(\mathbf{x}_t)\boldsymbol{\Sigma}_t^\top(\mathbf{x}_t))^{\underline{i_2i_3}}]$$

$$- \frac{1}{2}\mathbb{E}_{\pi_t}[\nabla\nabla^\top(\boldsymbol{v}_t(\mathbf{x}_t))_{\underline{i_1i_2i_3}}(\boldsymbol{\Sigma}_t(\mathbf{x}_t)\boldsymbol{\Sigma}_t^\top(\mathbf{x}_t))^{\underline{i_2i_3}}]\mathbb{E}_{\pi_t}^\top[\mathbf{x}_t]\Big\|_{\mathsf{F}}^2.$$

## F  FORM OF COVARIANCE POTENTIAL

Denoting $\boldsymbol{D} = \mathbb{E}_{\pi_t}[\mathbf{x}_t\mathbf{x}_t^\top] - \mathbb{E}_{\pi_t}[\mathbf{x}_t]\mathbb{E}_{\pi_t}^\top[\mathbf{x}_t]$ $\widetilde{\boldsymbol{D}} \in \mathbb{R}^{d\times d}, \widetilde{\boldsymbol{D}}_{[i,j]} = \left\{\begin{smallmatrix}0,\text{ if } i\neq j\\ 1/\sqrt{\boldsymbol{D}_{[i,j]}},\text{ else,}\end{smallmatrix}\right.$, the designated form of potential covariance is expressed as:

$$\sum_{\widetilde{\mathcal{M}}\in\mathcal{M}_{\mathrm{cov}}} L_{\mathrm{corr}}(\pi_t,\widetilde{\mathcal{M}},0) = \left\|\widetilde{\boldsymbol{D}}^\top\boldsymbol{D}\widetilde{\boldsymbol{D}} - \boldsymbol{Y}\right\|_{\mathsf{F}}^2.$$

## G  STANDARDIZED COVARIANCE KINETICS WITHIN A NON-LINEAR PROJECTED SPACE

We assume the diffusion generative model is built in a non-linear latent space as:

$$\mathbf{x}_t = \overleftarrow{h}(\mathbf{z}_t), \quad \mathrm{d}\mathbf{z}_t = f(\mathbf{z}_t)\mathrm{d}t + D(\mathbf{z}_t)\mathrm{d}\mathbf{w}_t,$$

and then the correlation Lagrangian can be computed as:

$$\frac{\mathrm{d}}{\mathrm{d}t}\mathbb{E}[\mathbf{x}_{t,[i]}\mathbf{x}_{t,[j]}] = \frac{\mathrm{d}}{\mathrm{d}t}\mathbb{E}[\overleftarrow{h}_{[i]}(\mathbf{z}_t)\overleftarrow{h}_{[j]}(\mathbf{z}_t)]$$

$$= \mathbb{E}\left[\nabla\{\overleftarrow{h}_{[i]}(\mathbf{z}_t)\overleftarrow{h}_{[j]}(\mathbf{z}_t)\}\cdot f(\mathbf{z}_t) + \frac{1}{2}\nabla^2\{\overleftarrow{h}_{[i]}(\mathbf{z}_t)\overleftarrow{h}_{[j]}(\mathbf{z}_t)\}\cdot D^2(\mathbf{z}_t)\right]$$

$$= \mathbb{E}\left[\overleftarrow{h}_{[j]}(\mathbf{z}_t)(\nabla\overleftarrow{h}_{[i]}(\mathbf{z}_t)\cdot f(\mathbf{z}_t)) + \overleftarrow{h}_{[i]}(\mathbf{z}_t)(\nabla\overleftarrow{h}_{[j]}(\mathbf{z}_t)\cdot f(\mathbf{z}_t))\right.$$

$$+ \frac{1}{2}\Big\{\nabla\overleftarrow{h}_{[i]}(\mathbf{z}_t)\nabla\overleftarrow{h}_{[j]}^\top(\mathbf{z}_t) + \nabla\overleftarrow{h}_{[j]}(\mathbf{z}_t)\nabla\overleftarrow{h}_{[i]}^\top(\mathbf{z}_t) + \overleftarrow{h}_{[i]}(\mathbf{z}_t)\nabla^2\overleftarrow{h}_{[j]}(\mathbf{z}_t) + \overleftarrow{h}_{[j]}(\mathbf{z}_t)\nabla^2\overleftarrow{h}_{[i]}(\mathbf{z}_t)\Big\} \cdot D^2(\mathbf{z}_t)\Big].$$

The first part of the last line can be written in the matrix form as:

$$\text{Part 1} = \mathbb{E}\Big[\overleftarrow{h}(\mathbf{z}_t)(\nabla\overleftarrow{h}(\mathbf{z}_t)f(\mathbf{z}_t))^\top + (\nabla\overleftarrow{h}(\mathbf{z}_t)f(\mathbf{z}_t))\overleftarrow{h}^\top(\mathbf{z}_t)\Big].$$

The second part of the last line can be written in the matrix form as:

$$\text{Part 2} = \frac{1}{2}\mathbb{E}\Big[\Big\langle\Big(\langle\nabla\overleftarrow{h}(\mathbf{z}_t)_{\underline{i_1i_2}}, \nabla\overleftarrow{h}(\mathbf{z}_t)_{\underline{i_3i_4}}\rangle^{\underline{i_1i_3i_2i_4}} + \langle\nabla\overleftarrow{h}(\mathbf{z}_t)_{\underline{i_1i_2}}, \nabla\overleftarrow{h}(\mathbf{z}_t)_{\underline{i_3i_4}}\rangle^{\underline{i_3i_1i_2i_4}}$$

$$+ \langle\overleftarrow{h}(\mathbf{z}_t)_{\underline{i_1}}, \nabla^2\overleftarrow{h}(\mathbf{z}_t)_{\underline{i_2i_3i_4}}\rangle^{\underline{i_1i_2i_3i_4}} + \langle\overleftarrow{h}(\mathbf{z}_t)_{\underline{i_1}}, \nabla^2\overleftarrow{h}(\mathbf{z}_t)_{\underline{i_2i_3i_4}}\rangle^{\underline{i_2i_1i_3i_4}}\Big)_{\underline{i_1i_2i_3i_4}}, D^2(\mathbf{z}_t)_{\underline{i_3i_4}}\Big\rangle^{\underline{i_1i_2}}\Big],$$

which can be simplified as follows if $D$ is diagonal:

$$\text{Part 2} = \frac{1}{2}\mathbb{E}\Big[\Big\langle\Big(\langle\nabla\overleftarrow{h}(\mathbf{z}_t)_{\underline{i_1i_2}}, \nabla\overleftarrow{h}(\mathbf{z}_t)_{\underline{i_3i_2}}\rangle^{\underline{i_1i_3i_2}} + \langle\nabla\overleftarrow{h}(\mathbf{z}_t)_{\underline{i_1i_2}}, \nabla\overleftarrow{h}(\mathbf{z}_t)_{\underline{i_3i_2}}\rangle^{\underline{i_3i_1i_2}}$$

$$+ \langle\overleftarrow{h}(\mathbf{z}_t)_{\underline{i_1}}, \nabla^2_{\text{Diag}}\overleftarrow{h}(\mathbf{z}_t)_{\underline{i_2i_3}}\rangle^{\underline{i_1i_2i_3}} + \langle\overleftarrow{h}(\mathbf{z}_t)_{\underline{i_1}}, \nabla^2_{\text{Diag}}\overleftarrow{h}(\mathbf{z}_t)_{\underline{i_2i_3}}\rangle^{\underline{i_2i_1i_3}}\Big)_{\underline{i_1i_2i_3}}, D^2(\mathbf{z}_t)_{\underline{i_3}}\Big\rangle^{\underline{i_1i_2}}\Big].$$

Since based on Propos. 1 we have:

$$\frac{\mathrm{d}}{\mathrm{d}t}\mathbb{E}[\overleftarrow{h}_{[i]}(\mathbf{z}_t)] = \mathbb{E}[\nabla\overleftarrow{h}_{[i]}(\mathbf{z}_t)\cdot f(\mathbf{z}_t) + \frac{1}{2}\nabla^2\overleftarrow{h}_{[i]}(\mathbf{z}_t)\cdot D^2(\mathbf{z}_t)],$$

and then we can express the normalized form as:

$$\frac{\mathrm{d}}{\mathrm{d}t}\Big(\mathbb{E}[\overleftarrow{h}(\mathbf{z}_t)\overleftarrow{h}^\top(\mathbf{z}_t)] - \mathbb{E}[\overleftarrow{h}(\mathbf{z}_t)]\mathbb{E}^\top[\overleftarrow{h}^\top(\mathbf{z}_t)]\Big)$$

$$= \text{Part 1} + \text{Part 2} - \mathbb{E}[\overleftarrow{h}(\mathbf{z}_t)]\mathbb{E}^\top[\nabla\overleftarrow{h}(\mathbf{z}_t)f(\mathbf{z}_t)] - \mathbb{E}[\nabla\overleftarrow{h}(\mathbf{z}_t)f(\mathbf{z}_t)]\mathbb{E}^\top[\overleftarrow{h}(\mathbf{z}_t)]$$

$$- \frac{1}{2}\mathbb{E}[\overleftarrow{h}(\mathbf{z}_t)]\mathbb{E}^\top[\langle\nabla^2\overleftarrow{h}(\mathbf{z}_t)_{\underline{i_1i_2i_3}}, D^2(\mathbf{z}_t)_{\underline{i_2i_3}}\rangle^{\underline{i_1}}] - \frac{1}{2}\mathbb{E}[\langle\nabla^2\overleftarrow{h}(\mathbf{z}_t)_{\underline{i_1i_2i_3}}, D^2(\mathbf{z}_t)_{\underline{i_2i_3}}\rangle^{\underline{i_1}}]\mathbb{E}^\top[\overleftarrow{h}(\mathbf{z}_t)].$$

# H  NEURAL NETWORK PARAMETRIZATION OF (CONDITIONAL) NEURAL SDEs

**Neural network parametrization of drift $v_t(\cdot;\theta)$ and diffusion $\Sigma_t(\cdot;\theta)$.** We follow the architecture in (Onken et al., 2021) to parametrize the drift and diffusion functions. Specifically, given the time embedding $z_t$ for a time stamp $t$, we first construct a potential function $\Phi_t : \mathbb{R}^d \to \mathbb{R}$ as:

$$\Phi_t(\boldsymbol{x}) = \boldsymbol{w}^T\text{MLP}(\text{CAT}(\boldsymbol{x}, \boldsymbol{z}_t);\phi_1) + \frac{1}{2}\boldsymbol{x}^\top\boldsymbol{A}^\top\boldsymbol{A}\boldsymbol{x} + \boldsymbol{b}^\top\boldsymbol{x} + c,$$

where $\text{MLP}(\cdot;\phi_1)$ is a multi-layer perceptron, and $\text{CAT}(\cdot)$ is the concatenation function. The drift is then computed by taking the gradient as:

$$\boldsymbol{v}_t(\boldsymbol{x};\theta) = \nabla_{\boldsymbol{x}}\Phi_t(\boldsymbol{x}).$$

For the diffusion, we simply construct it as:

$$\boldsymbol{\Sigma}_t(\boldsymbol{x};\theta) = \text{MLP}(\text{CAT}(\boldsymbol{x}, \boldsymbol{z}_t);\phi_2).$$

The learnable parameter collection is expressed as $\theta = \{\phi_1, \phi_2, \boldsymbol{w}, \boldsymbol{A}, \boldsymbol{b}, c\}$. Model predictions are generated through SDE simulation using Eq. (1) with the torchsde library (Li et al., 2020).

**Neural network parametrization of conditional drift and diffusion.** A small-molecule drug can be routinely represented as a graph $\mathcal{G}$ (You et al., 2020). Thus, we leverage graph neural networks (GNNs) to embed it into vector space as $\boldsymbol{z}_{\mathcal{G}} = \text{GNN}(\mathcal{G};\phi_3)$. The conditional drift and diffusion are then expressed as:

$$\boldsymbol{v}_t(\boldsymbol{x};\theta) = \nabla_{\boldsymbol{x}}\Big(\boldsymbol{w}^T\text{MLP}(\text{CAT}(\boldsymbol{x}, \boldsymbol{z}_t, \boldsymbol{z}_{\mathcal{G}});\phi_1) + \frac{1}{2}\boldsymbol{x}^\top\boldsymbol{A}^\top\boldsymbol{A}\boldsymbol{x} + \boldsymbol{b}^\top\boldsymbol{x} + c\Big),$$

$$\boldsymbol{\Sigma}_t(\boldsymbol{x};\theta) = \text{MLP}(\text{CAT}(\boldsymbol{x}, \boldsymbol{z}_t, \boldsymbol{z}_{\mathcal{G}});\phi_2).$$

The learnable parameter collection is expressed as $\theta = \{\phi_1, \phi_2, , \phi_3, \boldsymbol{w}, \boldsymbol{A}, \boldsymbol{b}, c\}$. We adopt the graph attention network architecture (Velickovic et al., 2017) for drug embedding.

**Computational resources.** Experiments are distributed on computer clusters with NVIDIA A100 GPU (40 GB memory), which in general can be finished within two days.

**Additional details on the dataset of conditional generation.** Dimensionality: The generative model training is conducted in 256 hidden dimensions. The hidden (latent) space is constructed by training an autoencoder on the training data, which contains the 2,000 most differentially expressed genes. Pre-processing: The pre-processing of sci-plex is standardized by adopting the code from (Wu et al., 2022). Steps include QC filtering, normalization, log1p transformation, and differentially expressed gene selection. Number of drugs: All 188 drugs contained in the dataset are used. Split: In the paper, we focus on dose-effect prediction conditional on different drug perturbations, each labeled with five dose effects ($t_0$-$t_4$). We use the dose effects of $t_0$&$t_4$ for training and validation, and perform testing on $t_1$-$t_3$ as described in the main text. We also used the perturbation split to test performances on new drugs.

Regarding the significance of dose splitting, understanding the dose-effect relationship is crucial in therapeutics. Intuitively, the dose impacts drug concentration, which can lead to very different phenotypic outcomes (Holford & Sheiner, 1981). The sci-plex dataset provides treated cellular expressions under various drugs and doses. We therefore treat dose as a pseudo-time variable and construct a conditional generative model to simulate the evolution of dose effects. Similar efforts are described in (Lotfollahi et al., 2023), which are useful for guiding the clinical use of new drugs.

We also conducted an experiment using a dataset split based on drug perturbations and compared it with the SOTA CellOT (Bunne et al., 2023) in our implementation. The results, presented in Tab. 5, demonstrate the effectiveness of our method.

**Table 5:** Experiments on the sci-plex dataset based on drug perturbation split (predicting dose-dependent cellular response to new drugs).

| Methods | VAE | CellOT | Ours |
|---------|-----|--------|------|
| WDist $\downarrow$ | 8.07 | 1.42 | 1.38 |

# I  MORE RESULTS AND VISUALIZATIONS

**Hyperparameter tuning.** The appropriate weighting of different loss functions in the unconstrained optimization (13) for an approximated CLSB solution is important. We perform tuning for $\alpha_{\mathrm{corr},0}$ in {1e-2, 1e-1, 1, 1e1, 1e2}, $\alpha_{\mathrm{corr},1}$ in {1, 1e1, 1e2, 1e3, 1e4}, and $\alpha_{\mathrm{corr},2}$ in {1, 1e1, 1e2, 1e3, 1e4} via grid search. Validation results are shown in Tab. 6 and test results in Tab. 7. $\alpha_{\mathrm{ind}}$ is tuned in {0, 1}, which does not lead to a significant impact on performance.

For the experiment in Sec. 4.1: Tab. 6 provides the validation performance for a single type of correlational regularization (out of a total of three as in Opt. (13)), and Tab. 7 showcases their corresponding test performances. The ultimate test performance in Tab. 2 is achieved by applying all three regularizations with weights tuned according to Tab. 6.

Intuition: By experimenting with the single regularization presented in Tab. 7, we aim to understand how the three types of regularizations contribute differently to the ultimate performance: Regularization on "position" provides less benefit compared to the other two; "acceleration" benefits the early stages more, and "velocity" provides more benefit in the later stages. For the experiment in Sec. 4.2: We simply adopt the hyperparameter setting from Sec. 4.1.

**Evaluation of the original gene expressions in conditional generation (Sec. 4.2).** We also perform evaluations on the original gene expressions beyond principal components, as shown in Tab. 8. We compute the Wasserstein distance between gene expressions and calculate both the mean and median across all drug conditions, with mean and standard deviation computed for all genes. The Wasserstein distance is computed using the SciPy library (Virtanen et al., 2020).

**Table 6:** Evaluation on the validation data in the unconditional generation scenario of developmental modeling of embryonic stem cells.

| Methods | All-Step Prediction | | | One-Step Prediction | | |
|---|---|---|---|---|---|---|
| | $t_1$ | $t_2$ | $t_3$ | $t_1$ | $t_2$ | $t_3$ |
| $\alpha_{corr,0}=0, \alpha_{corr,1}=0, \alpha_{corr,2}=0$ | 1.563±0.008 | 1.916±0.008 | 1.695±0.018 | 1.561±0.008 | 1.362±0.011 | 1.067±0.017 |
| $\alpha_{corr,0}=1e-2, \alpha_{corr,1}=0, \alpha_{corr,2}=0$ | 1.532±0.008 | 1.886±0.012 | 1.670±0.016 | 1.533±0.007 | 1.356±0.011 | 1.051±0.018 |
| $\alpha_{corr,0}=1e-1, \alpha_{corr,1}=0, \alpha_{corr,2}=0$ | 1.618±0.006 | 1.968±0.008 | 1.701±0.015 | 1.617±0.005 | 1.395±0.011 | 1.093±0.019 |
| $\alpha_{corr,0}=1, \alpha_{corr,1}=0, \alpha_{corr,2}=0$ | 1.598±0.007 | 1.949±0.010 | 1.700±0.017 | 1.598±0.008 | 1.367±0.012 | 1.053±0.020 |
| $\alpha_{corr,0}=1e1, \alpha_{corr,1}=0, \alpha_{corr,2}=0$ | 1.635±0.008 | 1.736±0.014 | 1.514±0.022 | 1.635±0.008 | 1.273±0.014 | 1.062±0.018 |
| $\alpha_{corr,0}=1e2, \alpha_{corr,1}=0, \alpha_{corr,2}=0$ | 1.653±0.009 | 1.743±0.018 | 1.672±0.030 | 1.651±0.010 | 1.471±0.014 | 1.209±0.013 |
| $\alpha_{corr,0}=0, \alpha_{corr,1}=1, \alpha_{corr,2}=0$ | 1.547±0.008 | 1.895±0.008 | 1.678±0.018 | 1.547±0.006 | 1.345±0.011 | 1.048±0.019 |
| $\alpha_{corr,0}=0, \alpha_{corr,1}=1e1, \alpha_{corr,2}=0$ | 1.471±0.009 | 1.801±0.009 | 1.642±0.018 | 1.471±0.007 | 1.293±0.011 | 1.040±0.018 |
| $\alpha_{corr,0}=0, \alpha_{corr,1}=1e2, \alpha_{corr,2}=0$ | 1.337±0.008 | 1.628±0.009 | 1.538±0.021 | 1.337±0.007 | 1.200±0.013 | 0.967±0.018 |
| $\alpha_{corr,0}=0, \alpha_{corr,1}=1e3, \alpha_{corr,2}=0$ | 1.053±0.007 | 1.484±0.010 | 1.549±0.019 | 1.052±0.007 | 1.098±0.015 | 0.910±0.015 |
| $\alpha_{corr,0}=0, \alpha_{corr,1}=1e4, \alpha_{corr,2}=0$ | 1.042±0.004 | 1.482±0.009 | 1.494±0.018 | 1.041±0.005 | 1.129±0.012 | 0.927±0.017 |
| $\alpha_{corr,0}=0, \alpha_{corr,1}=0, \alpha_{corr,2}=1$ | 0.982±0.005 | 1.470±0.010 | 1.482±0.019 | 0.983±0.005 | 1.135±0.013 | 0.985±0.018 |
| $\alpha_{corr,0}=0, \alpha_{corr,1}=0, \alpha_{corr,2}=1e1$ | 1.074±0.010 | 1.499±0.016 | 1.556±0.025 | 1.074±0.011 | 1.095±0.013 | 0.896±0.016 |
| $\alpha_{corr,0}=0, \alpha_{corr,1}=0, \alpha_{corr,2}=1e2$ | 1.110±0.015 | 1.498±0.016 | 1.511±0.024 | 1.111±0.012 | 1.064±0.014 | 0.901±0.017 |
| $\alpha_{corr,0}=0, \alpha_{corr,1}=0, \alpha_{corr,2}=1e3$ | 1.277±0.018 | 1.662±0.019 | 1.586±0.023 | 1.281±0.016 | 1.101±0.010 | 0.859±0.012 |
| $\alpha_{corr,0}=0, \alpha_{corr,1}=0, \alpha_{corr,2}=1e4$ | 1.341±0.014 | 1.786±0.015 | 1.761±0.020 | 1.345±0.013 | 1.142±0.009 | 0.849±0.014 |

**Table 7:** Evaluation on the test data in the unconditional generation scenario of developmental modeling of embryonic stem cells.

| Methods | All-Step Prediction | | | One-Step Prediction | | |
|---|---|---|---|---|---|---|
| | $t_1$ | $t_2$ | $t_3$ | $t_1$ | $t_2$ | $t_3$ |
| $\alpha_{corr,0}=0, \alpha_{corr,1}=0, \alpha_{corr,2}=0$ | 1.499±0.005 | 1.945±0.006 | 1.619±0.016 | 1.498±0.005 | 1.418±0.009 | 0.966±0.016 |
| $\alpha_{corr,0}=1e-2, \alpha_{corr,1}=0, \alpha_{corr,2}=0$ | 1.468±0.005 | 1.908±0.007 | 1.586±0.015 | 1.467±0.004 | 1.416±0.009 | 0.957±0.016 |
| $\alpha_{corr,0}=0, \alpha_{corr,1}=1e3, \alpha_{corr,2}=0$ | 1.035±0.005 | 1.557±0.012 | 1.523±0.021 | 1.034±0.005 | 1.164±0.011 | 0.865±0.014 |
| $\alpha_{corr,0}=0, \alpha_{corr,1}=0, \alpha_{corr,2}=1$ | 0.946±0.004 | 1.503±0.007 | 1.440±0.015 | 0.946±0.004 | 1.205±0.008 | 0.917±0.013 |

**Table 8:** Evaluation in the conditional generation scenario of dose-dependent cellular response prediction to chemical perturbations. Numbers ($\times$1e-3) indicate the mean and median Wasserstein distances for all genes, where lower values are preferable.

| Methods | $t_1$ | | $t_2$ (Most Challenging) | | $t_3$ | |
|---|---|---|---|---|---|---|
| | Mean | Median | Mean | Median | Mean | Median |
| Random | 573.0±51.3 | 516.9±24.2 | 578.7±51.7 | 520.2±24.6 | 577.5±52.9 | 519.0±25.1 |
| NeuralSDE(RandInit) | 529.9±73.4 | 578.7±46.2 | 536.5±73.4 | 592.5±45.3 | 536.3±75.8 | 585.7±53.0 |
| VAE | 227.6±87.5 | 168.6±107.1 | 215.7±74.7 | 159.6±87.8 | 210.0±70.0 | 150.5±77.7 |
| NeuralODE | 177.7±43.2 | 108.3±38.7 | 192.3±56.1 | 119.8±50.0 | 183.5±58.4 | 115.5±51.4 |
| NeuralSDE | 170.1±40.8 | 102.0±44.8 | 183.0±53.2 | 117.3±57.1 | 182.3±63.4 | 117.8±55.0 |
| NLSB(E) | 78.6±35.1 | 59.8±29.3 | 93.1±43.1 | 70.0±35.8 | 104.0±47.9 | 75.9±39.6 |
| CLSB($\alpha_{ind}>0$) | 79.3±36.4 | 61.0±29.6 | 87.3±44.8 | 67.0±34.6 | 92.0±50.0 | 69.4±38.1 |
| CLSB($\alpha_{ind}=0$) | 76.9±33.7 | 60.9±27.4 | 89.1±39.9 | 72.2±32.5 | 93.3±43.2 | 74.8±35.4 |

**Terminal state evaluation.** Our model not only demonstrates advantages in generating the intermediate state populations between $t_1$ and $t_3$ as shown in Sec. 4, but it also excels in generating the terminal state at $t_4$, as illustrated in the Tabs. 9 & 10.

**Table 9:** Evaluation on the terminal state $t_4$ for the stem-cell dataset.

| Methods | OT-Flow | OT-Flow+OT | TrajectoryNet | TrajectoryNet+OT | NLSB(E) | NLSB(E+D+V) | Ours(CLSB-$\alpha$ >0) | Ours(CLSB-$\alpha$ =0) |
|---|---|---|---|---|---|---|---|---|
| WDist ↓ | 0.799 | 0.748 | 0.702 | 0.692 | 0.755 | 0.716 | 0.707 | 0.687 |

**Table 10:** Evaluation on the terminal state $t_4$ for the sci-Plex dataset.

| Methods | NeuralSDE(RandInit) | VAE | NeuralODE | NeuralSDE | NLSB(E) | Ours(CLSB-$\alpha$ >0) | Ours(CLSB-$\alpha$ =0) |
|---|---|---|---|---|---|---|---|
| WDist ↓ | 2.26 | 1.03 | 1.04 | 1.07 | 1.28 | 0.80 | 0.70 |

**Experiment on an additional cell-differentiation dataset.** Beyond the experiment using the cell-differentiation dataset (Moon et al., 2019), we conducted additional experiments on a dataset curated from (Weinreb et al., 2020) to further validate the proposed population-level regularization. We adopted SBAlign (Somnath et al., 2023) as the base model (following the same experimental settings) and further restrained the covariance velocity (Eq. (8)) in the training objective. The results are shown in Tab. 11, demonstrating the effectiveness of our method on three out of four metrics.

**Table 11:** Means and standard deviations (in parentheses) of maximum-mean-discrepancy (MMD), $\ell_2$, RMSD, and cell-type classification accuracy on cellular expression simulation, following the evaluation pipeline of (Somnath et al., 2023).

| Methods | MMD ↓ | $\ell_2$ ↓ | RMSD ↓ | Classification Accuracy ↑ |
|---|---|---|---|---|
| w/o CorrLagr | 1.07e-2±0.01e-2 | 1.24±0.02 | 0.21e-1±0.01e-1 | 56.3%±0.7% |
| w/ CorrLagr | 1.61e-2±0.06e-2 | 1.07±0.04 | 8.93e-1±0.01e-1 | 57.6%±1.4% |

**Comparison with more SOTAs.** We further compare with more baselines including (Tong et al., 2023a;b). We follow the leave-one-out setting in (Tong et al., 2023a;b) and experiment on the embryonic body dataset with results shown in Tab. 12.

**Table 12:** Experiments on the embryonic stem cell dataset following the leave-one-out setting in (Tong et al., 2023a)

| Methods | DSBM | DSB | Reg.CFN | TrajNet | NLSB | OT-CFN | SF2M | Ours |
|---|---|---|---|---|---|---|---|---|
| WDist ↓ | 1.755 | 0.862 | 0.825 | 0.848 | 0.970 | 0.790 | 0.793 | 0.736 |

**Experiment on larger and higher-dimensional dataset of CITE-Seq.** We also examine the scalability of our model in the larger and higher-dimensional dataset of CITE-Seq (Kim et al., 2020). We follow the leave-one-out setting on 50 principal components as (Tong et al., 2023a), with the results shown in Tab. 13 strikingly demonstrate the distinguished scalability of our method. We believe the observed improvement is due to differences in the evaluation, where in the CITE-Seq experiment the distribution mismatch was evaluated using 50 PCs versus ≤10 PCs in the standard setting. Evaluating on more PCs further reveals the capability of different models in different aspects, showing how they capture the "main" distribution shifts (in the top PCs) versus how they handle "minor" distribution shifts. This interestingly demonstrates that our model effectively captures both "major" and "minor" distribution shifts during dynamic modeling.

**Table 13:** Experiments on the CITE-Seq dataset (high-dimensional setting) following the setting in (Tong et al., 2023a)

| Methods | DSBM | I-CFM | OT-CFM | SF2M | Ours |
|---|---|---|---|---|---|
| WDist ↓ | 53.81 | 41.83 | 38.76 | 38.52 | 9.07 |

**Experiment on a non-biological application of opinion depolarization.** Beyond single-cell applications, we conducted experiments on the application of opinion depolarization (Gaitonde et al., 2021) to further validate the proposed population-level regularization. Our focus is on a type of opinion dynamics that often results in strong polarization, meaning particles' opinions tend to form into distinct groups with opposite viewpoints. Take the party model as an example: particles receive random pieces of information from a predetermined distribution. They update their opinions based on these random inputs and an underlying rule: if the new information aligns with their current opinions, they are more likely to adopt it; if it contradicts, they typically reject it. This approach, known as biased assimilation, can easily lead to polarization, where the population divides into groups with strongly opposed views. For more background, we refer the readers to (Liu et al., 2022).

The dimension of opinion depolarization is 2 following the original setting. We adopt DeepGSB (Liu et al., 2022) as the base model, maintaining the same experimental settings, with two different parametrizations for the actor-critic and critic roles. Additionally, we further restrain the covariance

velocity (Eq. (8)) in the training objective. The results are shown in Tab. 14, demonstrating the effectiveness of our method.

**Table 14:** Wasserstein distance between the simulation and ground-truth in the opinion depolarization experiment, following the evaluation pipeline of (Liu et al., 2022).

| Methods | Actor-Critic Parametrization | Critic Parametrization |
|---|---|---|
| w/o CorrLagr | 8.45e-2 | 4.09e-2 |
| w/ CorrLagr | 8.36e-2 | 4.02e-2 |

**Experiment on synthetic datasets.** We conduct experiments on a synthetic dataset by referencing (Tong et al., 2023a), to learn the transport from 8 Gaussian (mixture of Gaussian) to 1 Gaussian distribution. To establish an ideal setting for evaluating our proposed regularization on correlation conservativeness, we intentionally ensure that the source and target distributions have the same covariance matrix. Thus, the PDFs of the source and target distributions are formulated as follows:

- Source 8 Gaussian: $\sum_{i=1}^{8} w_i \mathcal{N}(\mu_i, \Sigma_i)$ where $\sum_{i=1}^{8} w_i = 1, w_i \geq 0$;

- Target 1 Gaussian: $\mathcal{N}(\mu + d, \Sigma)$ where $\mu = \sum_{i=1}^{8} w_i \mu_i, \Sigma = \sum_{i=1}^{8} w_i \left(\Sigma_i + (\mu_i - \mu)(\mu_i - \mu)^\top\right)$;

- Here, $\|\mathbf{d}\|$ reflects the difficulty of learning such a transport from one aspect (the larger $\|\mathbf{d}\|$, the more difficult).

Building on the SF2M base model and its training paradigm (Tong et al., 2023a), we compare the performance with and without our correlational regularization, using the Wasserstein 1 distance as the metric. The Tab. 15 results demonstrate the effectiveness of our method, especially in difficult cases.

**Table 15:** Wasserstein distance between the simulation and ground-truth in the synthetic experiment, following the evaluation pipeline of (Tong et al., 2023a).

| $\|\mathbf{d}\| =$ | 50 | 100 | 200 |
|---|---|---|---|
| w/o CorrLagr | 1.48 | 1.92 | 3.27 |
| w/ CorrLagr | 1.39 | 1.63 | 1.77 |

**More visualization in unconditional generation (Sec. 4.1).** We provide more visualization of the simulated gene expressions (or their principal components) and trajectories as follows.

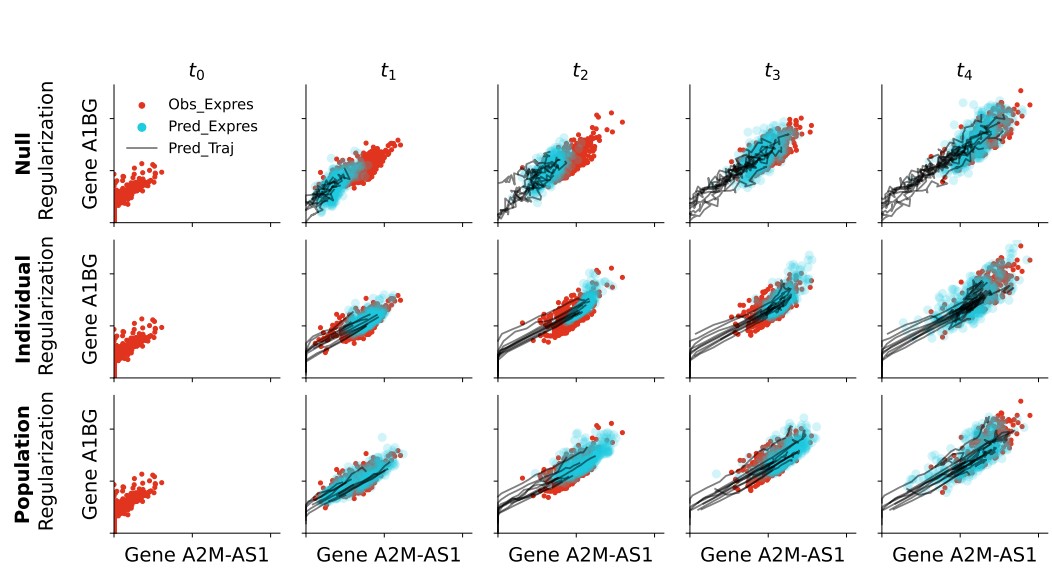

Figure 4: Visualization of local dynamics for genes A2M-AS1 and A1BG.

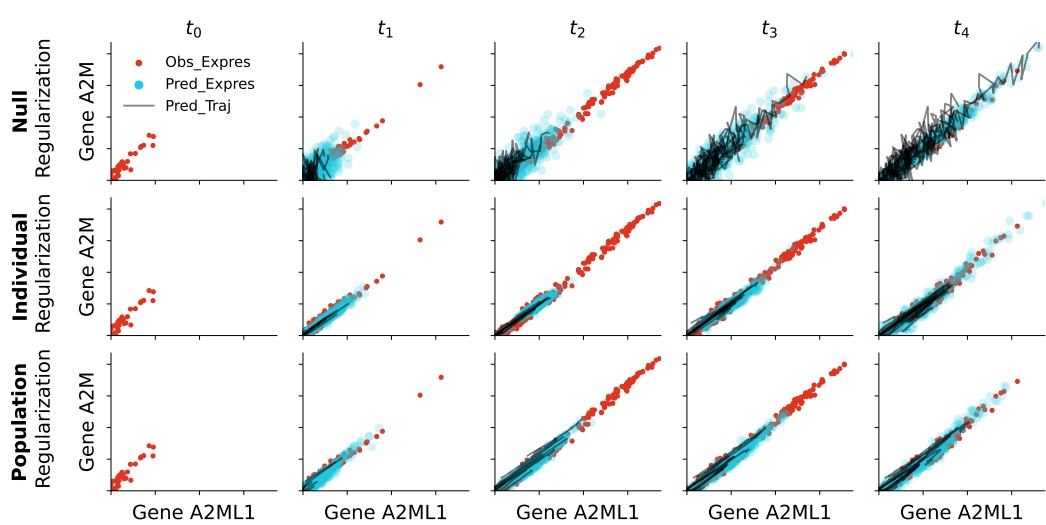

Figure 5: Visualization of local dynamics for genes A2ML1 and A2M.

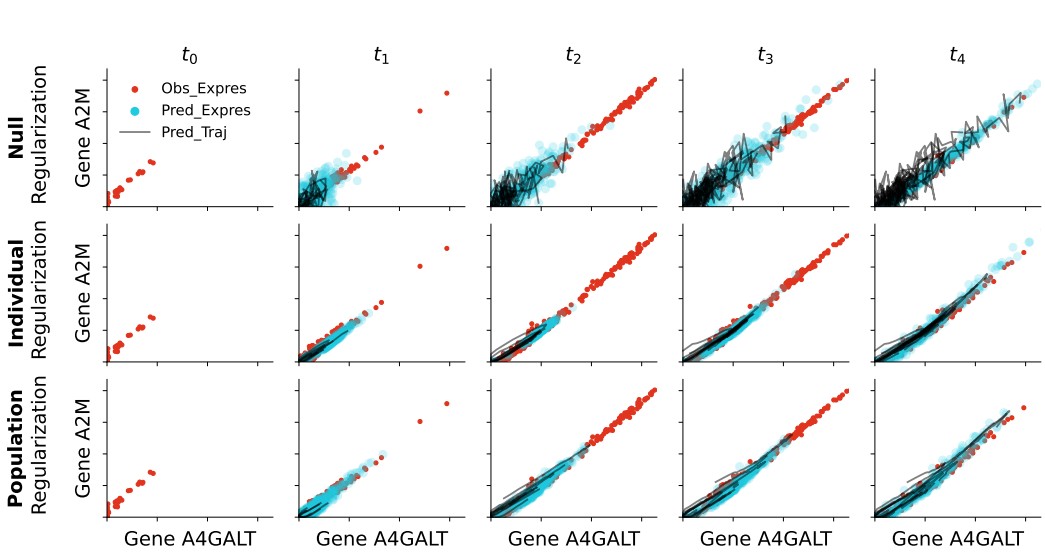

**Figure 6:** Visualization of local dynamics for genes A4GALT and A2M.

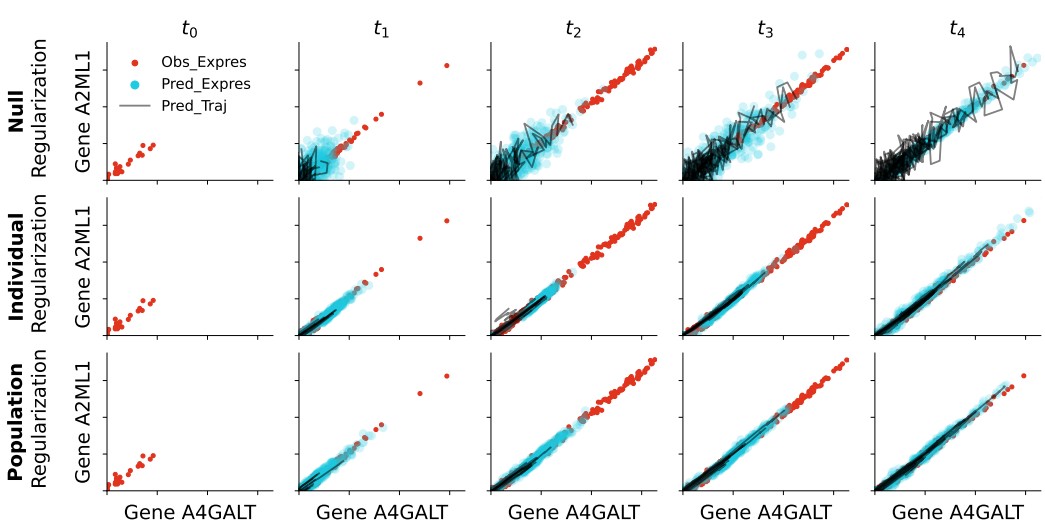

**Figure 7:** Visualization of local dynamics for genes A4GALT and A2ML1.

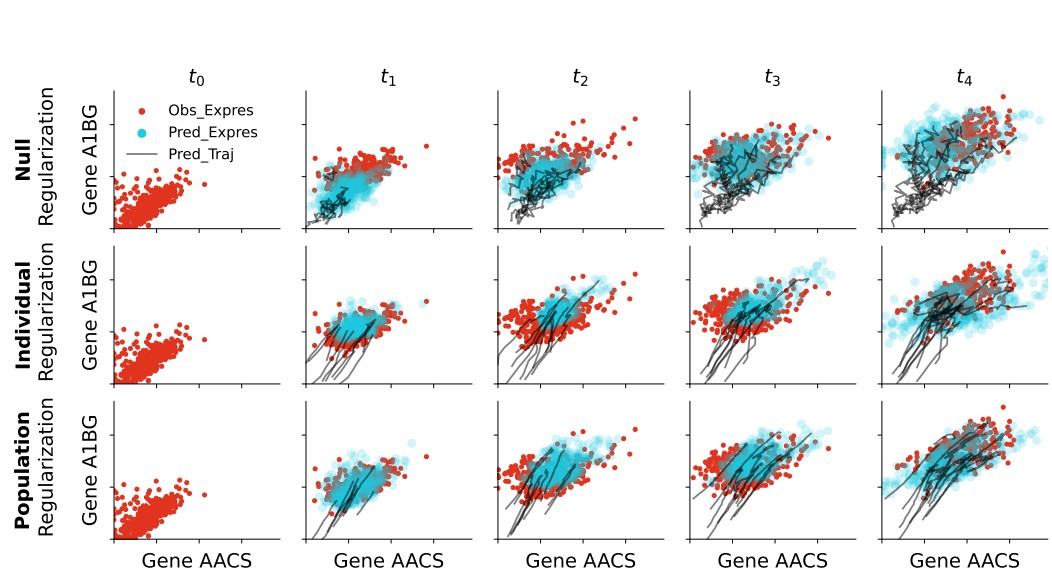

Figure 8: Visualization of local dynamics for genes AACS and A1BG.

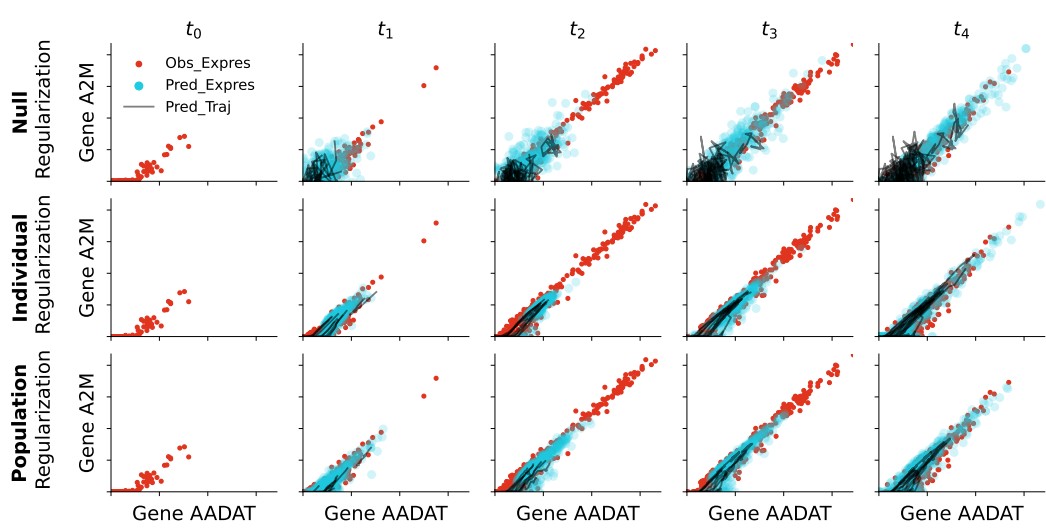

Figure 9: Visualization of local dynamics for genes AADAT and A2M.

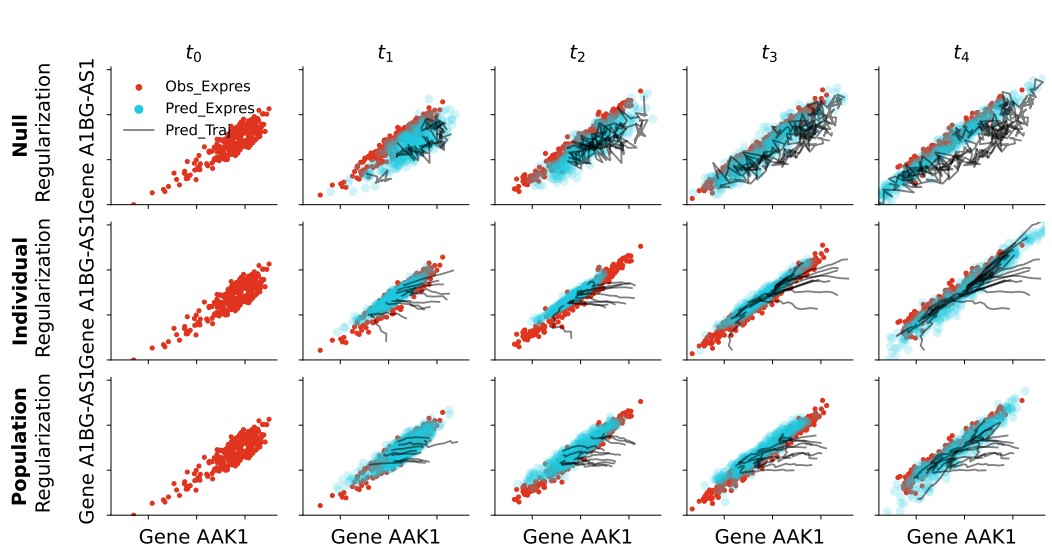

**Figure 10:** Visualization of local dynamics for genes AAK1 and A1BG-AS1.

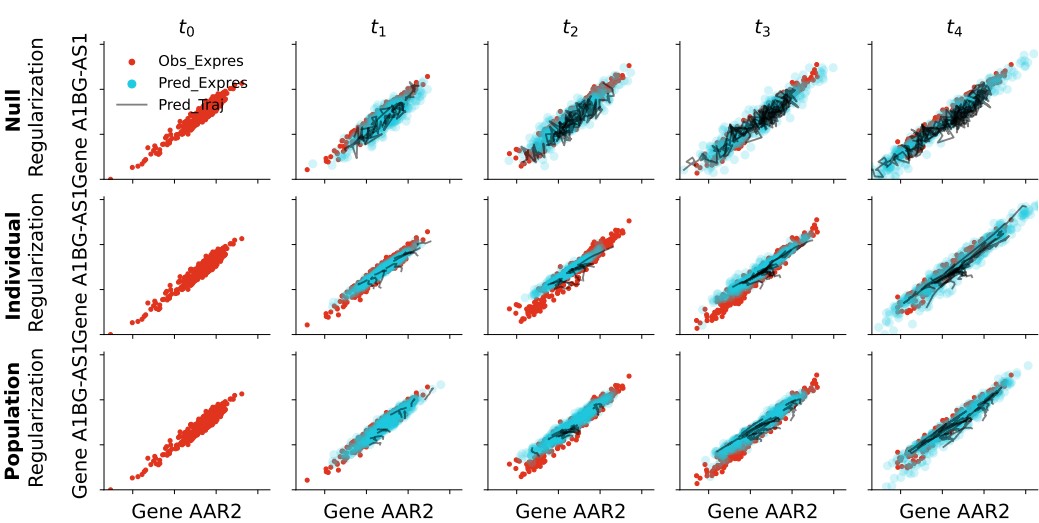

**Figure 11:** Visualization of local dynamics for genes AAR2 and A1BG-AS1.

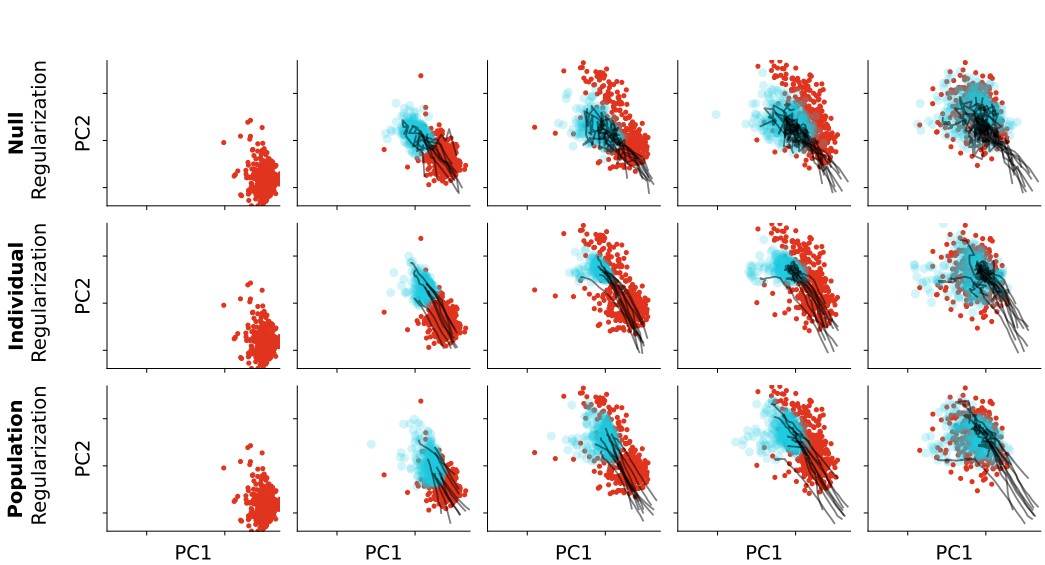

**Figure 12:** Visualization of global dynamics for principle components 1 and 2.

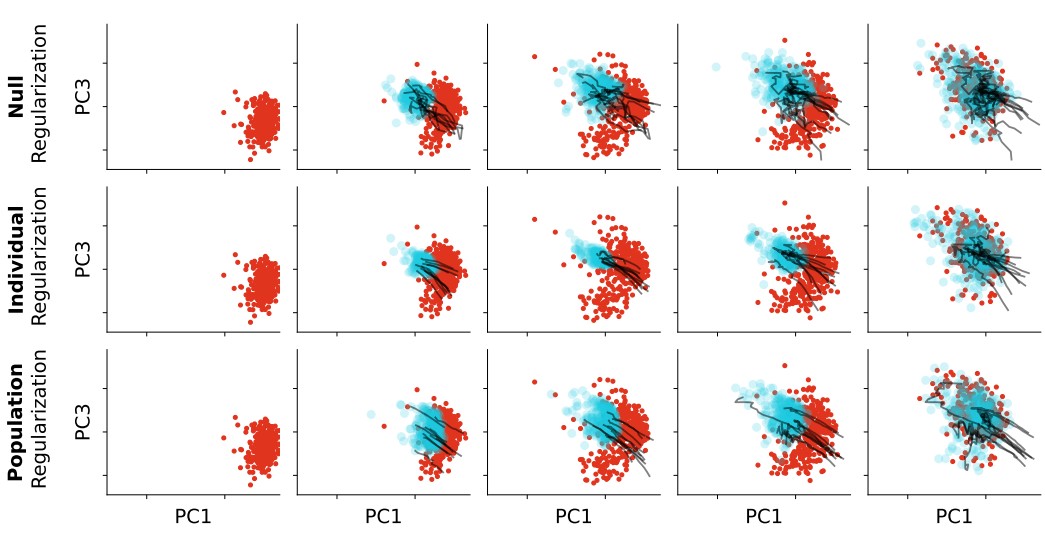

**Figure 13:** Visualization of global dynamics for principle components 1 and 3.

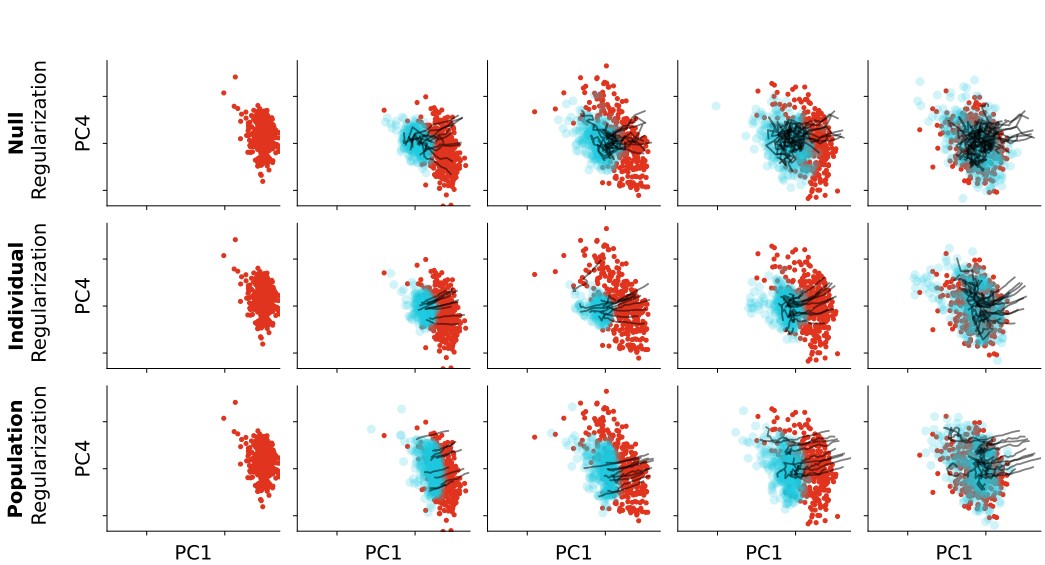

**Figure 14:** Visualization of global dynamics for principle components 1 and 4.

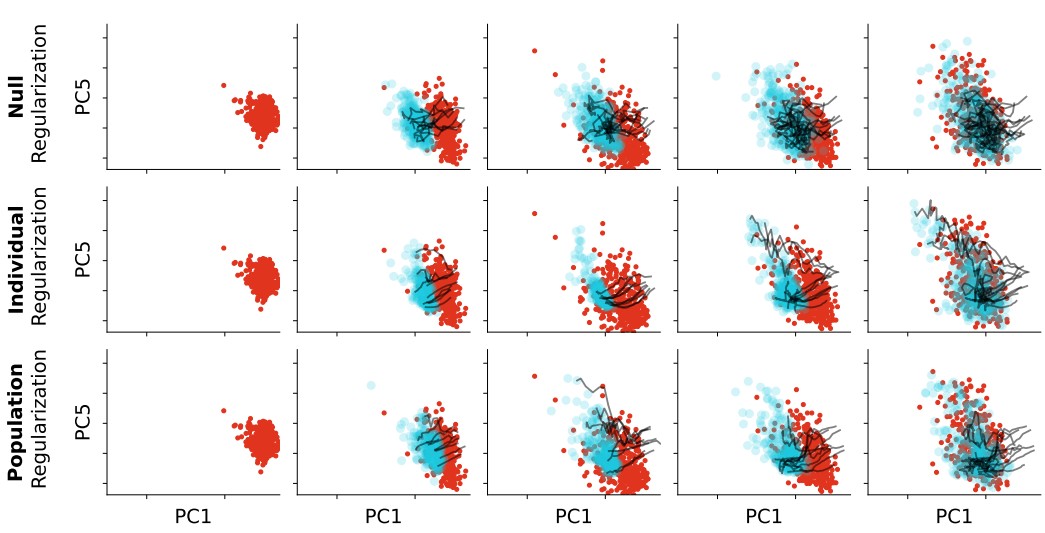

**Figure 15:** Visualization of global dynamics for principle components 1 and 5.

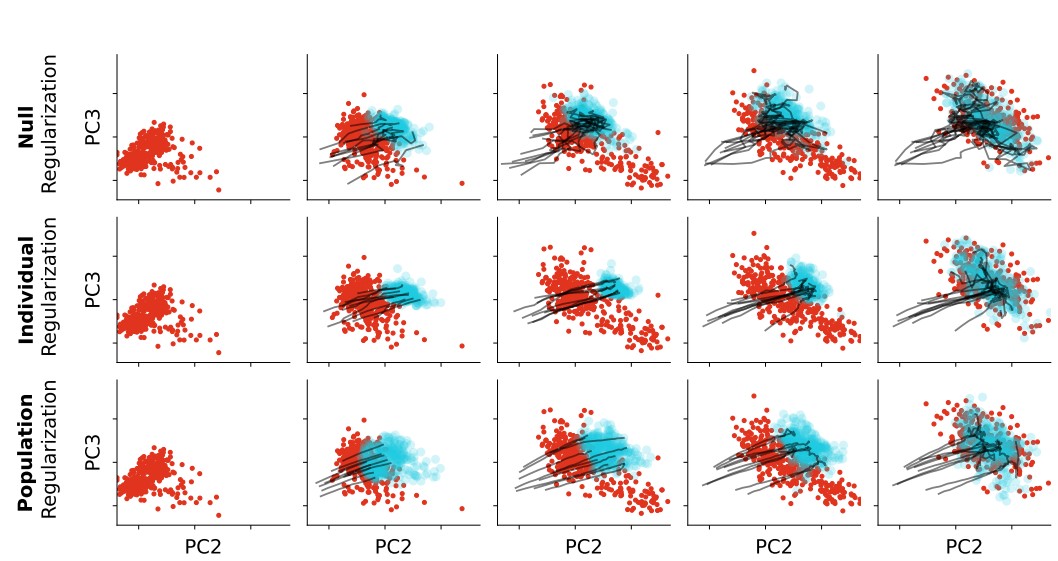

**Figure 16:** Visualization of global dynamics for principle components 2 and 3.

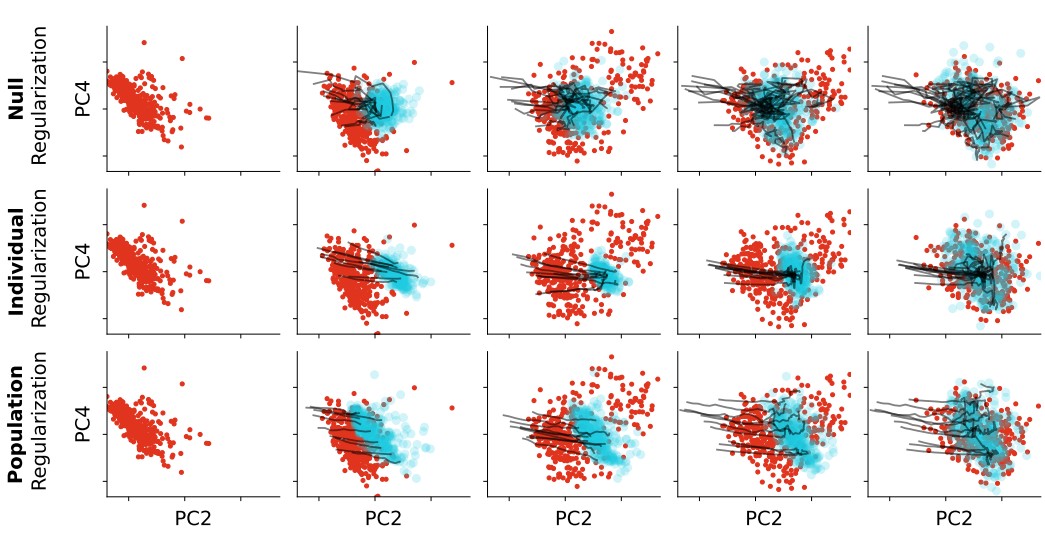

**Figure 17:** Visualization of global dynamics for principle components 2 and 4.

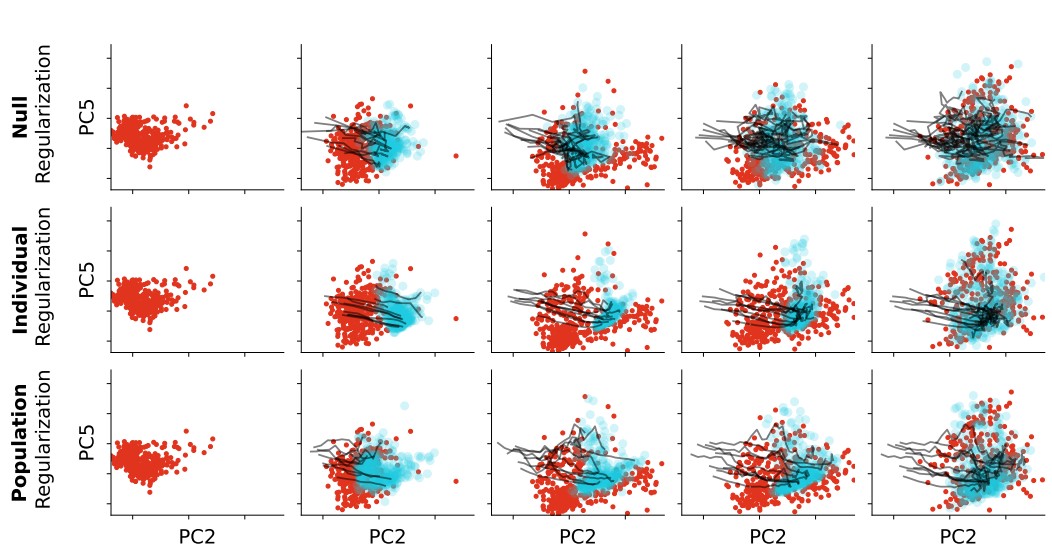

**Figure 18:** Visualization of global dynamics for principle components 2 and 5.

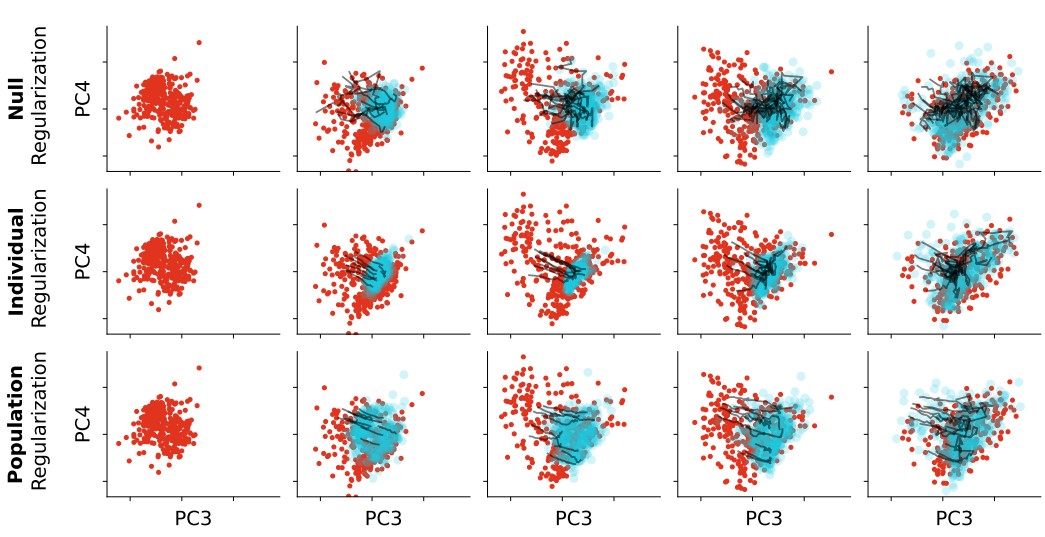

**Figure 19:** Visualization of global dynamics for principle components 3 and 4.

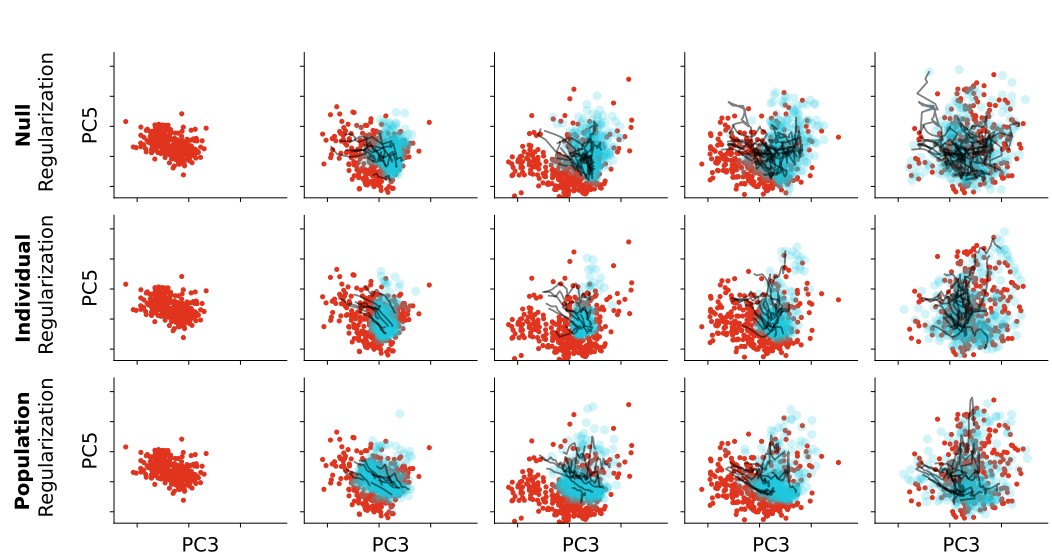

**Figure 20:** Visualization of global dynamics for principle components 3 and 5.

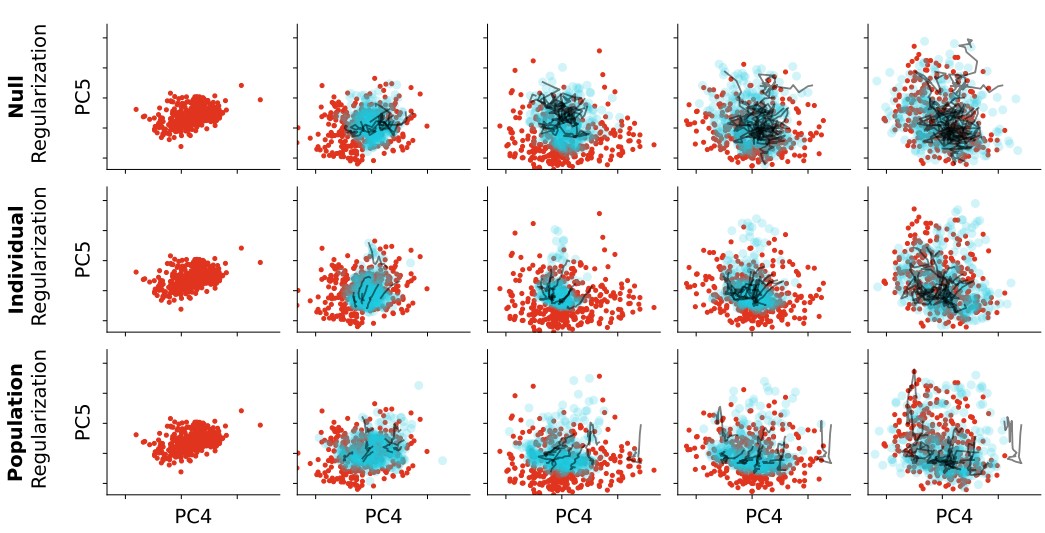

**Figure 21:** Visualization of global dynamics for principle components 4 and 5.

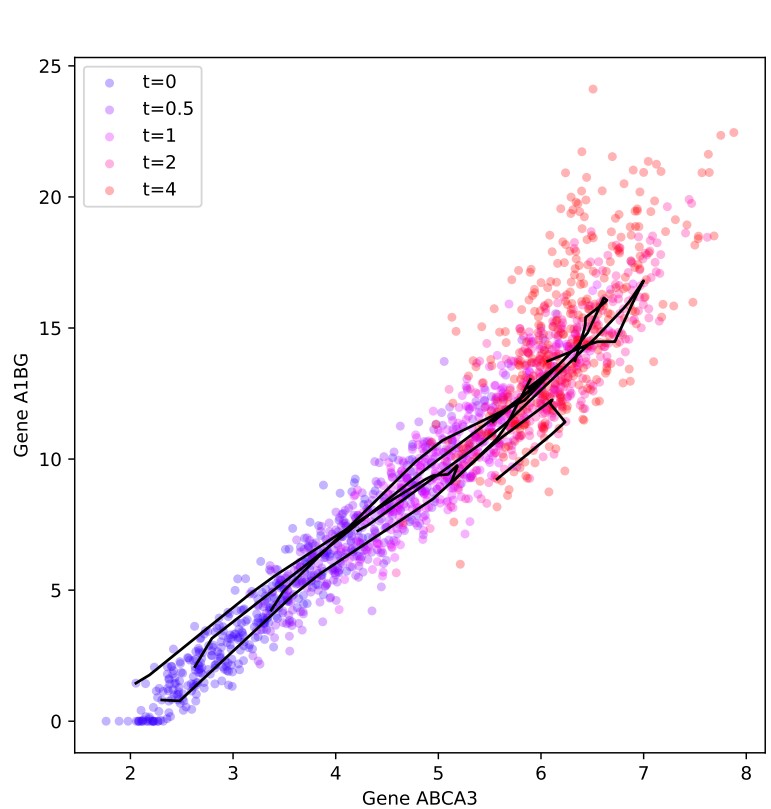

**Figure 22:** Aggregated visualization of generated dynamics for Fig. 2.

