# OpenReview forum: "Correlational Lagrangian Schrodinger Bridge: Learning Dynamics with Population-Level Regularization"
_ICLR.cc/2025/Conference — Submitted to ICLR 2025_

### Official Review · Reviewer_Rv9f · 2024-10-31

**Soundness:** 2
**Presentation:** 3
**Contribution:** 3
**Rating:** 3
**Confidence:** 2

**Summary:**

This paper introduces a population-level regularization adapted for learning heterogeneous biosystems dynamics with Schrödinger Bridge. As data for these systems is collected at each time step but without individual trajectories across time steps (cross sectional data), the Schrödinger bridge is typically extended with a least-action principle to make the optimization procedure tractable.

Lagrangian Schrödinger Bridge is less effective for heterogeneous systems as it assumes homogeneity.
The paper proposes a “principle of least population action” that shifts the conservation laws enforced by LSB from individual to population states. This is extended with a generalized conservation principle that incorporates relationships among data variables and uses the Fokker-Planck equation to derive a tractable estimator. Special cases of this general regularizer with biologically informed constraints are introduced.

The final regularizer proposed is a weighted sum of the original LSB regularizer and some of the new population regularizers.
Numerical experiments are then conducted to show the effectiveness of the proposed regularizer on developmental dynamics of cells.

**Strengths:**

The paper is well written and organized. The authors provide a clear motivation for the proposed method with a relevant example from biology. Limitations of the existing methods are clearly presented

**Weaknesses:**

For $L_{ind}$ the authors mention (l.192-194) that it is not suitable when time series data is not available and justify next introduced $L_{\cdot}$ as solutions that do not require individual trajectories as they get averaged. However, in section 3.4 $L_{ind}$ is used in the final formulation (13). Does this imply that individual trajectory data is actually available? Are the introduced $L_{\cdot}$ effective by themselves when trajectory data is not available?
 This point seems unclear.

I also fail to understand how $L_{pop}$ is an heterogeneous type regularizer. I get that it deals with a population state and not an individual action on single trajectory. However as $\mathbb{E}_{\pi_t}$ is involved in the definition isn't it still applied uniformly across the population?

The paper could benefit from a detailed explanation of how the final proposed formulation (13) helps with the "non convex and constrains wrt network parameters challenge" mentioned.

The paper introduces multiple correlation-based Lagrangians, though only a few are used in the experiments.

**Questions:**

- If $L_{ind}$ is unsuitable without time-series data, why is it included in the final formulation (13)?

- How does $L_{pop}$ act as a heterogeneous regularizer if it’s based on $\mathbb{E}_{\pi_t}$? Doesn’t this imply it still applies uniformly across the population?


- Could the authors provide a more detailed explanation of how the final formulation (Equation 13) addresses the challenge of non-convexity and constraints on network parameters?

- Could the authors clarify why certain Lagrangians were selected over others?

---

> ### Author Response · Authors · 2024-11-19
> **Response to Reviewer Rv9f**
>
> **Q.** For L_ind the authors mention (l.192-194) that it is not suitable when time series data is not available and justify next introduced L as solutions that do not require individual trajectories as they get averaged. However, in section 3.4 L_ind is used in the final formulation (13). Does this imply that individual trajectory data is actually available? Are the introduced L effective by themselves when trajectory data is not available? This point seems unclear.
>
> **A.**
> The reviewer has asked an important clarification question that is very relevant to our problem setup.  We see that a confusion might be related to our presentation and thus clarify as follows. We also noticed many other questions are clarification type questions.  We thank your questions that encourage us to further strengthen our presentation’s clarity and effectiveness.
>
> - Individual trajectory data are not available but regularization is available, for instance, restricting motions of individual particles.
> - $L_{ind}$ denotes individual regularization rather than trajectories. Trajectories are denoted as samples from distribution $p_{t_i, t_{i+1}}$ in formulation (3.1).
> - The final formulation (13) unifies individual $L_{ind}$ and population regularization $L_{pop}$ into a general framework, weighted by hyperparameters. Details on hyperparameter tuning are provided in Appendix I.
>
> -----
>
> **Q.** I also fail to understand how L_pop is an heterogeneous type regularizer. I get that it deals with a population state and not an individual action on single trajectory. However as E_\pi_t is involved in the definition isn't it still applied uniformly across the population?
>
> **A.**
> Again, thank you for asking an important question that needs clarification as follows:
>
> - Heterogeneity in systems suggests that states of individual particles can be very different from each other.
> - Individual regularization is against heterogeneity – The same prior is uniformly applied to all individual particles
>     - The prior is applied to individual trajectories first $prior(x) = h(x)$ and then rolled to all particles $E_{\pi_t} ( |\frac{d}{dt}prior(x)|^2)$.
>
> For the points above we used gene expression as an example in Lines 79-80: “Gene expression covariance is only among a population of cells, while each individual cell has different/heterogeneous gene expressions. Uniformly restraining the states of individuals is thus oblivious to such knowledge”
> - Population regularization better accommodates heterogeneity – The prior is applied on the population state:
>     - The regularization of population states does NOT stress that individual states must behave similarly, even though the population state represents a summary of individuals via expectation (Analogy: a change in average velocity does not necessarily require the same change in individual velocities).
>     - Specifically, the prior is first summarized on the population level $prior(x) = E_{\pi_t} (h(x))$ and then applied on the population-state trajectory $| \frac{d}{dt} prior(x) |^2$.
> - For reasons above we wrote in Lines 84-86, “As the ensemble statistics of individual states, population states (i) respect the diversity (heterogeneity) of individual states, and importantly, (ii) can accommodate domain priors previously not utilized at the population level”.  We also provided an illustration in  Figure 1 left.
>
> -----
>
> **Q.** The paper could benefit from a detailed explanation of how the final proposed formulation (13) helps with the "non convex and constrains wrt network parameters challenge" mentioned.
>
> **A.**
> That is a good suggestion.  We will strengthen the presentation with the following explanations:
>
> - The original non-convex constrained optimization problem (6) is highly challenging to solve.
> - We approximate the problem by reformulating it as an unconstrained optimization problem (12), which can be effectively handled using gradient-based approaches.
> - This reformulation involves replacing the hard constraint with a penalty term in the objective function, effectively introducing it as a soft constraint.
> - Such an approximation is also commonly adopted in other works, e.g. in [1] Section 3.2.
>
> -----
>
> **Q.** The paper introduces multiple correlation-based Lagrangians, though only a few are used in the experiments.
>
> **A.**
> The question might arise from some misunderstanding. We would like to clarify as follows:
>
> - We have used all the three regularizers of covariance velocity (8), acceleration (9), and potential (12).
> - As stated in the previous answer, The final formulation (13) unifies all regularizations into a general framework, weighted by hyperparameters.
> - Details on hyperparameter tuning are provided in Appendix I.
>
> -----
>
> [1] Neural Lagrangian Schrödinger Bridge: Diffusion Modeling for Population Dynamics

---

> > ### Comment · Reviewer_Rv9f · 2024-11-24
> >
> > I want to thank the authors for the detailed reply and for taking the time to clarify their method.
> >
> > - heterogeneous type regularizer: thank you for clarification. On a small comment, Lines 79-80 was confusing to me but I get the point now. Your analogy was instead much clearer. I would consider adding it for people less used to biology concepts.
> >
> > - individual trajectories and regularizers: following your comment, it still isn't clear to me how you would restrict motions of individual particles without having access to the motion of those individual particles. Can you compute L_ind without having access to trajectories samples from  $p_{t_i, t_{i+1}}$?
> >
> > As no other reviewer mentioned this, maybe I am missing something. But the way the problem is introduced in  l.192-194 it seemed to me that "time-series data for individuals may not be available" was a problem for estimating L_ind. If not, what is the "challenge of time-series data for individuals may not be available" that this paper adresses?
> >
> > - About L_corr-std, L_corr-std-linproj, are they used anywhere?

---

> > > ### Author Response · Authors · 2024-11-25
> > > **Response to Reviewer Rv9f**
> > >
> > > **Q.** heterogeneous type regularizer: thank you for clarification. On a small comment, Lines 79-80 was confusing to me but I get the point now. Your analogy was instead much clearer. I would consider adding it for people less used to biology concepts.
> > >
> > > **A.** Thank you for your comment. We will add this part of discussion in revision for better clarity.
> > >
> > > ------------
> > >
> > > **Q.** individual trajectories and regularizers: following your comment, it still isn't clear to me how you would restrict motions of individual particles without having access to the motion of those individual particles. Can you compute L_ind without having access to trajectories samples from p_{ti, ti+1}?
> > >
> > > **A.**
> > > - The regularization can be computed without access to observation.
> > > - That is because the regularization is computed on the generated samples from models, so that it regulates the behavior of models.
> > > - For the specific restriction on individual motions, a simple implementation is to regularize the individual velocities of $\mathbb{E}_{\pi_t} || v_t(x_t) ||^2$, where $v_t(\cdot)$ is the parametrized drift in diffusion models.
> > >
> > > ------------
> > >
> > > **Q.** As no other reviewer mentioned this, maybe I am missing something. But the way the problem is introduced in l.192-194 it seemed to me that "time-series data for individuals may not be available" was a problem for estimating L_ind. If not, what is the "challenge of time-series data for individuals may not be available" that this paper adresses?
> > >
> > > **A.**
> > > - Clarification: "time-series data for individuals may not be available" is NOT a problem for estimating $L_{ind}$.
> > > - The challenge is to seek for a more effective regularization, when the trajectory data are not available for direct supervision.
> > > - The paper proposes a much more effective regularization at the population level, that that at the individual level, to help models better learn the dynamics.
> > >
> > > ------------
> > >
> > > **Q.** About L_corr-std, L_corr-std-linproj, are they used anywhere?
> > >
> > > **A.**
> > > - The use of the population within a projected space depends on the problem setting, whether the prior knowledge lies in a certain known projected space.
> > > - In the single-cell experiment in Sec. 4.1, we do use the standardized regularization in a linear projected space $L_{corr-std-linproj}$ since the dynamics learning is in the PCA space while the prior of genetic stability is in the gene expression space (before PCA).
> > > - Such a formulation provides much greater flexibility for our method to be adopted into other problems, especially when the data are in a different space from the prior
> > >
> > > ------------
> > >
> > > We again thank you for the comments that have helped our revision. We would be thrilled to have more such exciting and constructive discussions with you.  If you have already found our responses satisfactory, we humbly remind you of a fitting update of the rating. Thank you!

---

> > > > ### Comment · Reviewer_Rv9f · 2024-11-25
> > > >
> > > > Thank you again for taking the time to clarify your method.
> > > >
> > > > Then the challenge of not having access to trajectory data is not clear to me. In my initial review I thought that motivations were clear but I do not longer think this is the case.
> > > >
> > > > As it stands I'll keep my score and my rather low confiance.

---

> > > > > ### Author Response · Authors · 2024-11-25
> > > > > **We are eager to help you understand the full story**
> > > > >
> > > > > Thank you for your follow up! We are eager to help you understand the full story of the paper as much as possible. So please do not hesitate to keep commenting on us for your remaining questions!
> > > > >
> > > > > **Q.** Why not having trajectory data is a challenge?
> > > > >
> > > > > **A.** Short answer: Since it is what we want to predict from models, sort of serving **“labels”** that can provide direct supervision in machine learning! Without it, we thus need some **alternative objectives** to supervise model which is proposed and termed population regularization in this paper.
> > > > >
> > > > > Long answer in points:
> > > > > * Why is it a challenge? Trajectory data can provide the most direct supervision to diffusion generative models as “labels”.
> > > > >   * Without it, we need to design alternative objectives to supervise the model.
> > > > >   * Such a challenge is also tackled in many other papers such as [1,2].
> > > > > * How do existing approaches tackle it? Existing approaches design alternative objectives (regularization) at the individual level [1,2].
> > > > >   * The objective is based on prior knowledge, e.g. restricting the motion of individual particles.
> > > > >   * The prior is uniformly applied to all individual particles
> > > > > * How do we advance on the existing approaches? We design a novel and more effective objective (regularization) at the population level (see Fig. 1 left for their point-to-point difference). The new regularization is valuable in:
> > > > >   * Brand-new approach: The population regularization is completely new and has never been explored in this problem setting
> > > > >   * Conceptually clean innovation in the formulation: Our approach in formulation is a clear shift from regularizing individual behaviors to population behavior, by switching two operators (Opt. (4) vs (5)) to facilitate heterogeneity in the system (see Sec. 3 for details).
> > > > >   * New theoretical results: The new formulation inspires new theoretical results in proposition 1.
> > > > >   * Super effective in the designated scenarios: We show very strong empirical evidence of the approach in the heterogeneous systems of single-cell data in experiments
> > > > >
> > > > > [1] Neural Lagrangian Schrödinger Bridge: Diffusion Modeling for Population Dynamics
> > > > > [2] TrajectoryNet: A Dynamic Optimal Transport Network for Modeling Cellular Dynamics

---

### Official Review · Reviewer_wXGs · 2024-11-01

**Soundness:** 3
**Presentation:** 3
**Contribution:** 3
**Rating:** 8
**Confidence:** 3

**Summary:**

Interesting method applying a population-level regularization to schrodinger bridge parameterized by neural nets for the drift and diffusion terms. They motivate their problem with the relevant biological setting of predicting the influence of drugs on cell state, where cell state can only be measured once in a destructive process. The paper dives into details pertinent to this biological problem such as how to apply the method when in a latent space imposed e.g. by PCA. My main critiques are in the presentation of the method which makes checking the math difficult at times. This concern about presentation and validity of the method is mainly due to the lack of convincing empirical evidence. The proposed method seems to only occasionally outperform in the unconditional evaluation. I would feel more confident in this paper with better presentation and clarity.

**Strengths:**

- The paper introduces a new way to regularize Lagrangian schrodinger bridges.
- This regularization can potentially improve population-level dynamics of single-cell analyses.
- There are different ways to use the correlation matrices, which provides flexibility and allows users to impose their own priors based on biological knowledge. I'd really like more discussion how to use this, if possible. Would make the paper much better.
- The authors provide extensive proofs as to how the regularization is implemented numerically.
- Extensive ablation studies and comparisons, which is great.

**Weaknesses:**

- There are some grammatical errors, such as "overwrote" in line 175, "we have" in line 319. "We achieves..." in line 299.
- A lot of variables are introduced in passing. I like the table that reminds what the difference between sub/superscript is and think this could be extended to definitions of variables $s, d, m, k, \mathcal{M}, \tilde{\mathcal{M}}, $
- No code to validate.
- The presentation of the results is a little confusing. Figure 2 uses PCs for the gene expression trajectories but it would make more sense to present the way Tong et al. 2023 did in this [paper](https://arxiv.org/pdf/2307.03672) in Figure 3. This would make the value of the method more immediate to computational biologists. The current method showing PCs of individual genes is unintelligible. This change would greatly help clarity for a more diverse audience. Additionally, I'd recommend using this compressed representation in the appendix. Just showing pairwise plots of seemingly random genes and expressed vs. predicted is not very helpful, unless I'm missing something related to the covariance that this method proposes. If there's something there that makes those plots more interpretable, you should highlight that in the main paper.
- In proofs for the k $\geq$ 2 case, I would prefer if you introduce the operators before just laying everything out in 14. It's very taxing to have to crossvalidate the equation with its parts and doesn't make me confident in the validity of the proof. Additionally, a sketch of proof saying how it works in summary would greatly help. Still, I have concerns related to the previous questions that don't allow me to fully check this proof, e.g. what is the purpose of using the product operator in this case.
- it's hard for me to validate the proofs missing interpretation about what the product is that defines the correlation structure in the first place.

**Questions:**

- Do you use the product over the probability densities in equation 6 because of the assumption of independence? This wasn't explained well and this lack of definition makes the whole paper hard to follow since it is used extensively.
- the velocity and acceleration of the covariance are interesting ideas to constrain. It's not clear to me how you actually do this from the paper. More clear exposition would help here. As well, why not investigate the hyperparameter sweep of these variables?
- You prove the light tailedness in the appendix but only mention its importance in the main paper. It would be helpful to have some more context about this and why it's cancelled from the proof in the paper.
- What's the computational complexity of this method? How are you sampling? You don't need a section on these but please mention a high-level complexity and state how you sample trajectories.
- In proof for $k \geq 2$ in the appendix, what is $\mathcal{S}'$ and $\Upsilon'$? in equation 14.

---

> ### Author Response · Authors · 2024-11-19
> **Response to Reviewer wXGs (1)**
>
> **Q.** There are some grammatical errors, such as "overwrote" in line 175, "we have" in line 319. "We achieves..." in line 299.
>
> **A.** We thank the reviewer for pointing them out and we have corrected them in the updated pdf.
>
> --------
>
> **Q.** A lot of variables are introduced in passing. I like the table that reminds what the difference between sub/superscript is and think this could be extended to definitions of variables xxx.
>
> **A.** We sincerely thank the reviewer for the comments. We have provided an extended version of the notation Table 1 in the updated pdf.
>
> -----------
>
> **Q.** No code to validate.
>
> **A.** We have uploaded our anonymous code in the supplementary materials.
>
> -----------
>
> **Q.** The presentation of the results is a little confusing. Figure 2 uses PCs for the gene expression trajectories but it would make more sense to present the way Tong et al. 2023 did in this paper in Figure 3. This would make the value of the method more immediate to computational biologists. The current method showing PCs of individual genes is unintelligible. This change would greatly help clarity for a more diverse audience. Additionally, I'd recommend using this compressed representation in the appendix. Just showing pairwise plots of seemingly random genes and expressed vs. predicted is not very helpful, unless I'm missing something related to the covariance that this method proposes. If there's something there that makes those plots more interpretable, you should highlight that in the main paper.
>
> **A.** We appreciate your great comment on improving the visualization.
>
> - We have added the new visualization as suggested in Figure 22.
> - Some clarification on gene selection: They are not randomly selected. The selected visualized gene pairs are the most correlated ones in the dataset, providing a “local” view of the dynamics.
> - For PC: Additionally, we offer a “global” view by visualizing the dynamics in the PC space, where the PC is computed based on all genes.
> - We will follow your suggestion to add more visualizations in the revision.
>
> --------------
>
> **Q.** Do you use the product over the probability densities in equation 6 because of the assumption of independence? This wasn't explained well and this lack of definition makes the whole paper hard to follow since it is used extensively. it's hard for me to validate the proofs missing interpretation about what the product is that defines the correlation structure in the first place.
>
> **A.**
> The question might arise from some misunderstanding.
>
> - We didn’t make assumptions on independence and conduct factorization on joint distributions.
> - Product is operated across variables within the same time stamp $t$, and the expectation is computed on the joint distribution $\pi_t$.
> - Such a formulation is to describe the correlation across variables within the same time stamp.

---

> > ### Author Response · Authors · 2024-11-19
> > **Response to Reviewer wXGs (2)**
> >
> > **Q.** In proofs for the k>=2 case, I would prefer if you introduce the operators before just laying everything out in 14. It's very taxing to have to crossvalidate the equation with its parts and doesn't make me confident in the validity of the proof. Additionally, a sketch of proof saying how it works in summary would greatly help. Still, I have concerns related to the previous questions that don't allow me to fully check this proof, e.g. what is the purpose of using the product operator in this case. In proof for k>=2 in the appendix, what is S’ and \Upsilon’? in equation 14.
> >
> > **A.** We sincerely appreciate your suggestions. We clarify the notations and proof sketch as follows:
> >
> > - We also have made a notation table for the proof in Appendix A.
> > - In short, to express the analytical expression in an iterative manner from $k=j-1$ to $k=j$, we abstract the formulation with a new set of notations including $\mathcal{S}$ and $\Upsilon$.
> > - We briefly discuss the proof sketch for the analytical expression of $\frac{d^k}{dt^k} E_{\pi_t} [h(x_t)]$ as follows.
> >     - Preliminary: Fokker-planck equation is a equation to establish connections between time evolution (derivative) of the pdf $\pi_t$ and force field $v_t, \Sigma_t$ as $\frac{d}{dt} \pi_t(x) = FKP( \pi_t(x), v_t(x), \Sigma_t(x) )$ (please refer Equation (2) for the specific form of $FKP(\cdot)$).
> >     - For $k=1$:
> >         - We can eliminate the time derivative by applying the Fokker-planck equation.
> >         - We write the expectation in the integration form as $\frac{d}{dt} E_{\pi_t} [h(x_t)] = \int h(x) \frac{d}{dt} (\pi_t(x)) dx$.
> >         - We then can apply the Fokker-planck equation to eliminate the derivative operation by replacing it with force field (generative models) as $\int h(x) FKP( \pi_t(x), v_t(x), \Sigma_t(x) ) dx$.
> >         - The rest is to simplify the above expression, by integrating terms and eliminating terms by applying integration by parts.
> >         - The final expression is of the form $E_{\pi_t} [h_1(x_t, \pi_t(x), v_t(x))]$.
> >     - For $k>=2$:
> >         - There is an iterative derivation from $k=j-1$ to $k=j$.
> >         - Since for $k=1$, we can derive the analytical expression containing no time derivative. For $k=2$, it is essentially the derivative of the $k=1$ term.
> >         - That is $\frac{d}{dt} E_{\pi_t} [h_1(x_t, \pi_t(x), v_t(x))]$.
> >         - We notice that it is of the same form as $\frac{d}{dt} E_{\pi_t} [h(x_t)]$ as in $k=1$, thus we can conduct the similar derivation to replace the time derivative with the force field. The final expression must be of the form $E_{\pi_t} [h_2(x_t, \pi_t(x), v_t(x))]$.
> >         - For $k=j$, similarly, we can iteratively derive the analytical expression from the $k=j-1$ term, which can be generalized to all $k$.
> >
> > --------------
> >
> > **Q.** the velocity and acceleration of the covariance are interesting ideas to constrain. It's not clear to me how you actually do this from the paper. More clear exposition would help here. As well, why not investigate the hyperparameter sweep of these variables?
> >
> > **A.**
> > - Proposition 1 (7) provides an analytical expression for implementing regularization.
> > - For the specific form to compute the covariance velocity, please refer to Equation (8).
> > - For the specific form to compute the covariance acceleration, please refer to Equation (9).
> > - We do provide the details on hyperparameter tuning for all regularization terms in Appendix I.
> >
> > -----------
> >
> > **Q.** You prove the light tailedness in the appendix but only mention its importance in the main paper. It would be helpful to have some more context about this and why it's cancelled from the proof in the paper.
> >
> > **A.**
> > - The assumption of light-tailedness can be easily satisfied through proper parameterization of the model [1].
> > - For instance, a Gaussian-like parameterized probability density function [1] actually features an exponentially light tail.
> > - The cancellation is then achieved by using both Gauss’s theorem [2] and the assumption itself.
> >
> > -----------
> >
> > **Q.** What's the computational complexity of this method? How are you sampling? You don't need a section on these but please mention a high-level complexity and state how you sample trajectories.
> >
> > **A.**
> > - The computational complexity depends on the specific form of the regularizer.
> > - For covariance velocity (8), it is $O(NTD^2)$, where $N$ is the number of sampled trajectories, $T$ is the number of sampled time point, and $D$ is the number of variables (dimensionality).
> > - We do the sampling by conditioning on the observed $x_0$, and then apply the learned force field $v_t, \Sigma_t$ to sample trajectories (a SDE process). In implementation, we rely on the torchsde library [3].
> >
> > -----------
> >
> > [1] Score-Based Generative Modeling through Stochastic Differential Equations
> >
> > [2] Gauss’s theorem in general relativity
> >
> > [3] Scalable gradients for stochastic differential equations

---

> > > ### Comment · Reviewer_wXGs · 2024-11-24
> > > **Official comment by reviewer wXGs**
> > >
> > > I think the authors for providing clarifications. The derivation of the method and added details make the validity of the method more clear. I think this is an interesting and promising approach to problems with destructive sampling e.g. single cell data.

---

> > > > ### Author Response · Authors · 2024-11-25
> > > > **Thank you for confirming the validity of our approach/analysis and adjusting rating 5-->8**
> > > >
> > > > We sincerely thank the reviewer for the time and effort in evaluating the validity of our approach and analysis. We have thoroughly proofread the content multiple times and are glad that our efforts successfully conveyed the intended message through the paper and our communication. Your adjustment of the rating from 5 to 8 is greatly encouraging to us. Thank you!

---

### Official Review · Reviewer_38fi · 2024-11-02

**Soundness:** 3
**Presentation:** 2
**Contribution:** 2
**Rating:** 5
**Confidence:** 2

**Summary:**

The paper introduces an approach to the modelling of dynamics of populations from which only cross-sectional observations are available. It is suggested that in the absence of association between observations and members of the population that it is more sensible to conduct modelling at the population level.

Within the Schroedinger bridge framework, i.e. with initial an final distributions known and fixed, the problem of determining the trajectories from observations is viewed as an optimal transport problem. A major element of the current work's contribution is to consider regularisation of this process at the level of the population rather than of the trajectories of individuals.

Numerical results are presented for a cell-sequencing setting which demonstrate the good performance of the proposed approach in that setting.

**Strengths:**

The paper clearly explains what is does and its consequences in terms of the resulting optimization problem.

Sections 3.2-3.3 provided details which will be valuable to anyone interested in applying these techniques in various contexts; and go beyond the usual abstract presentation of methodology giving nice details of how to implement the proposed approach to achieve particular outcomes.

Some consideration is given to implementation of the method: in addition to formulating the problem at the population level some space is dedicated to showing how one can actually perform learning using this revised objective.

The numerical results presented (e.g. in Tables 2 and 3) seemingly demonstrate good performance of the proposed method in the explored example setting.

**Weaknesses:**

One thing that wasn't completely clear to me (this isn't an area I am particularly familiar with) was what the main reasons for the audience of this conference to be interested in this work are. It seems like a nice contribution to learning for particular settings, in which a population-level view is reasonable but beyond the rather specific numerical examples in a cellular context it wasn't clear to a non-expert what those contexts might be. Are the authors able to explain this a little more explicitly?

The underlying motivations wasn't articulated as clearly as it might have been in my view -- beyond leading to a more tractable optimisation problem, when/why would one choose to view the problem of inferring dynamics in this population-level way? (The fact the individuals may behave heterogeneously and one may not be able easily to relate observations to individuals does not in and of itself seem to mean that a population-level regularisation is the natural way to proceed.)

The numerical evaluation appears to be limited to a single setting which is quite difficult for a non-expert to interpret, although there are numerical quantifications provide which appear to demonstrate good performance in that setting. It might be more persuasive if there were numerical results illustrating the proposed approach in several settings illustrating good performance across a wider range of scenarios particularly if the paper's contribution is to be viewed as general methodology rather than an application to the setting of the numerical example. There are some additional experiments in  Appendix I, but these are primarily very closely related to the original application with on exception on "opinion depolarisation" but the presentation here is so condensed that it is not completely obvious what is shown.



Minor details:
* There are places where more proofreading would help, e.g. in the references line 544 has `Schr\"odinger'

**Questions:**

Are there general settings in which the authors envisage the proposed approach being effective? Are there contexts of broad interest to the ICLR community which involve cross-sectional observations in which population-level regularization would be reasonable?

It seems that moving from individual level to population level regularization is a philosophical change in position rather than a choice to be made for computational reasons. To the non-expert, it feels like a reasonable shift where it is reasonable to suppose that there is some (implicit or explicit) population-level control of the dynamics -- perhaps in the context of hives, or coordinated systems -- but which is unlikely to model well the behaviour of a population of autonomous individuals. I was surprised not to see anything along these lines discussed in the manuscript. Are the authors able to comment on this?

Figure 1: How should the reader interpret this and the many similar figures in the paper? This isn't an area with which I am familiar and I found it difficult to understand what I was supposed to deduce from this figure.

Line 236: A very minor point, but can you clarify what this multiset is? I couldn't immediately understand what necessitated the use of a multiset rather than a regular set here.

---

> ### Author Response · Authors · 2024-11-19
> **Response to Reviewer 38fi (1)**
>
> **Q.** One thing that wasn't completely clear to me (this isn't an area I am particularly familiar with) was what the main reasons for the audience of this conference to be interested in this work are. It seems like a nice contribution to learning for particular settings, in which a population-level view is reasonable but beyond the rather specific numerical examples in a cellular context it wasn't clear to a non-expert what those contexts might be. Are the authors able to explain this a little more explicitly?
> Are there general settings in which the authors envisage the proposed approach being effective? Are there contexts of broad interest to the ICLR community which involve cross-sectional observations in which population-level regularization would be reasonable?
>
> **A.** We are grateful to the reviewer for the useful comments. We would like to clarify for this question (and also other questions) that our focus on methodology and applications is general rather than restricted to a specific domain.
>
> - We focus on the task of learning population dynamics from cross-sectional data, where individual trajectories cannot be tracked, and only samples of different individuals are available at various time points. In fact, the individuals across different time points are different (thus cannot be tracked) but drawn from the same time-dependent distribution.
> - This study is motivated by a biological problem involving single-cell data, where measurements are destructive, making it impossible to observe the same cell at multiple time points.
> - However, this setting is not limited to the biological domain; it extends to societal applications as well. For example, it can be used to model the evolution of population statistics [1] or even public opinion [2], where it is not guaranteed that data for the same individual will be available across different time periods.
> - Our experiments are also not confined to biological datasets; for instance, we explore opinion depolarization in Appendix I.
>
> ----------------
>
> **Q.** The underlying motivations wasn't articulated as clearly as it might have been in my view -- beyond leading to a more tractable optimisation problem, when/why would one choose to view the problem of inferring dynamics in this population-level way? (The fact the individuals may behave heterogeneously and one may not be able easily to relate observations to individuals does not in and of itself seem to mean that a population-level regularisation is the natural way to proceed.)
>
> **A.**
> - Clarification: Our end goal is still to predict the dynamics of individuals while we implement the regularization at the population level.
> - Specifically, we optimize the generative model of the form of $v_t(x), \Sigma_t(x)$ which is responsible for individual evolution.
> - The motivation to implement regularization at the population level is to respect the heterogeneity of individual behavior, where individual states could be very different in certain systems (e.g. cells differentiate into diverse cell types, or people’s opinions diverge into various focal points), the population state is a more robust metric to regularize.
>
> -----------
>
> **Q.** The numerical evaluation appears to be limited to a single setting which is quite difficult for a non-expert to interpret, although there are numerical quantifications provide which appear to demonstrate good performance in that setting. It might be more persuasive if there were numerical results illustrating the proposed approach in several settings illustrating good performance across a wider range of scenarios particularly if the paper's contribution is to be viewed as general methodology rather than an application to the setting of the numerical example. There are some additional experiments in Appendix I, but these are primarily very closely related to the original application with on exception on "opinion depolarisation" but the presentation here is so condensed that it is not completely obvious what is shown.
>
> **A.**
> - Our experiments encompass the biological field of single-cell analysis, the sociological study of opinion polarization, and the creation of synthetic datasets.
> - We have added more background discussions for the opinion polarization experiment in Appendix I.
>
> ----------
>
> **Q.** There are places where more proofreading would help, e.g. in the references line 544 has `Schr"odinger'
>
> **A.** We have updated the pdf to correct the typo.

---

> > ### Author Response · Authors · 2024-11-19
> > **Response to Reviewer 38fi (2)**
> >
> > **Q.** It seems that moving from individual level to population level regularization is a philosophical change in position rather than a choice to be made for computational reasons. To the non-expert, it feels like a reasonable shift where it is reasonable to suppose that there is some (implicit or explicit) population-level control of the dynamics -- perhaps in the context of hives, or coordinated systems -- but which is unlikely to model well the behaviour of a population of autonomous individuals. I was surprised not to see anything along these lines discussed in the manuscript. Are the authors able to comment on this?
> >
> > **A.**
> > - We include more thorough related work discussion in Appendix C
> > - In short, the population dynamics is studied before, while the regularization at the population level is never explored, in addition to being incorporated into diffusion generative models.
> > - Our work is the first to design, instantiate and implement population regularization together with diffusion generative models for a more effective dynamics modeling.
> >
> > ----------
> >
> > **Q.** Figure 1: How should the reader interpret this and the many similar figures in the paper? This isn't an area with which I am familiar and I found it difficult to understand what I was supposed to deduce from this figure.
> >
> > **A.**
> > - Different figures visualize the learned trajectories of different gene pairs and principal component pairs.
> > - Red color is for the ground truth and cyan color is for the generated trajectories.
> > - The figure shows the population regularization facilitates a more realistic trajectory inference compared to individual regularization.
> >
> > --------------
> >
> > **Q.** Line 236: A very minor point, but can you clarify what this multiset is? I couldn't immediately understand what necessitated the use of a multiset rather than a regular set here.
> >
> > **A.**
> > - Multiset: A generalized data structure extends from a regular set, which allows for repetitive elements.
> > - This is used to specify the variable index in the multivariate correlation prior described in (6).
> > - This allows for the inclusion of self-interaction when computing multivariate correlation. For example, the multiset for the variance (2nd-order) of the jth variable itself is {(j, 2)}, indicating that the index j appears twice in the multiset.
> >
> > ------------
> >
> > [1] Modelling Population Dynamics
> >
> > [2] Polarization in Geometric Opinion Dynamics

---

> > > ### Comment · Reviewer_38fi · 2024-11-23
> > > **Thanks for the detailed responses**
> > >
> > > I appreciate the time taken to address my comments.
> > >
> > > I have reduced my confidence to 2 because I am finding it rather difficult to assess the extent to which this particular problem is important / relevant to the wider community. Clearly the authors believe it is, but it seems to me rather specialised.
> > >
> > > The responses while in some sense comprehensive haven't completely address some of my underlying concerns:
> > > "Clarification: Our end goal is still to predict the dynamics of individuals while we implement the regularization at the population level."
> > > That much was clear, but *why* do you want to do that. It feels, as I said, like a qualitatively different thing to seek to do. The explanation "The motivation to implement regularization at the population level is to respect the heterogeneity of individual behavior, where individual states could be very different in certain systems (e.g. cells differentiate into diverse cell types, or people’s opinions diverge into various focal points), the population state is a more robust metric to regularize." seems rather difficult to interpret -- regularisation at the level of individuals can allow for heterogeneity; regularization at the population level seems a categorically different operation which would be appropriate in different circumstances.
> > >
> > > "The figure shows the population regularization facilitates a more realistic trajectory inference compared to individual regularization."  is a reader not expert in this particular problem realistically going to be able to deduce that from this figure and the information provided? I've looked at it again and it still seems far from obvious.
> > >
> > > "Multiset: A generalized data structure extends from a regular set, which allows for repetitive elements." Thanks for the clarification; I was aware what a multiset was but the way it was written with multiplicity included in the elements of the set it seemed to be encoded as a regular set over an extended space. It wasn't an important point, anyway.

---

> > > > ### Author Response · Authors · 2024-11-25
> > > > **Response to Reviewer 38fi**
> > > >
> > > > **Q.** I am finding it rather difficult to assess the extent to which this particular problem is important / relevant to the wider community
> > > >
> > > > **A.**
> > > > - The focal problem is significant and of interest to the broader community:
> > > >   - it is a **general challenge** of estimating the temporal evolution of a system,
> > > >   - constrained by a **practical restriction** of lacking access to individual trajectories (termed cross-sectional observations in the paper).
> > > > - The problem setting is not confined to the biological domain but extends to **societal applications** as well.
> > > >   - For instance, it can be applied to model the evolution of population statistics [1] or even public opinion [2].
> > > >   - The evolution of population statistics and opinions to be modeled, is highly valuable for understanding the social environment and holds significant potential in shaping public policies, guiding resource allocation, and informing decision-making in various societal domains.
> > > >   - This restriction is also prevalent in these problems: there is no guarantee of recording the trajectories of the same individuals over the period due to various reasons (e.g., lacking access to individual trajectory data).
> > > > - Our method can be applied to these problems, too. For instance, we explore opinion depolarization in Appendix I.
> > > >
> > > > ---------
> > > >
> > > > **Q.** Why to implement the regularization at the population level.
> > > >
> > > > **A.**
> > > > - Brand-new approach:
> > > >   - The population regularization is completely new and has never been explored in this problem setting
> > > >   - the investigation is not only valuable but inspiring.
> > > > - Conceptually clean innovation in the formulation:
> > > >   - Our approach in formulation is a clear shift from regularizing individual behaviors to population behavior, by switching two operators (Opt. (4) vs (5)).
> > > >   - The effect of the new formulation will exert less stress on individuals to behave similarly, allowing their flexibility to act differently (heterogeneously).
> > > >   - The consequence of this effect is that it would facilitate dynamic learning in systems where such heterogeneity is present.
> > > > - New theoretical results:
> > > >   - The new formulation inspires new theoretical results in proposition 1.
> > > > - Super effective in the designated scenarios, though not necessarily universally good:
> > > >   - We show very strong empirical evidence of the approach in the heterogeneous systems of single-cell data in experiments.
> > > >
> > > > ----------
> > > >
> > > > **Q.** "The figure shows the population regularization facilitates a more realistic trajectory inference compared to individual regularization." is a reader not expert in this particular problem realistically going to be able to deduce that from this figure and the information provided? I've looked at it again and it still seems far from obvious.
> > > >
> > > > **A.**
> > > > - The figure demonstrates the effectiveness of our population regularization.
> > > > - The statement is drawn by checking the matching between cyan dots and red dots:
> > > >   - The cyan dots are generated population from models,
> > > >   - and red dots are observed population
> > > > - Different rows of figures represent the results of different approaches.
> > > >
> > > > ----------
> > > >
> > > > **Q.** "Multiset: A generalized data structure extends from a regular set, which allows for repetitive elements." Thanks for the clarification; I was aware what a multiset was but the way it was written with multiplicity included in the elements of the set it seemed to be encoded as a regular set over an extended space. It wasn't an important point, anyway.
> > > >
> > > > **A.** Thank you for your helpful comment. We will add more clarification in revision.
> > > >
> > > >
> > > > [1] Modelling Population Dynamics  [2] Polarization in Geometric Opinion Dynamics
> > > >
> > > > ----------
> > > >
> > > > We again thank you for the comments that have helped our revision. We would be thrilled to have more such exciting and constructive discussions with you.  If you have already found our responses satisfactory, we humbly remind you of a fitting update of the rating. Thank you!

---

### Official Review · Reviewer_9zdy · 2024-11-08

**Soundness:** 3
**Presentation:** 3
**Contribution:** 3
**Rating:** 6
**Confidence:** 2

**Summary:**

This paper looks at the question of modeling population dynamics. The setting is one where the data is sampled cross-sectionally, and the goal is to bridge these cross-sectional snapshots. For one, these snapshots may even lack observations on the same population due to the way the data is acquired, for example, in immunomics-type of applications. Traditional approaches to such problems attempt to utilize methods that bridge the distributions together across time with, for example, techniques such as optimal transport. This paper proposes a way to do this bridging in a more principled way, through utilizing regularization at the population level. The key idea here appears to be, effectively, restraining population covariance kinetics-related quantities. The proposed method is compared to existing ones, with promising results.

**Strengths:**

The paper is nicely written and well-illustrated. The problem presented by the authors is well-motivated with real grounding in applications. The results seem promising and worth investigating further.

**Weaknesses:**

The authors should perhaps give some idea of the computational costs of their methods. I am also not sure why in Figure 2 the cyan dots (in the lower panel) have much smaller variance than the red ones.

**Questions:**

What happens if the covariance structure undergoes a change over time? Can the method adapt to this scenario? Have you considered things like normalizing flows as a comparison?

---

> ### Author Response · Authors · 2024-11-19
> **Response to Reviewer 9zdy**
>
> **Q.** The authors should perhaps give some idea of the computational costs of their methods. I am also not sure why in Figure 2 the cyan dots (in the lower panel) have much smaller variance than the red ones.
>
> **A.** Thank you very much for the great comments.
>
> - The computational complexity depends on the specific form of the regularizer.
> - For covariance velocity (8), it is $O(NTD^2)$, where $N$ is the number of sampled trajectories, $T$ is the number of sampled time point, and $D$ is the number of variables (dimensionality).
> - The variance can be either larger or smaller depending on many reasons, as shown in more visualizations in Appendix I.
>
> ------------
>
> **Q.** What happens if the covariance structure undergoes a change over time? Can the method adapt to this scenario? Have you considered things like normalizing flows as a comparison?
>
> **A.** This is a wonderful idea.
>
> - The method can be incorporated into a dynamically evolving covariance structure.
> - To incorporate it, we need only specify that $L_pop$​ in equation (13) is also dependent on the index $t$.
> - In our paper, the normalizing flow is compared under a different name, for example, OT-Flow in Table 2.
> - In the future, we can also combine normalizing flow and population regularization for a potentially more effective dynamics modeling.

---

> > ### Comment · Reviewer_9zdy · 2024-11-27
> >
> > Thank you for clarifying these points, I will retain my score as of now.

---

### Meta-Review · Area_Chair_J8F9 · 2024-12-19

**Metareview:**

The paper proposes a new way to regularize the population dynamics (time-evolution of a distribution) by proposing population-level regularizers. That is, instead of defining the total action of the evolution as an average over individual Lagrangians
$$\text{Individual-level action} = \int dt \\ \mathbb{E}_{x_t \sim q_t}\mathcal{L}(x_t,\dot{x}_t,t),$$
where $q_t(x)$ is the population-density depending on time $t$ and $\dot{x}_t$ is the corresponding velocity, the authors propose to define the Lagrangians that depend on the population density as
$$\text{Population-level action} = \int dt \\ \mathcal{L}(q_t,\dot{q}_t,t),$$
and potentially some derivatives.

The authors propose to apply this formalism to describe the single-cell dynamics, which is essentially population-based (due to inaccessible trajectories). For the evaluation, they tested the dynamics for leave-one-out marginals, as was previously studied in the literature.

All the reviewers agree that the work is promising. However, the work cannot be published in its current state due to the presentation issues and the lack of a clear motivation for the proposed developments. In particular, the reviewers do not see the merit of the proposed developments beyond this particular problem, and, at the same time, the lack of domain expertise does not allow us to appreciate the progress made in modelling the developmental process of cells.

**Additional Comments On Reviewer Discussion:**

The reviewers did a great job explaining their concerns and proposing different ways to strengthen the paper. Most of the concerns were regarding the presentation and motivation. The authors partially addressed the concerns by updating the manuscript and providing a respectful and detailed rebuttal. Namely, the authors clarified the concerns of Reviewer wXGs but didn't manage to do so for Reviewers 38fi and Rv9f. I hope this encourages the reviewers to put extra effort into communicating and motivating their conceptually dense work.

---

### Decision · Program_Chairs · 2025-01-22

Reject